

# Backpropagation Neural Network as Earthquake Early Warning Tool using a new Elementary Modified Levenberg–Marquardt Algorithm to minimise Backpropagation Errors

**Jyh-Woei Lin, Chun-Tang Chao, Juing-Shian Chiou**

Department of Electrical Engineering, Southern Taiwan University of Science and Technology, No. 1, Nan-Tai Street, Yungkang Dist., Tainan City, Taiwan

*Correspondence Author*: Juing-Shian Chiou (jschiou@stust.edu.tw)

**Abstract.** A new Elementary Modified Levenberg–Marquardt Algorithm (M-LMA) was used to minimise backpropagation errors in training a backpropagation neural network (BPNN) to predict the records related to the Chi-Chi earthquake from
four seismic stations, Station-TAP003, Station-TAP005, Station-TCU084 and Station-TCU078, with the learning rates of 0.3, 0.05, 0.2 and 0.28, respectively. For these four recording stations, the M-LMA has been shown to produce smaller predicted errors compared to LMA. A sudden predicted error could be an indicator for Early Earthquake Warning (EEW), which indicated the initiation of strong motion due to large earthquakes. a trade-off decision-making process with BPNN (TDPB), using two alarms, adjusted the threshold of the magnitude of predicted error without a mistaken alarm. This approach was
not necessary to consider the problems of characterising the wave phases and pre-processing, but did not require complex hardware; an existing seismic monitoring network-covered researched area was already sufficient for these purposes.

## 1 Introduction

Optimal weight and bias calculation using backpropagation correction in a neural network is commonly known as a
backpropagation neural network (BPNN) (Fukushima, 1980). The traditional Levenberg–Marquardt Algorithm (LMA) (Levenberg, 1944) determines the desired minimum error by locating the minimum of a multivariate function of an independent variable, expressed as the sum of the squares of nonlinear real-valued functions. While the traditional LMA serves as a backpropagation correction to train a BPNN, it cannot update two independent variables, i.e. weight and bias, simultaneously. The process of updating the weight and bias simultaneously is called the parallel distributed processing
(PDP) (Finsterle and Kowalsky, 2010). Such previous processing of LMA is not like the operation of a biological neuron, because a biological neuron operates using PDP (Ferrier, 1876). However, LMA has become a popular method for general nonlinear least squares problems when encountering rank-deficient nonlinear least squares – for example, Texas Hold'em and Telltale Texas Hold'em – which have badly behaved data and bad databases. When ill-conditioned data are encountered, the global minimum could be easily accessed after a single completed iteration using LMA (Eslamian, 2014). In this study, a

new modified Elementary Levenberg–Marquardt Algorithm (M-LMA) with PDP is employed to determine the desired minimum error in the backpropagation correction algorithm of the BPNN to predict records of stations belonging to a seismic monitoring network by implementing an adaptive learning rate, wherein the learning rate was varied depending on the convergence of the objective function. Related to this topic, Naveen et al. (2010) used a type of classical LMA for

inverse problems. Chen. (2016) also used another type of classical LMA with line search for nonlinear equations. He corrected the computation style of the classical LMA, wherein at every iteration, both an LMA step and two-additional approximating LMA steps were computed in order to save the Jacobian calculation and employ line search for the step size. Their results were very efficient and saved many Jacobian calculations. These methods have modified the classical LMA, but their performances did not employ PDP. An earthquake early warning (EEW) system is a warning issued whenever an

earthquake is detected. The Japan Meteorological Agency (JMA) proposed an EEW system, assisted by a combined system of accelerometers, seismometers, communication, computers, and alarms devised for regional notification of a strong earthquake while it is in progress (Wu and Teng, 2002; Allen and Kanamori, 2003; Wu and Kanamori, 2008). Wu and Teng (2002) employed the Rapid Earthquake Information Release System (RTD) and virtual subnetwork (VSN) system-hardware for EEW. The VSN system must wait for S wave records from remote stations, and therefore introduces problems for a clear

indication of the S wave. Allen and Kanamori. (2003) used data from previous earthquakes during the installation of the Earthquake Alarm System (ElarmS), to serve the function of an EEW in Southern California. However, in cases where data from previous earthquakes was not assembled correctly, the effectiveness of the EEW system was compromised. Wu and Kanamori (2008) reported that some parameters are necessary for EEW, e.g. the magnitude and strength of shaking in the initial P wave. Unfortunately, obtaining a clear indication of the initial P wave is not trivial similar to those outlined by the

work of Wu and Teng. (2002). Failure to derive the initial P wave could affect the ability to conduct EEW for the records of nearby recording stations. Pre-processing, such as filtering out noise, could perhaps be used to assist in characterising the initial P wave; however, this might also require complex additional equipment.

Artificial neural networks (ANN) could be used for the EEW; Gentili and Michelini. (2006) designed automatic picking of P and S wave phases using artificial neural networks for EEW by training the network using 342 earthquakes recorded by

23 different stations (about 5000 traces). Pre-processing was necessary for this method, and a failure could affect the ability of an EEW if the traces have high noises. Böse et al., (2008) developed a method for EEW called PreSEIS (Pre-SEISmic) based on single station observations applied to the Istanbul Earthquake Rapid Response and EEW System (IERREWS). A two-layer feedforward neural network was used to estimate the earthquake hypocenter location, its moment magnitude, and the expansion of the evolving seismic rupture that could lead to clear alert maps before the arrival of seismic waves.

However, when the estimated errors of the hypocenter location, moment magnitude and the expansion of the evolving seismic rupture were large, the EEW as a whole would become uncertain due to the complicated faults. Arjun and Kumar. (2009) estimated the peak ground acceleration (PGA), which could enhance the ability of EEW by including seismic data from the earthquakes with magnitudes greater than 5.0, however, earthquake magnitude, hypocentral distance must be precisely determined as possible. The processing was also complicated. Kuyuk et al., (2014) designed a network based on



the EEW algorithm for California called ElarmS-2. This algorithm had complicated processes and the P wave parameters must be triggered. An artificial neural network was only processed clearly identified P wave parameters.

For a special work, Wu et al., (2013) developed a robust automated decision process called the earthquake probability-based automated decision-making (ePAD) framework. The ePAD Framework was used to broadcast a warning of the predicted location and magnitude shortly before an earthquake hits a site as part of the EEW in California: CISN ShakeAlert System. The ePAD Framework is a robust automated decision process, however, the location and magnitude shortly before a large earthquake must be predicted, and then ePAD Framework would make an action decision – these processes were complicated. Moreover, probability in the ePAD framework is represents a log-normal distribution, which has an exceedance probability and therefore the ground motion parameters e.g. return periods (Peres and Cancelliere, 2016) and intensities require determination. However, these parameters are uncertain and difficult to identify (Pavlenko, 2017; Yazdani, 2018). However, return periods were changed recently due to some factors e.g. global changes (Brown, et al., 2008; Read and Vogel, 2015), and therefore the return period, as an estimated parameter for EEW was already uncertain.

The aim of this paper is to determine whether the EEW develops a better real-time and on-line performable training method in BPNN. The microseismic data in the records are used as training data for the BPNN model; in each station shown, the behaviour of microseismic data at each station records the ray tracing path, allowing for the prediction of upcoming signal. When the large predicted errors are presented, then it is expected that the behaviour of microseismic data has changed. In this situation, it is possible that these errors record the initiation of strong motions due to a large earthquake. Therefore, this method could be used as part of the EEW when the EEW is not validated for proximal receiver stations, e.g. some mistaken wave phases using other methods, and the installation of additional seismic monitoring network is not necessary (Wu and Teng, 2002; Allen and Kanamori, 2003; Gentili and Michelini, 2006; Böse et al, 2008; Wu and Kanamori, 2008; Kuyuk et al, 2014). The seismic receiver stations belong to an existing seismic monitoring network called Free Field Strong Earthquake Observation Network, and the P, S and surface wave phases in their records may be identified (Central Weather Bureau, Taiwan, CWB). Even though identifying the wave phase is not necessary, as previous stated, when using the method in the study. Expectedly, a certain anomalous predicted error may be an indicator of the initiation of strong motions due to a large earthquake. The study examines the Chi-Chi earthquake, which on September 21, 1999 (TST), caused by slip on the Chelungpu fault with corresponding parameters shown in Figure 1. In this figure, four corresponding seismic records of the stations are used by the BPNN for prediction. These records are ground accelerations because they are the primary concept used to define the seismic intensity scales, which are used to represent the degree of seismic hazard. Therefore, these records are used for EEW. Two corresponding, distant stations are close to Taipei city, and two stations are close to the epicentre. For the two far stations, one is Station-TAP003 with the record shown in Figure 2a. This station is located at the coordinates (25.08° N 121.45° E), and is indicated in Figure 2b by a dark blue coloured spot and another seismic station (Station-TAP005) located at (25.11° N 121.50° E) is shown by a baby blue coloured spot. For the two closer stations, Station-TCU084 is recorded by a yellow coloured circle in Figure 2c at the coordinates (23.88° N 120.90° E). The other, Station-



TCU078, is very close to the epicentre, and its record is shown in Figure 2d. The green coloured circle at coordinate (23.81° N 120.84° E) indicates Station-TCU078. The sampling rate of the records of these four stations is 200 (Hz).

**2 Modified Elementary Levenberg–Marquardt Algorithm (M-LMA)**

Lin (2017) performed the BPNN to predict the Physionet EMG Signals. In that study, a modified M-LMA served as the

backpropagation correction when training the BPNN. In this section, a newer modified M-LMA, called a modified elementary M-LMA, is introduced with the concept of an area element to extend simulation e.g. for some surface problems. This algorithm is a modified version of the M-LMA used in Lin's study (2017). The algorithm can be transformed to a backpropagation correction in BPNN and is expected to have smaller predicted errors, following Lin's study (2017). In three-dimensional space, represented by three Cartesian coordinates (Descartes, 1667), a function of $F$ is composed of two

independent variables $x$ and $y$ . For simulated surface problems, the function $F$ generates

$Z_i = F(x_i, y_i), i = 1,2....$ . The initial codomain is i=1. $Z_t = F(x_t, y_t)$ is defined as a target output (Rumelhart

and McClelland, 1986). When $x^o$ and $y^o$ best satisfy the surface function to minimise $\varepsilon^T \varepsilon$ , the error is $\varepsilon = Z_i - Z_t$.

$\delta xy$ and can be defined as an area element when simulating a surface. Therefore, a Taylor-series expansion with two

variations ($\delta x$ and $\delta y$ ) approximates $F$ as follows;

$F(x + \delta x, y + \delta y) \approx F(x, y) + J_a \delta xy$            (1)

where $J_a$ is defined as the Jacobian matrix with $\dfrac{\partial F(x, y)}{\partial xy}$ ; a series of ($x_1, y_1$), ($x_2, y_2$), ($x_3, y_3$) is produced

and converges toward a local minimiser $Z^o = F(x^o, y^o)$ , called an optimised output, so that

$\|Z_k - F(x + \delta x, y + \delta y)\| \approx \|\varepsilon - J_a \delta xy\|$ can be estimated. When $\varepsilon - J_a \delta xy$ is orthogonal to the column space of

$J_a$ and $J_a^T(\varepsilon - J_a \delta xy)$, $I$ is defined as an identity matrix. A formula is proposed to import a parameter of $r$ as

follows;

$\delta xy(rI) + J_a^T J_a \delta xy = J_a^T \varepsilon$            (2)

Finally, Formula (2) becomes;

$(I_r + H_a)(\delta xy) = J_a^T \varepsilon$            (3)

here $H_a = J_a^T J_a$ is defined as the approximated Hessian matrix and $rI = I_r$ , where $r$ is the learning rate. In

BPNN, if the learning rate is too high, the system will either oscillate about the true solution, or it will diverge completely. If the learning rate is too low, the system will take a long time to converge on the final solution. In this study, based on the



concepts in this section, the parameters $x$ and $y$ can serve as the weight and bias that can be simultaneously optimised with PDP in a BPNN (Fukushima, 1980) because of the concept of an area element, for a specially designated area element $\delta xy$ used to simulate a optimised surface. Therefore when an $x$ value is altered by a $y$ value due to a specially designated relationship, real simultaneity can be achieved to simulate optimised area elements $\delta xy$ (Abbena, et al., 2006).

This means that when an area element $\delta xy$ is designated, then the $x$, $y$ values related to $\delta xy$ are simultaneously designated. Finally, an optimised surface is obtained from these elements. When simulating a surface with BPNN and LMA, which is a function of a variable (Lin, 2017), it serves as a backpropagation correction to. A supposed initial $x$ value is offered and is updated by a weight. The bias updates the $y$ value. Both update processes are independent; without the concept of an area element, these processes are not simultaneous. In this situation, a simulated surface is obtained through
simulating two nonlinear real-valued curves. One corresponds to the weight-updated $x$ value, another corresponds to $y$ value, which has been updated by bias. Therefore, LMA is not a PDP for simulating a surface.

## 3 Results

The results of many theoretical researches and engineering works regarding simulations have shown that using a two hidden layer network, with a small number of neurons in each layer, can replace a large number of neurons in a hidden layer
network (Wagarachchi and Karunananda, 2014). The basic mathematical framework of an artificial neural network (ANN) was already introduced in section two of the study by Lin (2017). Using the previously mentioned network framework, microseismic data, from the four stations stated in section one, serves as training data to build the new BPNN models after testing with different learning rates between 0 and 1, with an increment of 0.01, from which the magnitude of upcoming microseismic data will be predicted. The M-LMA introduced in section two is used as the algorithm of backpropagation
correction. For comparison, the LMA was simultaneously used as the algorithm of backpropagation correction. For both algorithm backpropagation, the initial weights and the initial biases were set to random variables (Nguyen and Widrow, 2009), and then feature scaling was performed (Bo et al, 2006; Xie et al, 2016). Using feature scaling, the variables were in the range of 0 and 1 because the value of the sigmoid function, called the activation function, which was used to train the BPNN in this study, was in the range of 0 and 1. Randomly distributing the weights between 0 and 1 helped prevent biases
toward any particular output. If initialisation was non-random, the network would consistently and prematurely connect to certain outputs that were undermining the training. The 1000 epochs were given for sample signals in the seismic records (Sinha, et al., 2010). The vertical component of an earthquake was the most dangerous (Douglas, 2003). Therefore, these four vertical component, taken from the records, were predicted by the previously mentioned parameters and framework as a BPNN, using two hidden layers with 10 neurons in each layer to update the weight and bias to minimise backpropagation
errors; these learning rates were found to be the best for minimising backpropagation errors. In this situation during training



processing, the weights and biases would be updated, in which the neurons of BPNN were learning through updating the weights and biases as a PDP. The target output was defined as the size of present signal at a last time point in the part of microseismic data of the records. The predicted error was defined as "target output subtracting expected output". Therefore, the present time of the expected output, which was the predicted  signal, was prior to the upcoming real signal. An expected

output with a minimised predicted error was trended to give to build a BPNN model after training. Finally, when a BPNN model was used to predict the upcoming real signal,  the designed instrument, including the hardware and software, were controlled to validate the present time of the predicted outputs between the two real seismic signals. Therefore, the computing speed of the designed instrument, which usually depended on the selected epoch in software (Sinha, et al., 2010) - e.g. Matlab program - and hardware e.g. computer situation, such as the temperature of CPU of computer, must be fast and

stable to achieve this goal.

Figure 3a has shown the predicted results of LMA and M-LMA with the same learning rate of 0.3 for records from Station-TAP003. The predicted results of M-LMA were better, with smaller predicted errors, compared to the results of LMA. Simultaneously, M-LMA was also shown to have better training with the microseismic data in the BPNN in order to produce a better BPNN model, and its processing was more similar to a biological neuron network with the PDP. Usually

when the predicted errors were too large, then the predicting was lost, resulting in incorrect predicted upcoming data. Therefore, the results of the M-LMA were more meaningful and a significance was given for the predicting. When any anomalous output occurred - e.g. sudden amplified output with larger predicted error at approximately 9.5s, prior to the strong motion at about 15s, with a warning time of 5.5s. It should be a good idea using larger predicted errors as the predictors prior to the strong motion. Figure 3b presented the predicted results of the LMA and M-LMA, based on the

records from Station-TAP005 using the best learning rate of 0.05 for both methods. The predicted results of the M-LMA were superior to those of the LMA, with smaller predicted errors. A sudden amplification in the output with large predicted errors occurred at approximately 9s, which was nearly simultaneous with the start of strong motion and the first strong motion alert. Supposed the larger predicted error amplitudes, beginning from about 22s, was seriously dangerous and a second strong motion alarm was sent, with a warning time of 13s. Therefore, when predicting a record, the results of

predicting must be similar to the larger amplitudes of predicted errors to avoid an erroneous secondary alarm. With this concept, a decision-making process using a suitable threshold of predicted error magnitudes at each station of an existing seismic monitoring network, as second alarms from different stations was necessary. A decision-making process called "trade-off decision-making process with BPNN (TDPB)" was performed. In this study, the past records of the Chi-Chi earthquake was examined by TDPB, and then the thresholds and were subjectively determined.

Different stations had different thresholds, and the selections of their suitable thresholds for secondary alarms should be calculated after being subject to significant testing in tandem with many stations in the same existing seismic monitoring network and after observing the behaviours of waveforms for these records according to the many past earthquakes, including related microseismic, using some methods. These suggested methods included nonlinear dynamic aseismic diagnosis (Xu et al, 2015; Ouazzani et al., 2017), surveying of local geological conditions near each station including the



consideration of local building damage from past events that were evaluated by the earthquake-resistant and seismic coefficients, and seismic capacity evaluation of existing reinforced concrete buildings (Moustafa, 2015**).**

Figure 3c presented the predicted results of the LMA and M-LMA from the proximal Station-TCU084 using the best learning rate for both methods, determined to be 0.2. The predicted results of the M-LMA had smaller predicted errors. A sudden amplification in the output, with large predicted error, occurred at approximately 22s, which was almost simultaneous with the start of strong ground motion. The larger amplitudes beginning at about 34s could be seriously dangerous and the warning time was 12s. Figure 3d presented the predicted results of the LMA and M-LMA from near Station-TCU078 using a best learning rate of 0.28 for both methods. The predicted results of the M-LMA, similar to the other stations, also had smaller predicted errors. A sudden amplification in the output, with large predicted error, occurred at approximately 21s, which coincided with the start of strong motion. When large amplitudes beginning at around 27s met real serious damages, and the warning time became 6s. The TDPB was applied to avoid sending a false strong ground motion alarm, similarly to the aim, previously stated about the decision-making process (Rath et al., 2017). For example, from the record of Station-TAP005 in Figure 3b, a smaller predicted error occurred at about 9s with the first alarm of strong motion, and through adjusting the threshold of the magnitude of the predicted errors, an alarm was sent at about 22s, as previously stated, with a second alarm of strong motion in order to have a warning time of 13s. When a sudden predicted error was first presented at a time point, and no sudden predicted error appears later, it would become misinformation, resulting in being defined as a "False alarm (nuisance alarm)" (Rath et al., 2017). The TDPB was suitable to solve this problem with a second alarm, similar to the processing of Iervolino et al (2007). Therefore, for this topic, future research was necessary. If the seismic records record strong ground motion at 22.5s, without a microsseismic data, larger predicted errors would be found. Therefore, for this case, the microseismic data were necessary to train a BPNN model; the EEW was not validated without training microseismic data because the sudden predicted error appeared, and it had no warning time. Figure 3e has shown the predicted results of the LMA and M-LMA on the signals of Station-TCU084 with the same learning rate of 0.28. As previously stated, the processing of the method in this study was supplemented. The 1000 epochs were selected for Matlab 2013a in windows 10 as the software, the computer as hardware - as in Figure 4 - and seismic records with same sampling rate of 200Hz.

This BPNN approach was well-suited, and it was not necessary to consider the problems of characterising the wave phases and pre-processing, as stated previously. Furthermore, BPNN was a mature technology, which was expected to develop rapidly in the future, and did not require complex hardware. Determining an initial location and magnitude of the event was not necessary for this technique. An existing seismic monitoring network e.g. Free Field Strong Earthquake Observation Network of CWB was already sufficient for these purposes. At each station in the monitoring network, adjusting the different leaning rates can minimise the predicted error. Therefore, all of the training records in the monitoring network would cause different station to have different learning rates. Finally, these BPNN models were built using the past microseismic data for sending second alarms through a decision-making process that belonged to post-mortem predicting. Therefore the warning time was afterwards retrieved. As stated previously, future research was necessary to apply real-time





microseismic data to build the corresponding BPNN models. Such as nonlinear dynamic aseismic diagnosis may lead into TDPB so that warning time was subjectively determined as soon as. However, This study has proposed this possibility for EEW system which was different from previous works in section one with the results of four stations..

## 4 Conclusions

M-LMA was determined to be better for predicting the vertical component of the Chi-Chi earthquake, using a learning rate of 0.3 for Station-TAP003. An anomalous output with larger predicted error was detected at about 15s, and a sudden amplified output with smaller predicted error occurred at approximately 9.5s prior to the strong motion due to Chi-Chi earthquake. The warning time was 5.5s. However, from the predicted results of the M-LMA with a learning rate of 0.05 related to Station-TAP005, a sudden amplified output was detected with large predicted error. This occurred at

approximately 9s, almost simultaneously with the initiation of strong motion. In this situation, using the TDPB , the large amplitudes starting at about 22s had a warning time of 13s. For Station-TCU084, which used a learning rate of 0.2, a sudden amplified output, with large predicted error, occurred at approximately 22s, almost simultaneously with the starting of strong motion. When the TDPB was applied to larger amplitudes at about 34s, the warning time became 12s. For Station-TCU078, with a learning rate of 0.28, a sudden amplified output with large predicted error occurred at approximately 21s. This

coincided with the start of strong ground motion. When the TDPB was applied with larger amplitudes at about 27s, and the warning time became 6s. For these four recording stations, the M-LMA has been shown to produce smaller predicted errors. When predicting the records for these four stations, the sudden predicted errors, just after the microseismic data, could be considered as the beginning of the strong motion. Therefore, this method could serve as a real-time and on-line EEW tool, using an existing seismic monitoring network to assess the occurring risk of strong motions caused by a large earthquake,

and considering the problems of characterising the wave phases and pre-processing was not necessary. Complex hardware did not require to setup.

## Acknowledgements

The authors are grateful to the support of data source of from Central Weather Bureau, Taiwan (CWB) and supported by the Ministry of Science and Technology, Taiwan, under Grant nos. 106-2221-E-218 -001 -MY2.

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





**Figure Captions**

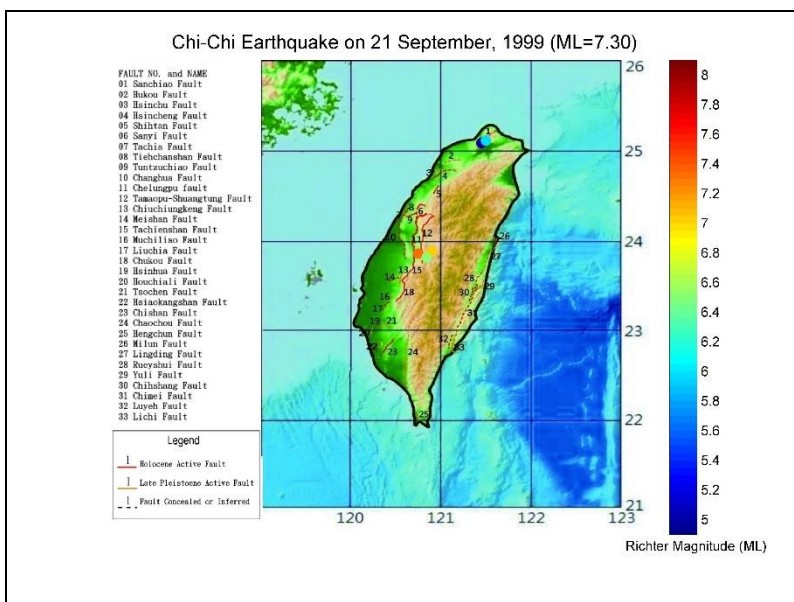

Figure 1 The figure shows the position of Chelungpu fault (No.11) on a map of Taiwan. Slip on this fault caused the Chi-Chi earthquake, which occurred at 01:47:15 on September 21, 1999 (TST), at a depth of 8.00 km, with a Richter magnitude (ML) of
5   7.3. The epicentre was at the coordinates (23.85° N, 120.82° E) (Orange-colour spot near the Chelungpu fault for No.11). The four corresponding positions of the research stations are shown by a dark blue coloured spot (Station-TAP003), baby blue coloured (Station-TAP005) spot, yellow coloured spot (Station-TCU084) and green coloured spot (Station-TCU078) in this figure. Station-TCU078 is very close to the epicentre.

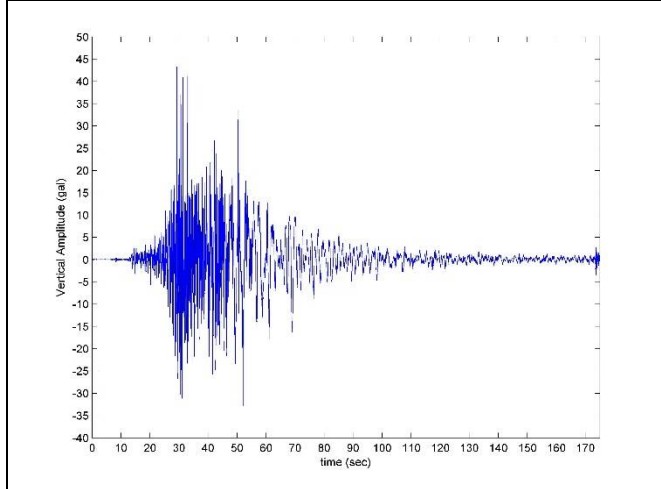

10   Figure 2a This figure shows the vertical component of the Chi-Chi earthquake. The unit of the record is gal ($cm$ / $s^2$ ). This was recorded by Station-TAP003 (25.08° N 121.45° E).





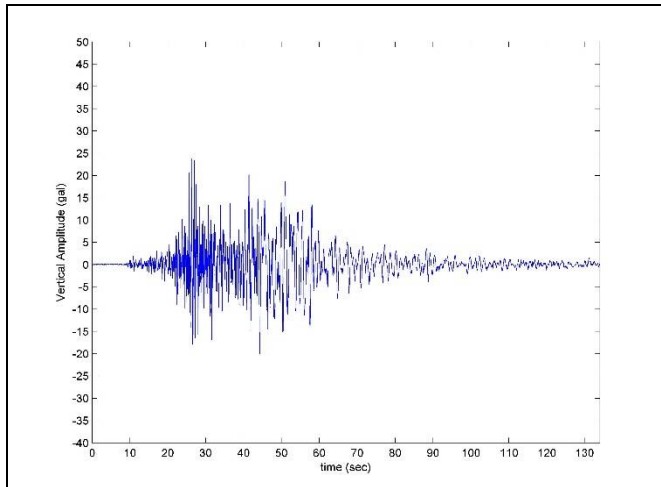

**Figure 2b This figure shows the vertical component at Station-TAP005 (25.11° N 121.50° E) related to the Chi-Chi earthquake.**

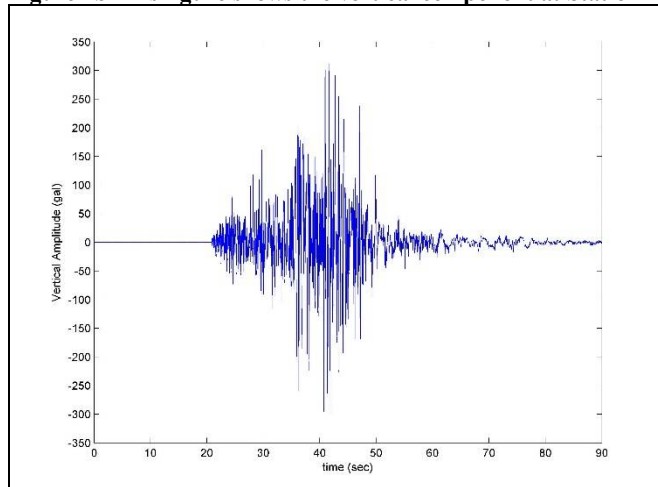

**Figure 2c This figure shows the vertical component at Station-TCU084 (23.88° N 120.90° E) related to the Chi-Chi earthquake.**





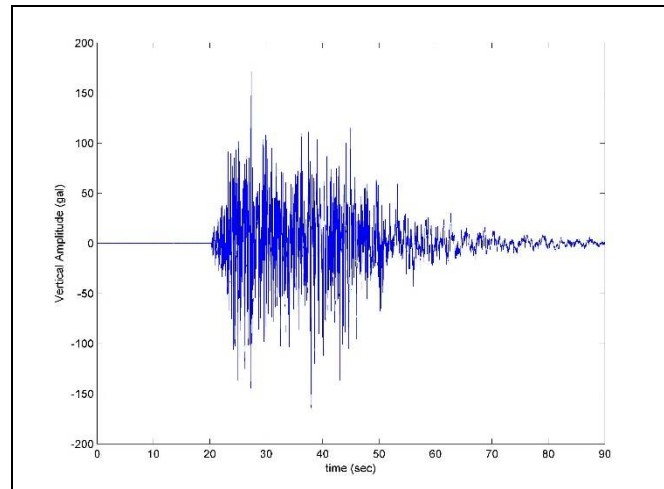

**Figure 2d This figure shows the vertical component at Station-TCU078 (23.81° N 120.84° E) related to the Chi-Chi earthquake.**

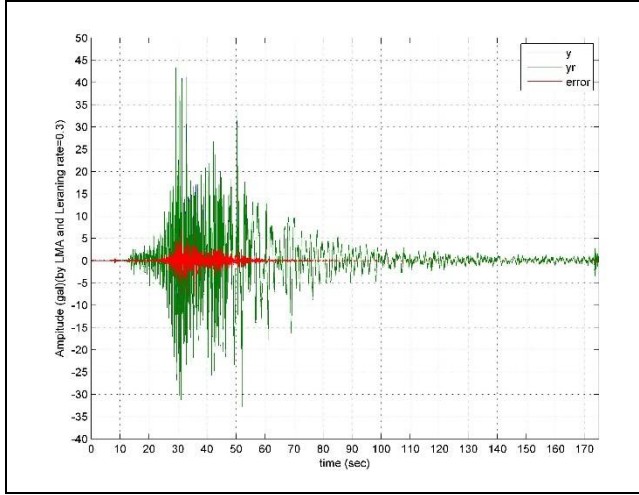

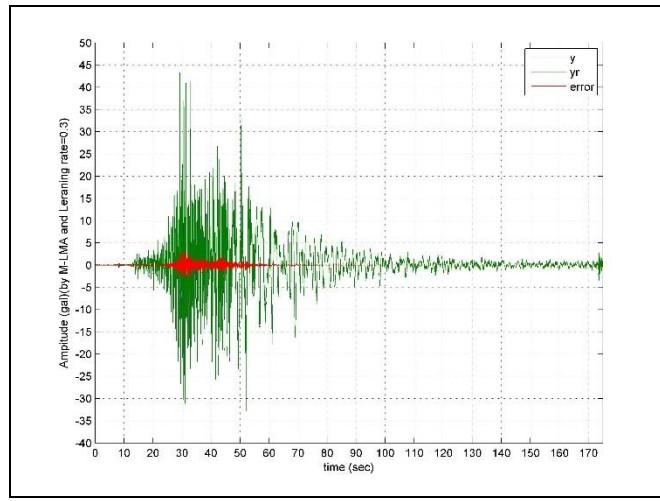

**Figure 3a These two figures show the predicted results of the LMA and M-LMA of the signals shown in Figure 2a, with a learning rate of 0.3. The predicted errors of the M-LMA are smaller than the LMA. The blue lines indicate the signals in Figure 2a. The units are gal ( $cm/s^2$ ). The green lines indicate the outputs of the BPNN. The red lines indicate the predicted errors.**

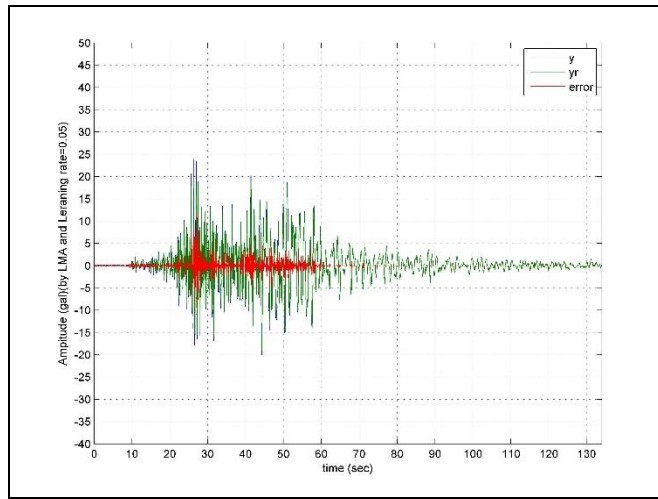

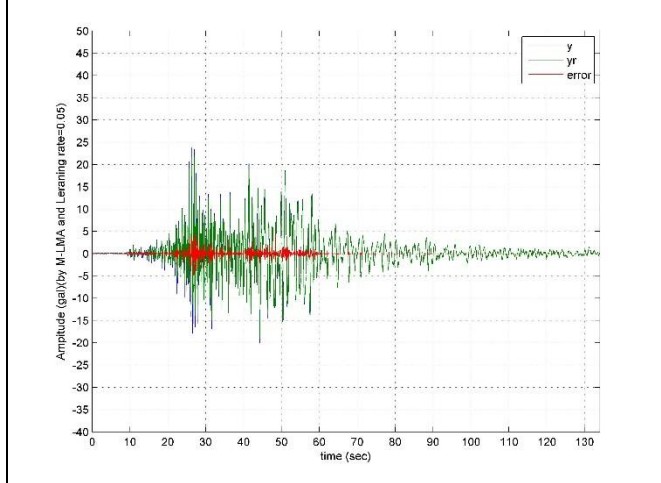

**Figure 3b These two figures show the predicted results of the LMA and M-LMA of the signals shown in Figure 2b with a learning rate of 0.05. The predicted errors of the M-LMA are smaller. The blue lines indicate the signals in Figure 2b. The green lines indicate the outputs of the BPNN. The red lines indicate the predicted errors.**





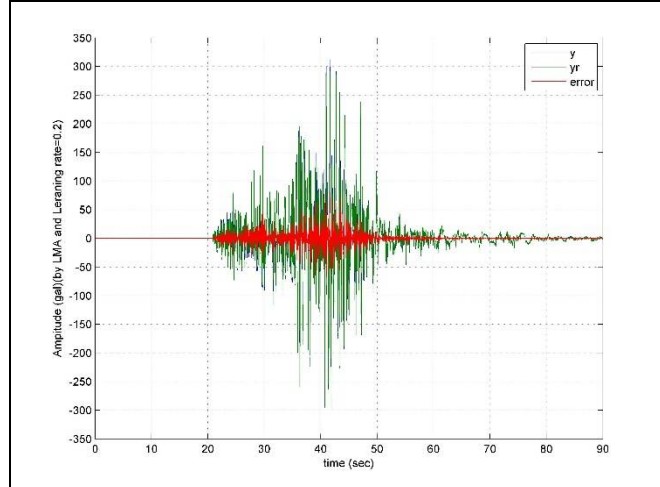

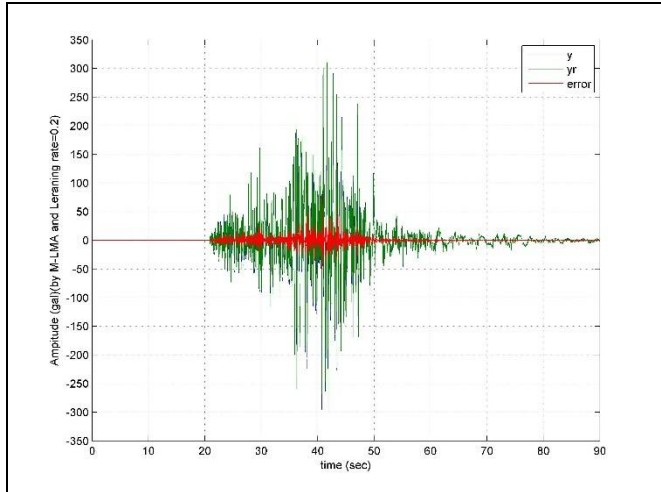

**Figure 3c The predicted results of the LMA and M-LMA of the signals shown in Figure 2c using the same learning rate of 0.2. The predicted errors of the M-LMA are smaller. The blue lines indicate the signals in Figure 2c. The green lines indicate the outputs of BPNN. The red lines indicate the predicted errors.**





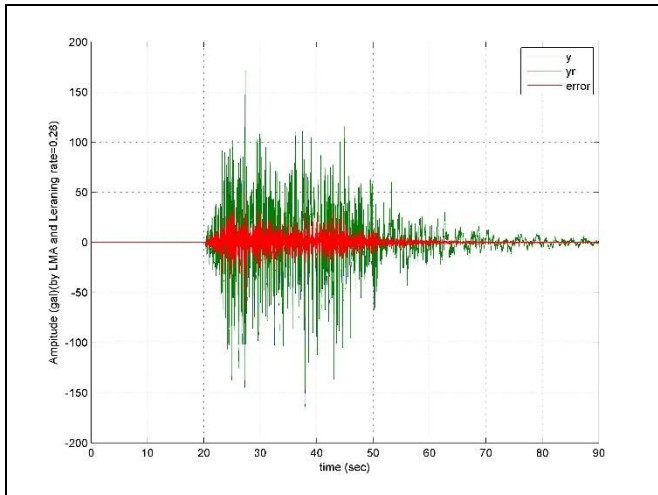

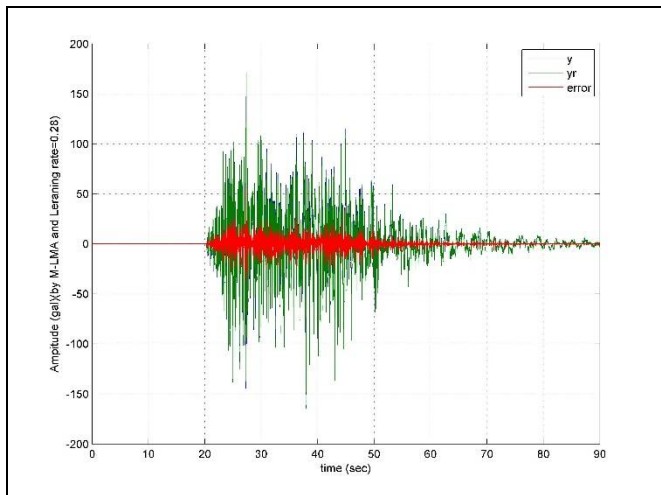

**Figure 3d Predicted results of LMA and M-LMA for the signals shown in Figure 2d, with a learning rate of 0.28. The predicted errors of the M-LMA are smaller. The blue lines indicate the signals in Figure 2d. The green lines indicate the outputs of the BPNN. The red lines indicate the predicted errors.**





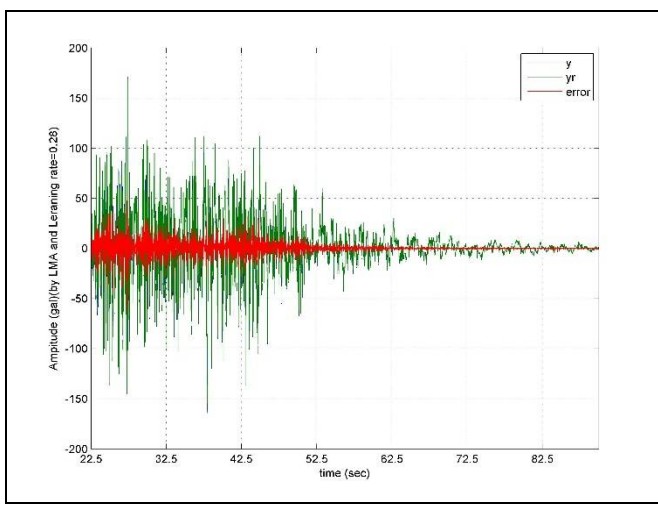

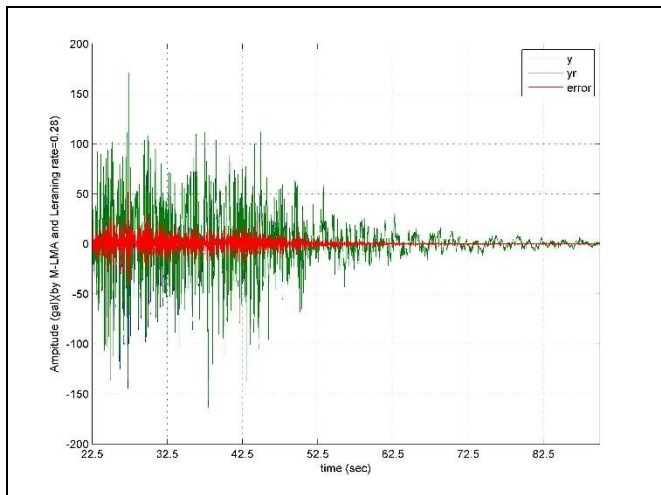

**Figure 3e These two figures show the predicted results of the LMA and M-LMA for the signals shown in Figure 2d, with a learning rate of 0.28. However, the records are predicted at 22.5s, after the beginning of strong motions due to the Chi-Chi earthquake. The predicted errors of the M-LMA are still smaller. The blue lines indicate the signals in Figure 2d. The green lines indicate the outputs of the BPNN. The red lines indicate the predicted errors.**