# Peer review of "Backpropagation Neural Network as Earthquake Early Warning Tool using a new Elementary Modified Levenberg–Marquardt Algorithm to minimise Backpropagation Errors"

_Geoscientific Instrumentation, Methods and Data Systems, 2018_

## Referee Comment (RC1) · Anonymous Referee #1 · 20 May 2018

Dear Author After I have read your paper, This paper can be accepted for publication, after address my questions (1) In page 3 You wrote ================= The aim of this paper is to determine whether the EEW develops a better real-time and on-line performable training method in BPNN. The microseismic data in the records are used as training data for the BPNN model; in each station shown, the behaviour of microseismic data at each station records the ray tracing path, allowing for the prediction of upcoming signal. When the large predicted errors are presented, then it is expected that the behaviour of microseismic data has changed.

[Figure]

===================== Please explain them more clear? (2) In page 5 you wrote ===================== The vertical component of an earthquake was the most dangerous =========================== Why? (3) In page 6 you wrote ============== surveying of local geological conditions near each station including the consideration of local building damage from past events that were evaluated by the earthquake-resistant and seismic coefficients, and seismic capacity evaluation of existing reinforced concrete buildings ============ These mention are not logic! That is: local geological conditions including the consideration of local building damage? no

---

## Author Comment (AC1) · 22 May 2018

Dear Reviewer#1 Thank you for your comments. Now I address your comments point-by-point and marked the changes with red words. (1) I wrote more clear as follows; The aim of this paper is to determine whether the EEW can be on the stage by a better real-time and on-line performable training method in BPNN than the past works as stated previously. The microseismic data in the records are firstly used as training data for the BPNN model; in each station shown, the behaviour of microseismic data at each station records the ray tracing path, allowing for the prediction of upcoming

signal. When the large predicted errors are presented, then it is expected that the behaviour of the microseismic data has changed because of this model reflecting the pattern of microseismic data. (2) Because the earthquake forces mostly acted on the center of gravity of the sliding soil mass, and the influences of vertical ground motions were on the seismic-induced displacements of the structures. Therefore I wrote the reasons more clear as follows; The vertical component of an earthquake was the most dangerous because the earthquake forces mostly acted on the centre of gravity of the sliding soil mass, and the influences of vertical ground motions were on the seismic-induced displacements of the structures (Sawicki, et al. 2007; Zhao, et al. 2017). (3) I re-wrote in the text as follows; surveying of the consideration of local building damages from past events under different local geological conditions. By the way in section 2, for reader to understand clearly, I add some statements with red words. I also upload the revise paper.

Jyh-Woei Lin, Chun-Tang Chao, Juing-Shian Chiou 23, May, 2018, Taiwan

Please also note the supplement to this comment:
https://www.geosci-instrum-method-data-syst-discuss.net/gi-2018-13/gi-2018-13-AC1-supplement.pdf
* * *
[Figure]

**Supplement:**

[revised manuscript text omitted]
 instead of using line element in LMA, which is a function of an independent variable as stated previously.

10 This algorithm is a modified version of the M-LMA used in Lin's study (2017). The algorithm can be transformed to a backpropagation correction in BPNN and is expected to have smaller predicted errors, following Lin's study (2017). In three-dimensional space, represented by three Cartesian coordinates (Descartes, 1667), a function of $F$ is composed of two independent variables $x$ and $y$. For simulated surface problems, the function $F$ generates $Z_i = F(x_i, y_i), i = 1,2....$ . The initial codomain is i=1. $Z_t = F(x_t, y_t)$ is defined as a target output (Rumelhart and McClelland, 1986). When $x^o$ and $y^o$ best satisfy the surface function to minimise $\varepsilon^T \varepsilon$, the error is $\varepsilon = Z_i - Z_t$.

15 $\delta xy$ and can be defined as an area element when simulating a surface. Therefore, a Taylor-series expansion in two variations ($\delta x$ and $\delta y$) approximates $F$ as follows;

$$F(x + \delta x, y + \delta y) \approx F(x, y) + J_a \delta xy \tag{1}$$

where $J_a$ is defined as the Jacobian matrix with $\dfrac{\partial F(x, y)}{\partial xy}$; a series of ($x_1, y_1$), ($x_2, y_2$), ($x_3, y_3$) is produced

20 and converges toward a local minimiser $Z^o = F(x^o, y^o)$, called an optimised output, so that $\|Z_k - F(x + \delta x, y + \delta y)\| \approx \|\varepsilon - J_a \delta xy\|$ can be estimated. When $\varepsilon - J_a \delta xy$ is orthogonal to the column space of $J_a$ and $J_a^T(\varepsilon - J_a \delta xy)$, $I$ is defined as an identity matrix. A formula is proposed to import a parameter of $r$ as follows;

$$\delta xy(rI) + J_a^T J_a \delta xy = J_a^T \varepsilon \tag{2}$$

25 Finally, Formula (2) becomes;

$$(I_r + H_a)(\delta xy) = J_a^T \varepsilon \tag{3}$$

here $H_a = J_a^T J_a$ is defined as the approximated Hessian matrix and $rI = I_r$ , where $r$ is the learning rate. In BPNN, if the learning rate is too high, the system will either oscillate about the true solution, or it will diverge completely. If the learning rate is too low, the system will take a long time to converge on the final solution. In this study, based on the concepts in this section, the parameters $x$ and $y$ can serve as the weight and bias that can be simultaneously optimised with PDP in a BPNN (Fukushima, 1980) because of the concept of an area element, for a specially designated area element $\delta xy$ used to simulate a optimised surface. Therefore when an $x$ value is altered by a $y$ value due to a specially designated relationship, real simultaneity can be achieved to simulate optimised area elements $\delta xy$ (Abbena, et al., 2006).

This means that when an area element $\delta xy$ is designated, then the $x$ , $y$ values related to $\delta xy$ are simultaneously designated. Finally, an optimised surface is obtained from these elements. When simulating a surface with BPNN and LMA, which is a function of a variable (Lin, 2017), it serves as a backpropagation correction to. A supposed initial $x$ value is offered and is updated by a weight. The bias updates the $y$ value. Both update processes are independent; without the concept of an area element, these processes are not simultaneous. In this situation, a simulated surface is obtained through simulating two nonlinear real-valued curves. One corresponds to the weight-updated $x$ value, another corresponds to $y$ value, which has been updated by bias. Therefore, LMA is not a PDP for simulating a surface.

**3 Results**

The results of many theoretical researches and engineering works regarding simulations have shown that using a two hidden layer network, with a small number of neurons in each layer, can replace a large number of neurons in a hidden layer network (Wagarachchi and Karunananda, 2014). The basic mathematical framework of an artificial neural network (ANN) was already introduced in section two of the study by Lin (2017). Using the previously mentioned network framework, microseismic data, from the four stations stated in section one, serves as training data to build the new BPNN models after testing with different learning rates between 0 and 1, with an increment of 0.01, from which the magnitude of upcoming microseismic data will be predicted. The M-LMA introduced in section two is used as the algorithm of backpropagation correction. For comparison, the LMA was simultaneously used as the algorithm of backpropagation correction. For both algorithm backpropagation, the initial weights and the initial biases were set to random variables (Nguyen and Widrow, 2009), and then feature scaling was performed (Bo et al, 2006; Xie et al, 2016). Using feature scaling, the variables were in the range of 0 and 1 because the value of the sigmoid function, called the activation function, which was used to train the BPNN in this study, was in the range of 0 and 1. Randomly distributing the weights between 0 and 1 helped prevent biases toward any particular output. If initialisation was non-random, the network would consistently and prematurely connect to certain outputs that were undermining the training. The 1000 epochs were given for sample signals in the seismic records (Sinha, et al., 2010). The vertical component of an earthquake was the most dangerous because the earthquake forces mostly

acted on the centre of gravity of the sliding soil mass, and the influences of vertical ground motions were on the seismic-induced displacements of the structures (Sawicki, et al. 2007; Zhao, et al. 2017). 
[revised manuscript text omitted]

---

## Referee Comment (RC2) · Anonymous Referee #1 · 23 May 2018

Dear Authors After I checking the answers of my 3 questions, I recommend this paper to publication. It is accepted submission for me. good luck!

---

## Referee Comment (RC3) · Anonymous Referee #2 · 25 May 2018

1. On page 1, no line 13: The word 'a' in the sentence 'a trade-off decision-making process ......' should be capital. 2.The yellow colored spot in figure 1 is too light to be distinguished. 3.Figure 4 which is mentioned in page 7 can't be found in figure captions.
* * *

---

## Referee Comment (RC4) · Anonymous Referee #2 · 26 May 2018

Authors have made a serious answer to my questions. So I recommend that this paper can be accepted for publication. Good luck!

---

## Author Comment (AC2) · 26 May 2018

Dear Reviewer 2 Thank you for you comments. Now I address your comments point-by-point. Author: Lin, Jyh-Woei 26, May, 2018 1. On page 1, no line 13: The word 'a' in the sentence 'a trade-off decision-making process .....' should be capital. ANS: In Abstract, I have change as follows; Abstract. A new Elementary Modified Levenberg–Marquardt Algorithm (M-LMA) was used to minimise backpropagation errors in training a backpropagation neural network (BPNN) to predict the records related to the Chi-Chi earthquake from four seismic stations, Station-TAP003, Station-TAP005, Station-

[Figure]

TCU084 and Station-TCU078, with the learning rates of 0.3, 0.05, 0.2 and 0.28, respectively. For these four recording stations, the M-LMA has been shown to produce smaller predicted errors compared to Levenberg–Marquardt Algorithm (LMA). A sudden predicted error could be an indicator for Early Earthquake Warning (EEW), which indicated the initiation of strong motion due to large earthquakes. A Trade-Off Decision-Making Process with BPNN (TDPB), using two alarms, adjusted the threshold of the magnitude of predicted error without a mistaken alarm. This approach was not necessary to consider the problems of characterising the wave phases and pre-processing, but did not require complex hardware; an existing seismic monitoring network-covered researched area was already sufficient for these purposes. In page 6, line 30, the text is also changed as follows; A decision-making process called " Trade-Off Decision-Making Process with BPNN (TDPB)" was performed. In this study, the past records of the Chi-Chi earthquake was examined by TDPB, and then the thresholds and were subjectively determined.

2.The yellow colored spot in figure 1 is too light to be distinguished. ANS: I have also change in the figure caption for Figure.1 with more clear colors as follows; Figure 1 The figure shows the position of Chelungpu fault (No.11) on a map of Taiwan. Slip on this fault caused the Chi-Chi earthquake, which occurred at 01:47:15 on September 21, 1999 (TST), at a depth of 8.00 km, with a Richter magnitude (ML) of 7.3. The epicentre was at the coordinates (23.85° N, 120.82° E) (Orange-colour spot near the Chelungpu fault for No.11). The four corresponding positions of the research stations are shown by a dark blue coloured spot (Station-TAP003), baby blue coloured (Station-TAP005) spot, red coloured spot (Station-TCU084) and dark red coloured spot (Station-TCU078) in this figure. Station-TCU078 is very close to the epicentre. 3.Figure 4 which is mentioned in page 7 can't be found in figure captions. ANS: I have remove this figure. By Supplement file is my revise paper with figures

Please also note the supplement to this comment:
https://www.geosci-instrum-method-data-syst-discuss.net/gi-2018-13/gi-2018-13-

AC2-supplement.pdf

**Supplement:**

[revised manuscript text omitted]
 instead of using line element in LMA, which is a function of an independent variable as stated previously. This algorithm is a modified version of the M-LMA used in Lin's study (2017). The algorithm can be transformed to a backpropagation correction in BPNN and is expected to have smaller predicted errors, following Lin's study (2017). In three-dimensional space, represented by three Cartesian coordinates (Descartes, 1667), a function of $F$ is composed of two independent variables $x$ and $y$. For simulated surface problems, the function $F$ generates $Z_i = F(x_i, y_i), i = 1,2....$ . The initial codomain is i=1. $Z_t = F(x_t, y_t)$ is defined as a target output (Rumelhart and McClelland, 1986). When $x^o$ and $y^o$ best satisfy the surface function to minimise $\varepsilon^T \varepsilon$, the error is $\varepsilon = Z_i - Z_t$. $\delta xy$ and can be defined as an area element when simulating a surface. Therefore, a Taylor-series expansion in two variations ($\delta x$ and $\delta y$) approximates $F$ as follows;

$$F(x + \delta x, y + \delta y) \approx F(x, y) + J_a \delta xy \tag{1}$$

where $J_a$ is defined as the Jacobian matrix with $\dfrac{\partial F(x, y)}{\partial xy}$; a series of ($x_1, y_1$), ($x_2, y_2$), ($x_3, y_3$) is produced and converges toward a local minimiser $Z^o = F(x^o, y^o)$, called an optimised output, so that $\|Z_k - F(x + \delta x, y + \delta y)\| \approx \|\varepsilon - J_a \delta xy\|$ can be estimated. When $\varepsilon - J_a \delta xy$ is orthogonal to the column space of $J_a$ and $J_a^T(\varepsilon - J_a \delta xy)$, $I$ is defined as an identity matrix. A formula is proposed to import a parameter of $r$ as follows;

$$\delta xy(rI) + J_a^T J_a \delta xy = J_a^T \varepsilon \tag{2}$$

Finally, Formula (2) becomes;

$$(I_r + H_a)(\delta xy) = J_a^T \varepsilon \tag{3}$$

here $H_a = J_a^T J_a$ is defined as the approximated Hessian matrix and $rI = I_r$, where $r$ is the learning rate. In BPNN, if the learning rate is too high, the system will either oscillate about the true solution, or it will diverge completely. If the learning rate is too low, the system will take a long time to converge on the final solution. In this study, based on the concepts in this section, the parameters $x$ and $y$ can serve as the weight and bias that can be simultaneously optimised with PDP in a BPNN (Fukushima, 1980) because of the concept of an area element, for a specially designated area element $\delta xy$ used to simulate a optimised surface. Therefore when an $x$ value is altered by a $y$ value due to a specially designated relationship, real simultaneity can be achieved to simulate optimised area elements $\delta xy$ (Abbena, et al., 2006).

This means that when an area element $\delta xy$ is designated, then the $x$ , $y$ values related to $\delta xy$ are simultaneously designated. Finally, an optimised surface is obtained from these elements. When simulating a surface with BPNN and LMA, which is a function of a variable (Lin, 2017), it serves as a backpropagation correction to. A supposed initial $x$ value is offered and is updated by a weight. The bias updates the $y$ value. Both update processes are independent; without the concept of an area element, these processes are not simultaneous. In this situation, a simulated surface is obtained through simulating two nonlinear real-valued curves. One corresponds to the weight-updated $x$ value, another corresponds to $y$ value, which has been updated by bias. Therefore, LMA is not a PDP for simulating a surface.

**3 Results**

The results of many theoretical researches and engineering works regarding simulations have shown that using a two hidden layer network, with a small number of neurons in each layer, can replace a large number of neurons in a hidden layer network (Wagarachchi and Karunananda, 2014). The basic mathematical framework of an artificial neural network (ANN) was already introduced in section two of the study by Lin (2017). Using the previously mentioned network framework, microseismic data, from the four stations stated in section one, serves as training data to build the new BPNN models after testing with different learning rates between 0 and 1, with an increment of 0.01, from which the magnitude of upcoming microseismic data will be predicted. The M-LMA introduced in section two is used as the algorithm of backpropagation correction. For comparison, the LMA was simultaneously used as the algorithm of backpropagation correction. For both algorithm backpropagation, the initial weights and the initial biases were set to random variables (Nguyen and Widrow, 2009), and then feature scaling was performed (Bo et al, 2006; Xie et al, 2016). Using feature scaling, the variables were in the range of 0 and 1 because the value of the sigmoid function, called the activation function, which was used to train the

BPNN in this study, was in the range of 0 and 1. Randomly distributing the weights between 0 and 1 helped prevent biases toward any particular output. If initialisation was non-random, the network would consistently and prematurely connect to certain outputs that were undermining the training. The 1000 epochs were given for sample signals in the seismic records (Sinha, et al., 2010). The vertical component of an earthquake was the most dangerous because the earthquake forces mostly acted on the centre of gravity of the sliding soil mass, and the influences of vertical ground motions were on the seismic- induced displacements of the structures (Sawicki, et al. 2007; Zhao, et al. 2017). 
[revised manuscript text omitted]

---

## Referee Comment (RC5) · Anonymous Referee #1 · 8 Jun 2018

**64**

**Strong-Motion Instrumentation Programs in Taiwan**

T. C. Shin

*Central Weather Bureau, Taipei, Taiwan*

Y. B. Tsai

*National Central University, Chung-li, Taiwan*

Y. T. Yeh

*Kao-Yuan Institute of Technology, Kaohsiung, Taiwan*

C. C. Liu

*Institute of Earth Sciences, Academia Sinica, Taipei, Taiwan*

Y. M. Wu

*Central Weather Bureau, Taipei, Taiwan*

**1. Introduction**

Taiwan is located on the Circum-Pacific seismic belt. On the east side of Taiwan, the Philippine Sea plate subducts beneath the Eurasian plate at the Ryukyu trench, while at the south end of Taiwan, the South China Sea lithosphere subducts eastward under the Philippine Sea plate. The active convergent margin, connecting these two subduction zones, is characterized by rapid crustal deformation, regional-scale crustal faulting, and high seismicity. The densely populated western Taiwan, with high-rise buildings as a consequence of developing economy, is vulnerable to increasing earthquake hazard. Therefore, earthquake research has a high priority in Taiwan and considerable amounts of resources have been devoted to seismic instrumentation in general, and strong-motion instrumentation in particular.

Many disastrous earthquakes have occurred in the past, the most recent one being the 1999 Chi-Chi earthquake (Teng *et al.*, 2001). About 2500 people died and 300,000 were left homeless. The importance of strong-motion instrumentation has long been recognized, and we will summarize here the history of the strong-motion instrumentation programs in Taiwan. The earlier instrumentation programs were conducted primarily by the Institute of Earth Sciences, Academia Sinica, and have been mostly research oriented (Tsai, 1997; see also Report of the Institute of Earth Sciences under China (Taipei) in Chapter 79). The later efforts, involving an order-of-magnitude increase in the number of instruments, were conducted primarily by the Central Weather Bureau.

**2. Strong-Motion Instrumentation Program by IES**

**2.1 Strong-Motion Accelerographs Network (SMA)**

An islandwide strong-motion network was deployed by the Institute of Earth Sciences (IES), Academia Sinica, beginning in 1974, and by 1983, this network consisted of 72 stations. The instruments used were the standards at that time, i.e., the SMA-1s. The first strong-motion record obtained by this network was in April 1976. The purpose of this network is mainly to study earthquake source, structure responses, attenuation of ground motions, and risk analysis. By 1990, accelerographs of this network increased to 79, as a mix of analogy and digital recording units. Most of them were installed on free-field sites, while some were on the man-made structures. Most of those free-field stations were installed on the populous plain areas. After 1990, all stations of this network have been continuously upgraded to force-balance accelerometers with 16-bit resolution. Numbers of accelerograph stations on plain areas were reduced, and new stations were installed in the Central Range Mountain

*INTERNATIONAL HANDBOOK OF EARTHQUAKE AND ENGINEERING SEISMOLOGY, VOLUME 81B*     ISBN: 0-12-440652-1

of Taiwan. The total number of stations in this network is currently 74. The purpose of the new installation is to study the topographic effects and attenuation behavior of strong motion in the mountainous area (Huang, 2000). Leaders of this project include Y. B. Tsai (1974–1978), C. S. Wang (1979–1980), Y. T. Yeh (1981–1992), and B. S. Huang (1992–present).

**2.2 Strong-Motion Accelerograph Array in Taiwan, Phase 1 (SMART-1 Array)**

SMART-1 Array was set up in Lotung in 1980 and closed at the end of 1990. This was a cooperative project between the Institute of Earth Sciences, Academia Sinica and University of California, Berkeley. The SMART-1 Array consisted of a central site and accelerographs in three concentric circles, with radii of 200 m, 1 km, and 2 km, respectively. Each circle had 12 evenly spaced sensors. All 43 accelerographs were tied to a common time base, with timing to better than $\pm 0.01$ sec. Each accelerograph consisted of a triaxial force-balance accelerometer, capable of recording $\pm 2$ g, connected to a digital event recorder that uses a magnetic tape cassette for recording. The accelerographs were triggered on either vertical or horizontal acceleration at an adjustable preset threshold. Signals were digitized with a 1-bit resolution at 100 samples per second. Each recorder had a digital preevent memory that stored the output signals from the force-balance accelerometer for approximately 2.5 sec before trigger. Such accelerographs had an obvious advantage of providing synchronous time history of the ground-motion acceleration. Hence, we could perform spatial and temporal correlation across the whole array. The recorded data on digital cassettes were played back at the central laboratory and transferred onto a regular 9-track magnetic tape in ASCII format. During the playback, a seismologist scanned the digital signals displayed on a minicomputer console and made corrections for glitches, gaps, time code errors, and offsets in DC level. A regular magnetic tape containing edited data was available for further analysis only hours after the recording, whereas the analog recording/processing commonly used at that time would take days to digitize and process (Tsai and Bolt, 1983). Many research papers were published using the SMART-1 data (e.g., Loh *et al.*, 1982; Abrahamson, 1988).

**2.3 Lotung Large Scale Seismic Test Array (LSST)**

The LSST program was set up for evaluating the soil-structure interaction effects and the backfill effects. These effects are important in seismic design of nuclear reactor facilities. A quarter-scale and a 1/12-scale model of the nuclear reactor containment structure were constructed inside the SMART-1 Array on October 1985. It was closed at the end of 1990, the same time that SMART-1 Array had completed its mission. The LSST program was a joint project between the Taiwan Power Company (Taipower) and the Electric Power Research Institute of USA (EPRI), under the management of H. T. Tang. Under a contract

of Taipower, IES installed and maintained the instruments, as well as carried out data collection, reduction, and analysis. In the initial phase, four types of sensors were installed in the fields for data acquisition: the surface accelerometer, the downhole accelerometer, the structural response accelerometer, and the interfacial pressure transducer. These sensors were triaxial type, except the pressure transducer. The output of all accelerometers and pressure gauges was transmitted by hard wire to the central recording unit and was recorded on cassette tapes. These tapes were then processed and transcribed onto 9-track tapes using the ASCII format.

**2.4 SMART-2 Array**

The SMART-2 strong-motion array was deployed by IES in the northern part of the Longitudinal Valley in Hualien in December 1990, and was fully operational in 1992 (Chiu *et al.*, 1994). It consists of 45 Kinemetrics SSR-1 instruments as surface stations and two sets of downhole subarrays. All sensors used in this array are force-balance accelerometers. This array is designed to study the rupture process of the seismic fault and the characteristics of near-source ground motions. Furthermore, the high-quality data from SMART-2 may be used for research in seismology and earthquake engineering (e.g., Huang and Chiu, 1996).

Chiu *et al*. (1995) studied the coherency of ground motions based on the SMART-2 data and compared it with the results of the SMART-1. A comparison of coherency functions for both vertical and horizontal motions from a magnitude 5.5 earthquake recorded by the SMART-2 indicated no significant difference in the range of 1 to 10 Hz for separation distance of 400, 800, and 1500 m.

**2.5 Hualien Large Scale Seismic Test Array (HLSST)**

Since 1993, EPRI and Taipower have sponsored a dense multiple-element array, the HLSST network, located at the Veteran's Marble Plant of Hualien within the SMART-2 deployment area of northeastern Taiwan. This is an international joint project operated by IES with the objectives of investigating the behaviors of soil-structure interaction during severe earthquakes and verifying the validity of various analysis methods using the strong-motion records. To serve this purpose, a one-quarter-scale cylindrical reactor model and a cylindrical liquid-storage-tank model were constructed in Hualien, a high-seismicity region. The cylindrical liquid-storage-tank model was closed in July 1998.

**2.6 Downhole Accelerometer Arrays in the Taipei Basin (DART)**

A research project, "Integrated Survey of Subsurface Geology and Engineering Environment of the Taipei Basin," was proposed in early 1990s to collect data for the purposes of engineering construction, groundwater management, ground subsidence

prediction, study of the basin effects of seismic waves, and geological sciences. This project has been sponsored by the Central Geological Survey (CGS), Ministry of Economic Affairs since August 1991. CGS contracted the study of the basin effects on seismic waves to IES, which proposed the DART program, in which one site was installed per year to analyze the variation of seismic waves propagating from the basement to ground surface. Each site includes one free-surface accelerometer and some downhole sensors. These force-balance accelerometers are connected to a PC-based central recording system or a K2 digital recording system with GPS timing and position information.

Wen *et al*. (1995) studied basin effects using a dense strong-motion array in Taipei Basin. Their results showed that site amplification is frequency dependent. They also indicated that both horizontal peak ground acceleration and the spectral ratio in low-frequency band are closely correlated with the geological structure of Taipei Basin.

**3. Strong-Motion Instrumentation Program by CWB**

In the late 1980s, Y. B. Tsai proposed an extensive strong-motion instrumentation program for the urban areas in Taiwan. Since the Central Weather Bureau has the official responsibility to monitor earthquakes in the Taiwan region, the Taiwan Strong-Motion Instrument Program (TSMIP; see Shin, 1993) was successfully implemented during 1991–1996.

The main goal of this program is to collect high-quality instrumental recordings of strong ground shaking from earthquakes, both at free-field sites and in buildings and bridges. These data are crucial for improving earthquake-resistant design of buildings and bridges and for understanding the earthquake source mechanisms, as well as seismic wave propagation from the source to the site of interest, including local site effects.

Two types of digital strong-motion instruments were deployed throughout Taiwan in this program, with special emphasis in nine metropolitan areas. One type is a digital triaxial accelerograph for recording free-field ground shakings (Liu *et al*., 1999). The other type is a multichannel (32 or 64 channels), central-recording, accelerograph array system for monitoring shakings caused by earthquakes in buildings and other structures (Lee and Shin, 1997). By the end of 2000, a total of 640 free-field accelerographs and 56 structural arrays had been deployed. Locations of the free-field accelerographs are shown in Figure 1, and locations for the building arrays are shown in Figure 2.

**4. The Taiwan Rapid Earthquake Information Release System**

The desire for seismological observation in real time has long been recognized, and significant advances have been made during the past decade in many countries (Kanamori *et al*., 1997).

[Figure]

**FIGURE 1** Locations of the CWB free-field, three-component, digital accelerograph stations. The star indicates the location of the Chi-Chi earthquake. Surface ruptures extending about 80 km north–south are shown to the left of the epicenter.

The idea of an islandwide early earthquake warning system using the existing telemetry in Taiwan was first proposed by T. L. Teng in the early 1990s. The Taiwan Rapid Earthquake Information Release System (RTD) is based on a simple hardware/software design first introduced by Lee *et al*. (1989), and was subsequently improved and refined (Lee, 1994; Lee *et al*., 1996; Shin *et al*., 1996; Teng *et al*., 1997; Wu *et al*., 1997, 1998, 1999). The RTD system consists of 61 telemetered strong-motion accelerographs in Taiwan (Fig. 3). Digital signals are continuously telemetered to the headquarters of the Central Weather Bureau (CWB) in Taipei via 4800-baud leased telephone lines. Each telemetered signal contains three-component seismic data digitized at 50 samples per second and at 16-bit resolution. The full recording range is $\pm 2$ g. The incoming digital data streams are processed by a computer program called XRTPDB (Tottingham and Mayle, 1994). Whenever the prespecified trigger criteria are met, the digital waveforms are stored in memory and are automatically analyzed by a series of programs (Wu *et al*., 1998).

[Figure]

**FIGURE 2** Locations of the CWB structural strong-motion arrays in buildings and bridges. The star indicates the location of the Chi-Chi earthquake. Surface ruptures extending about 80 km north–south are shown to the left of the epicenter.

[Figure]

**FIGURE 3** Map showing the telemetered stations of the Taiwan Earthquake Rapid Information Release System (RTD).

The results are immediately disseminated to emergency response agencies electronically in four ways, namely, by e-mail, World Wide Web, fax, and a pager system (Fig. 4).

**5. The Chi-Chi Earthquake of September 21, 1999**

The Chi-Chi earthquake occurred at 1:47 on September 21, 1999 (Taiwan local time) or at 17:47 on September 20, 1999 UTC. It was the largest ($M_\mathrm{w} = 7.6$) earthquake to have occurred on land in Taiwan in the 20th century. For the main shock, 441 digital three-component, strong-motion records were successfully retrieved by the Taiwan Central Weather Bureau (CWB) from about 640 accelerographs deployed at the free-field sites. These preliminary strong-motion data from the Chi-Chi main shock were released on December 13, 1999, in the form of a prepublication data CD (Lee *et al.*, 1999). During the first 6 hours after the main shock, about 10,000 strong-motion records were recovered, and since then another 20,000 records were obtained. This is by far the best-recorded major earthquake in the world. There are over 60 three-component strong-motion records within 20 km of the fault ruptures.

A preliminary report of this earthquake is given in Shin *et al.* (2000), and a detailed report of the processed free-field acceleration data is given in Lee *et al.* (2001a). These data are also archived in Lee *et al.* (2001b). Characteristics of the strong ground motion are given by Tsai and Huang (2000). At the time of the earthquake, the Taiwan Rapid Earthquake Information Release System (RTD) automatically determined the location and magnitude for the main shock and prepared a shake map within 102 seconds after the earthquake's origin time. This information was then sent out by a pager-telephone system, by an e-mail server, and by fax. Its performance during the Chi-Chi earthquake and numerous strong aftershocks has been documented in Wu *et al.* (2000).

[Figure]

**FIGURE 4** A block diagram showing the hardware of the RTD system.

**6. Concluding Remarks**

The Chi-Chi earthquake clearly demonstrated the usefulness of the extensive strong-motion instrumentation in Taiwan for purposes of emergency response and research in seismology and earthquake engineering. Readers are referred to several special issues and proceedings on the Chi-Chi earthquake (e.g., BERI, 2000; Loh and Liao, 2000; Wang *et al.*, 2000; Teng *et al.*, 2001).

**Acknowledgments**

The strong-motion instrumentation programs in Taiwan would not be possible without the labor of many people. We wish to thank many advisors and collaborators: B. A. Bolt, K. C. Chen, H. C. Chiu, B. S. Huang, G. C. Lee, W. H. K. Lee, K. S. Liu, S. C. Liu, C. H. Loh, S. T. Mau, G. B. Ou, M. S. Sheu, H. T. Tang, T. L. Teng, C. Y. Wang, K. L. Wen, C. F. Wu, F. T. Wu, Y. H. Yeh, and G. K. Yu.

**References**

Abrahamson, N. A. (1988). Statistical properties of peak ground acceleration recorded by the SMART 1 array. *Bull. Seism. Soc. Am.* **78**, 26–41.

BERI (2000). "Special Issue: The 1999 Chi-Chi, Taiwan earthquake." *Bull. Earthquake Res. Inst. Tokyo Univ.* **75**, Paret 1.

Chiu, H. C., Y. T. Yeh, S. D. Ni, L. Lee, W. S. Liu, C. F. Wen, and C. C. Liu (1994). A new strong-motion array in Taiwan—SMART-2. *Terres. Atmos. Ocean. Sci.* **5**, 443–455.

Chiu, H. C., R. V. Amirbekian, and B. A. Bolt (1995). Transferability of strong ground motion coherency between the SMART1 and SMART2 arrays. *Bull. Seism. Soc. Am.* **85**, 342–348.

Huang, B. S. (2000). Reconstruction of 2-D ground motions. *Geophys. Res. Lett.* **27**, 3025–3028.

Huang, H. C., and H. C. Chiu (1996). Estimation of site amplification from Dahan Downhole recording. *Int. J. Earthquake Eng. Struct. Dyn.* **25**, 319–332.

Kanamori, H., E. Hauksson, and T. Heaton (1997). Real-time seismology and earthquake hazard mitigation. *Nature* **390**, 461–464.

Lee, W. H. K. (Editor) (1994). "Realtime Seismic Data Acquisition and Processing," IASPEI Software Library, Volume 1 (2nd Edition), Seismological Society of America, El Cerrito, CA, (285 pp. and 3 diskettes).

Lee, W. H. K., D. M. Tottingham, and J. O. Ellis (1989). Design and implementation of a PC-based seismic data acquisition, processing, and analysis system. *IASPEI Software Library* **1**, 21–46.

Lee, W. H. K., T. C. Shin, and T. L. Teng (1996). Design and implementation of earthquake early warning systems in Taiwan. *Proc. 11th World Conf. Earthq. Eng., Paper No. 2133.*

Lee, W. H. K., T. C. Shin, K. W. Kuo, and K. C. Chen (1999). CWB Free-Field Strong-Motion Data from the 921 Chi-Chi Earthquake: Volume 1. Digital Acceleration Data, Pre-Publication CD, Central Weather Bureau, Taipei, Taiwan.

Lee, W. H. K., T. C. Shin, K. W. Kuo, K. C. Chen, and C. F. Wu (2001a). CWB Free-Field Strong-Motion Data from the 921 Chi-Chi Earthquake: Processed Acceleration Files on CD-ROM, *Strong-Motion Data Series CD-001*, Central Weather Bureau, Taipei, Taiwan.

Lee, W. H. K., T. C. Shin, K. W. Kuo, K. C. Chen, and C. F. Wu (2001b). Data Files from "CWB Free-Field Strong-Motion Data from the 21 September Chi-Chi, Taiwan, Earthquake," *Bull. Seism. Soc. Am.* **91**, 1390 and CD-ROM.

Liu, K. S., T. C. Shin, and Y. B. Tsai (1999). A free-field strong motion network in Taiwan: TSMIP. *Terres. Atmos. Ocean. Sci.* **10**, 377–396.

Loh, C. H., J. Penzien, and Y. B. Tsai (1982). Engineering analysis of SMART 1 array accelerograms. *Int. J. Earthquake Eng. Struct. Dyn.* **10**, 575–591.

Loh, C. H., and W. I. Liao (Editors) (2000). "Proceedings of International Workshop on Annual Commemoration of Chi-Chi Earthquake," 4 volumes, National Center for Research on Earthquake Engineering, Taipei, Taiwan.

Shin, T. C. (1993). Progress summary of the Taiwan strong-motion instrumentation program. In: "Symposium on the Taiwan Strong-Motion Program," Central Weather Bureau, 1–10.

Shin, T. C. (2000). Some seismological aspects of the 1999 Chi-Chi earthquakes in Taiwan. *Terres. Atmos. Ocean. Sci.* **11**, 555–566.

Shin, T. C., Y. B. Tsai, and Y. M. Wu (1996). Rapid response of large earthquake in Taiwan using a realtime telemetered network of digital accelerographs. *Proc. 11th World Conf. Earthq. Eng., Paper No. 2137.*

Shin, T. C., K. W. Kuo, W. H. K. Lee, T. L. Teng, and Y. B. Tsai (2000). A preliminary report on the 1999 Chi-Chi (Taiwan) earthquake. *Seism. Res. Lett.* **71**, 24–30.

Teng, T. L., Y. M. Wu, T. C. Shin, Y. B. Tsai, and W. H. K. Lee (1997). One minute after: strong motion map, effective epicenter, and effective magnitude. *Bull. Seism. Soc. Am.* **87**, 1209–1219.

Teng, T. L., Y. B. Tsai, and W. H. K. Lee (Editors) (2001). Dedicated Issue on the Chi-Chi (Taiwan) Earthquake of September 20, 1999. *Bull. Seism. Soc. Am.* **91**, 893–1395.

Tottingham, D. M., and A. J. Mayle (1994). User manual for XRTPDB. *IASPEI Software Library* **1** (2nd Edition), 255–263.

Tsai, Y. B. (1997). A brief review of strong motion instrumentation in Taiwan. In: "Vision 2005: An Action Plan for Strong Motion Programs to Mitigate Earthquake Losses in Urbanized Areas," A Workshop held in Monterey, California on April 2–4, 1997.

Tsai, Y. B., and B. A. Bolt (1983). An analysis of horizontal peak ground acceleration and velocity from SMART 1 array date. *Bull. Inst. Earth Sci., Acad. Sinica* **3**, 105–126.

Tsai, Y. B., and M. W. Huang (2000). Strong ground motion characteristics of the Chi-Chi, Taiwan earthquake of September 21, 1999. *Earthquake Eng. & Eng. Seism.* **2**, 1–21.

Wang, C. S., S. K. Hsu, H. Kao, and C. Y. Wang (Editors) (2000). Special Issue on the 1999 Chi-Chi Earthquake in Taiwan. *Terres. Atmos. Ocean. Sci.* **11**, 555–752.

Wen, K. L., H. Y. Peng, L. F. Liu, and T. C. Shin (1995). Basin effects analysis from a dense strong motion observation network. *Int. J. Earthquake Eng. Struct. Dyn.* **24**, 1069–1083

Wu, Y. M., T. C. Shin, C. C. Chen, Y. B. Tsai, W. H. K. Lee, and T. L. Teng (1997). Taiwan rapid earthquake information release system. *Seism. Res. Lett.* **68**, 931–943.

Wu, Y. M., C. C. Chen, J. K. Chung, and T. C. Shin (1998). An automatic phase picker of the real-time acceleration seismic network. *Meteorol. Bull., Central Weather Bureau, Taipei* **42**, 103–117 (in Chinese).

Wu, Y. M., J. K. Chung, T. C. Shin, N. C. Hsia, Y. B. Tsai, W. H. K. Lee, and T. L. Teng (1999). Development of an integrated seismic early warning system in Taiwan—case for Hualien area earthquakes. *Terres. Atmos. Ocean. Sci.* **10**, 719–736.

Wu, Y. M., W. H. K. Lee, C. C. Chen, T. C. Shin, T. L. Teng, and Y. B. Tsai (2000). Performance of the Taiwan Rapid Earthquake Information Release System (RTD) during the 1999 Chi-Chi (Taiwan) earthquake. *Seism. Res. Lett.* **71**, 338–343.

---

## Referee Comment (RC6) · Anonymous Referee #1 · 9 Jun 2018

It can be accepted to your paper for me, could you let me know your comments about AI. Do you know, a famous Physicist " Stephen Hawking" worried about the future of AI ? You do not need to revise your paper. A document about AI is by attached file which from internet. Please check it, maybe it is positive for you.

Please also note the supplement to this comment:
https://www.geosci-instrum-method-data-syst-discuss.net/gi-2018-13/gi-2018-13-RC6-supplement.pdf

[Figure]

**Supplement:**

**Artificial intelligence**

**Artificial intelligence** (**AI**, also **machine intelligence**, **MI**) is intelligence demonstrated by machines, in contrast to the **natural intelligence** (**NI**) displayed by humans and other animals. In computer science AI research is defined as the study of "intelligent agents": any device that perceives its environment and takes actions that maximize its chance of successfully achieving its goals.[1] Colloquially, the term "artificial intelligence" is applied when a machine mimics "cognitive" functions that humans associate with other human minds, such as "learning" and "problem solving".[2]

The scope of AI is disputed: as machines become increasingly capable, tasks considered as requiring "intelligence" are often removed from the definition, a phenomenon known as the AI effect, leading to the quip, "AI is whatever hasn't been done yet."[3] For instance, optical character recognition is frequently excluded from "artificial intelligence", having become a routine technology.[4] Capabilities generally classified as AI as of 2017 include successfully understanding human speech,[5] competing at the highest level in strategic game systems (such as chess and Go[6]), autonomous cars, intelligent routing in content delivery network and military simulations.

Artificial intelligence was founded as an academic discipline in 1956, and in the years since has experienced several waves of optimism,[7][8] followed by disappointment and the loss of funding (known as an "AI winter"),[9][10] followed by new approaches, success and renewed funding.[8][11] For most of its history, AI research has been divided into subfields that often fail to communicate with each other.[12] These sub-fields are based on technical considerations, such as particular goals (e.g. "robotics" or "machine learning"),[13] the use of particular tools ("logic" or "neural networks"), or deep philosophical differences.[14][15][16] Subfields have also been based on social factors (particular institutions or the work of particular researchers).[12]

The traditional problems (or goals) of AI research include reasoning, knowledge representation, planning, learning, natural language processing, perception and the ability to move and manipulate objects.[13] General intelligence is among the field's long-term goals.[17] Approaches include statistical methods, computational intelligence, and traditional symbolic AI. Many tools are used in AI, including versions of search and mathematical optimization, neural networks and methods based on statistics, probability and economics. The AI field draws upon computer science, mathematics, psychology, linguistics, philosophy and many others.

The field was founded on the claim that human intelligence "can be so precisely described that a machine can be made to simulate it".[18] This raises philosophical arguments about the nature of the mind and the ethics of creating artificial beings endowed with human-like intelligence, issues which have been explored by myth, fiction and philosophy since antiquity.[19] Some people also consider AI to be a danger to humanity if it progresses unabatedly.[20] Others believe that AI, unlike previous technological revolutions, will create a risk of mass unemployment.[21]

| Artificial intelligence |
| --- |
| **Major goals** |
| Knowledge reasoning |
| Planning |
| Machine learning |
| Natural language processing |
| Computer vision |
| Robotics |
| Artificial general intelligence |
| **Approaches** |
| Symbolic |
| Deep learning |
| Bayesian networks |
| Evolutionary algorithms |
| **Philosophy** |
| Ethics |
| Existential risk |
| Turing test |
| Chinese room |
| Friendly AI |
| **History** |
| Timeline |
| Progress |
| AI winter |
| **Technology** |
| Applications |
| Projects |
| Programming languages |
| **Glossary** |
| Glossary |

In the twenty-first century, AI techniques have experienced a resurgence following concurrent advances in computer power, large amounts of data, and theoretical understanding; and AI techniques have become an essential part of the technology industry, helping to solve many challenging problems in computer science.[22][11]

**Contents**

**History**

Thought-capable artificial beings appeared as storytelling devices in antiquity,[23] and have been common in fiction, as in Mary Shelley's *Frankenstein* or Karel Čapek's *R.U.R. (Rossum's Universal Robots).*[24] These characters and their fates raised many of the same issues now discussed in the ethics of artificial intelligence.[19]

The study of mechanical or "formal" reasoning began with philosophers and mathematicians in antiquity. The study of mathematical logic led directly to Alan Turing's theory of computation, which suggested that a machine, by shuffling symbols as simple as "0" and "1", could simulate any conceivable act of mathematical deduction. This insight, that digital computers can simulate any process of formal reasoning, is known as the Church–Turing thesis.[25] Along with concurrent discoveries in neurobiology, information theory and cybernetics, this led researchers to consider the possibility of building an electronic brain. Turing proposed that "if a human could not distinguish between responses from a machine and a human, the machine could be considered "intelligent".[26] The first work that is now generally recognized as AI was McCullouch and Pitts' 1943 formal design for Turing-complete "artificial neurons".[27]

[Figure]

Talos, an ancient mythical automaton with artificial intelligence

The field of AI research was born at a workshop at Dartmouth College in 1956.[28] Attendees Allen Newell (CMU), Herbert Simon (CMU), John McCarthy (MIT), Marvin Minsky (MIT) and Arthur Samuel (IBM) became the founders and leaders of AI research.[29] They and their students produced programs that the press described as "astonishing":[30] computers were learning checkers strategies (c. 1954)[31] (and by 1959 were reportedly playing better than the average human),[32] solving word problems in algebra, proving logical theorems (Logic Theorist, first run c. 1956) and speaking English.[33] By the middle of the 1960s, research in the U.S. was heavily funded by the Department of Defense[34] and laboratories had been established around the world.[35] AI's founders were optimistic about the future: Herbert Simon predicted, "machines will be capable, within twenty years, of doing any work a man can do". Marvin Minsky agreed, writing, "within a generation ... the problem of creating 'artificial intelligence' will substantially be solved".[7]

They failed to recognize the difficulty of some of the remaining tasks. Progress slowed and in 1974, in response to the criticism of Sir James Lighthill[36] and ongoing pressure from the US Congress to fund more productive projects, both the U.S. and British governments cut off exploratory research in AI. The next few years would later be called an "AI winter",[9] a period when obtaining funding for AI projects was difficult.

In the early 1980s, AI research was revived by the commercial success of expert systems,[37] a form of AI program that simulated the knowledge and analytical skills of human experts. By 1985 the market for AI had reached over a billion dollars. At the same time, Japan's fifth generation computer project inspired the U.S and British governments to restore funding for academic research.[8] However, beginning with the collapse of the Lisp Machine market in 1987, AI once again fell into disrepute, and a second, longer-lasting hiatus began.[10]

In the late 1990s and early 21st century, AI began to be used for logistics, data mining, medical diagnosis and other areas.[22] The success was due to increasing computational power (see Moore's law), greater emphasis on solving specific problems, new ties between AI and other fields (such as statistics, economics and mathematics), and a commitment by researchers to mathematical

methods and scientific standards.[38] Deep Blue became the first computer chess-playing system to beat a reigning world chess champion, Garry Kasparov on 11 May 1997.[39]

In 2011, a *Jeopardy!* quiz show exhibition match, IBM's question answering system, Watson, defeated the two greatest Jeopardy champions, Brad Rutter and Ken Jennings, by a significant margin.[40] Faster computers, algorithmic improvements, and access to large amounts of data enabled advances in machine learning and perception; data-hungry deep learning methods started to dominate accuracy benchmarks around 2012.[41] The Kinect, which provides a 3D body–motion interface for the Xbox 360 and the Xbox One use algorithms that emerged from lengthy AI research[42] as do intelligent personal assistants in smartphones.[43] In March 2016, AlphaGo won 4 out of 5 games of Go in a match with Go champion Lee Sedol, becoming the first computer Go-playing system to beat a professional Go player without handicaps.[6][44] In the 2017 Future of Go Summit, AlphaGo won a three-game match with Ke Jie,[45] who at the time continuously held the world No. 1 ranking for two years.[46][47] This marked the completion of a significant milestone in the development of Artificial Intelligence as Go is an extremely complex game, more so than Chess.

According to Bloomberg's Jack Clark, 2015 was a landmark year for artificial intelligence, with the number of software projects that use AI within Google increased from a "sporadic usage" in 2012 to more than 2,700 projects. Clark also presents factual data indicating that error rates in image processing tasks have fallen significantly since 2011.[48] He attributes this to an increase in affordable neural networks, due to a rise in cloud computing infrastructure and to an increase in research tools and datasets.[11] Other cited examples include Microsoft's development of a Skype system that can automatically translate from one language to another and Facebook's system that can describe images to blind people.[48]

**Basics**

A typical AI perceives its environment and takes actions that maximize its chance of successfully achieving its goals.[1] An AI's intended goal function can be simple ("1 if the AI wins a game of Go, 0 otherwise") or complex ("Do actions mathematically similar to the actions that got you rewards in the past"). Goals can be explicitly defined, or can be induced. If the AI is programmed for "reinforcement learning", goals can be implicitly induced by rewarding some types of behavior and punishing others.[a] Alternatively, an evolutionary system can induce goals by using a "fitness function" to mutate and preferentially replicate high-scoring AI systems; this is similar to how animals evolved to innately desire certain goals such as finding food, or how dogs can be bred via artificial selection to possess desired traits.[49] Some AI systems, such as nearest-neighbor, instead reason by analogy; these systems are not generally given goals, except to the degree that goals are somehow implicit in their training data.[50] Such systems can still be benchmarked if the non-goal system is framed as a system whose "goal" is to successfully accomplish its narrow classification task.[51]

AI often revolves around the use of algorithms. An algorithm is a set of unambiguous instructions that a mechanical computer can execute.[b] A complex algorithm is often built on top of other, simpler, algorithms. A simple example of an algorithm is the following recipe for optimal play at tic-tac-toe:[52]

1. If someone has a "threat" (that is, two in a row), take the remaining square. Otherwise,
2. if a move "forks" to create two threats at once, play that move. Otherwise,
3. take the center square if it is free. Otherwise,
4. if your opponent has played in a corner, take the opposite corner. Otherwise,
5. take an empty corner if one exists. Otherwise,
6. take any empty square.

Many AI algorithms are capable of learning from data; they can enhance themselves by learning new heuristics (strategies, or "rules of thumb", that have worked well in the past), or can themselves write other algorithms. Some of the "learners" described below, including Bayesian networks, decision trees, and nearest-neighbor, could theoretically, if given infinite data, time, and memory, learn to approximate any function, including whatever combination of mathematical functions would best describe the entire world. These learners could therefore, in theory, derive all possible knowledge, by considering every possible hypothesis and matching it against the data. In practice, it is almost never possible to consider every possibility, because of the phenomenon of "combinatorial explosion", where the amount of time needed to solve a problem grows exponentially. Much of AI research involves figuring out how to identify and avoid considering broad swaths of possibililities that are unlikely to be fruitful.[53][54] For example, when viewing a

map and looking for the shortest driving route from Denver to New York in the East, one can in most cases skip looking at any path through San Francisco or other areas far to the West; thus, an AI wielding an pathfinding algorithm like A* can avoid the combinatorial explosion that would ensue if every possible route had to be ponderously considered in turn.[55]

The earliest (and easiest to understand) approach to AI was symbolism (such as formal logic): "If an otherwise healthy adult has a fever, then they may have influenza". A second, more general, approach is Bayesian inference: "If the current patient has a fever, adjust the probability they have influenza in such-and-such way". The third major approach, extremely popular in routine business AI applications, is analogizers such as SVM and nearest-neighbor: "After examining the records of known past patients whose temperature, symptoms, age, and other factors mostly match the current patient, X% of those patients turned out to have influenza". A fourth approach is harder to intuitively understand, but is inspired by how the brain's machinery works: the neural network approach uses artificial 'neurons' that can learn by comparing itself to the desired output and altering the strengths of the connections between its internal neurons to "reinforce" connections that seemed to be useful. These four main approaches can overlap with each other and with evolutionary systems; for example, neural nets can learn to make inferences, to generalize, and to make analogies. Some systems implicitly or explicitly use multiple of these approaches, alongside many other AI and non-AI algorithms; the best approach is often different depending on the problem.[56][57]

Learning algorithms work on the basis that strategies, algorithms, and inferences that worked well in the past are likely to continue working well in the future. These inferences can be obvious, such as "since the sun rose every morning for the last 10,000 days, it will probably rise tomorrow morning as well". They can be nuanced, such as "X% of families have geographically separate species with color variants, so there is an Y% chance that undiscovered black swans exist". Learners also work on the basis of "Occam's razor": The simplest theory that explains the data is the likeliest. Therefore, to be successful, a learner must be designed such that it prefers simpler theories to complex theories, except in cases where the complex theory is proven substantially better. Settling on a bad, overly complex theory gerrymandered to fit all the past training data is known as overfitting. Many systems attempt to reduce overfitting by rewarding a theory in accordance with how well it fits the data,

[Figure]

The blue line could be an example of overfitting a linear function due to random noise.

but penalizing the theory in accordance with how complex the theory is.[58] Besides classic overfitting, learners can also disappoint by "learning the wrong lesson". A toy example is that an image classifier trained only on pictures of brown horses and black cats might conclude that all brown patches are likely to be horses.[59] A real-world example is that, unlike humans, current image classifiers don't determine the spatial relationship between components of the picture; instead, they learn abstract patterns of pixels that humans are oblivious to, but that linearly correlate with images of certain types of real objects. Faintly superimposing such a pattern on a legitimate image results in an "adversarial" image that the system misclassifies.[c][60][61][62]

Compared with humans, existing AI lacks several features of human "commonsense reasoning"; most notably, humans have powerful mechanisms for reasoning about "naïve physics" such as space, time, and physical interactions. This enables even young children to easily make inferences like "If I roll this pen off a table, it will fall on the floor". Humans also have a powerful mechanism of "folk psychology" that helps them to interpret natural-language sentences such as "The city councilmen refused the demonstrators a permit because they advocated violence". (A generic AI has difficulty inferring whether the councilmen or the demonstrators are the ones alleged to be advocating violence.)[65][66][67] This lack of "common knowledge" means that AI often makes different mistakes than humans make, in ways that can seem incomprehensible. For example, existing self-driving cars cannot reason about the location nor the intentions of pedestrians in the exact way that humans do, and instead must use non-human modes of reasoning to avoid accidents.[68][69][70]

**Problems**

The overall research goal of artificial intelligence is to create technology that allows computers and machines to function in an intelligent manner. The general problem of simulating (or creating) intelligence has been broken down into sub-problems. These consist of particular traits or capabilities that researchers expect an intelligent system to display. The traits described below have received the most attention.[13]

**Reasoning, problem solving**

Early researchers developed algorithms that imitated step-by-step reasoning that humans use when they solve puzzles or make logical deductions.[71] By the late 1980s and 1990s, AI research had developed methods for dealing with uncertain or incomplete information, employing concepts from probability and economics.[72]

These algorithms proved to be insufficient for solving large reasoning problems, because they experienced a "combinatorial explosion": they became exponentially slower as the problems grew larger.[53] In fact, even humans rarely use the step-by-step deduction that early AI research was able to model. They solve most of their problems using fast, intuitive judgements.[73]

[Figure]

A self-driving car system may use a neural network to determine which parts of the picture seem to match previous training images of pedestrians, and then model those areas as slow-moving but somewhat unpredictable rectangular prisms that must be avoided.[63][64]

**Knowledge representation**

Knowledge representation[74] and knowledge engineering[75] are central to classical AI research. Some "expert systems" attempt to gather together explicit knowledge possessed by experts in some narrow domain. In addition, some projects attempt to gather the "commonsense knowledge" known to the average person into a database containing extensive knowledge about the world. Among the things a comprehensive commonsense knowledge base would contain are: objects, properties, categories and relations between objects;[76] situations, events, states and time;[77] causes and effects;[78] knowledge about knowledge (what we know about what other people know);[79] and many other, less well researched domains. A representation of "what exists" is an ontology: the set of objects, relations, concepts, and properties formally described so that software agents can interpret them. The semantics of these are captured as description logic concepts, roles, and individuals, and typically implemented as classes, properties, and individuals in the Web Ontology Language.[80] The most general ontologies are called upper ontologies, which attempt to provide a foundation for all other knowledge[81] by acting as mediators between domain ontologies that cover specific knowledge about a particular knowledge domain (field of interest or area of concern). Such formal knowledge representations can be used in content-based indexing and retrieval,[82] scene interpretation,[83] clinical decision support,[84] knowledge discovery (mining "interesting" and actionable inferences from large databases),[85] and other areas.[86]

[Figure]

An ontology represents knowledge as a set of concepts within a domain and the relationships between those concepts.

Among the most difficult problems in knowledge representation are:

**Default reasoning and the qualification problem**
Many of the things people know take the form of "working assumptions". For example, if a bird comes up in conversation, people typically picture an animal that is fist sized, sings, and flies. None of these things are true about all birds. John McCarthy identified this problem in 1969[87] as the qualification problem: for any commonsense rule that AI researchers care to represent, there tend to be a huge number of exceptions. Almost nothing is simply true or false in the way that abstract logic requires. AI research has explored a number of solutions to this problem.[88]

**The breadth of commonsense knowledge**
The number of atomic facts that the average person knows is very large. Research projects that attempt to build a complete knowledge base of commonsense knowledge (e.g., Cyc) require enormous amounts of laborious ontological engineering—they must be built, by hand, one complicated concept at a time.[89]

**The subsymbolic form of some commonsense knowledge**

Much of what people know is not represented as "facts" or "statements" that they could express verbally. For example, a chess master will avoid a particular chess position because it "feels too exposed"[90] or an art critic can take one look at a statue and realize that it is a fake.[91] These are non-conscious and sub-symbolic intuitions or tendencies in the human brain.[92] Knowledge like this informs, supports and provides a context for symbolic, conscious knowledge. As with the related problem of sub-symbolic reasoning, it is hoped that situated AI, computational intelligence, or statistical AI will provide ways to represent this kind of knowledge.[92]

**Planning**

Intelligent agents must be able to set goals and achieve them.[93] They need a way to visualize the future—a representation of the state of the world and be able to make predictions about how their actions will change it—and be able to make choices that maximize the utility (or "value") of available choices.[94]

[Figure]

A hierarchical control system is a form of control system in which a set of devices and governing software is arranged in a hierarchy

In classical planning problems, the agent can assume that it is the only system acting in the world, allowing the agent to be certain of the consequences of its actions.[95] However, if the agent is not the only actor, then it requires that the agent can reason under uncertainty. This calls for an agent that can not only assess its environment and make predictions, but also evaluate its predictions and adapt based on its assessment.[96]

Multi-agent planning uses the cooperation and competition of many agents to achieve a given goal. Emergent behavior such as this is used by evolutionary algorithms and swarm intelligence.[97]

**Learning**

Machine learning, a fundamental concept of AI research since the field's inception,[98] is the study of computer algorithms that improve automatically through experience.[99][100]

Unsupervised learning is the ability to find patterns in a stream of input. Supervised learning includes both classification and numerical regression. Classification is used to determine what category something belongs in, after seeing a number of examples of things from several categories. Regression is the attempt to produce a function that describes the relationship between inputs and outputs and predicts how the outputs should change as the inputs change. In reinforcement learning[101] the agent is rewarded for good responses and punished for bad ones. The agent uses this sequence of rewards and punishments to form a strategy for operating in its problem space.

**Natural language processing**

Natural language processing[102] gives machines the ability to read and understand human language. A sufficiently powerful natural language processing system would enable natural language user interfaces and the acquisition of knowledge directly from human-written sources, such as newswire texts. Some straightforward applications of natural language processing include information retrieval, text mining, question answering[103] and machine translation.[104] Many current approaches use word co-occurrence frequencies to construct syntactic representations of text. "Keyword spotting" strategies for search are popular and scalable but dumb; a search query for "dog" might only match documents with the literal word "dog" and miss a document with the word "poodle". "Lexical affinity" strategies use the occurrence of words such as "accident" to assess the sentiment of a document. Modern statistical NLP approaches can combine all these strategies as well as others, and often achieve acceptable accuracy at the page or paragraph level, but continue to lack the semantic understanding required to classify isolated sentences well. Besides the usual difficulties with

encoding semantic commonsense knowledge, existing semantic NLP sometimes scales too poorly to be viable in business applications. Beyond semantic NLP, the ultimate goal of "narrative" NLP is to embody a full understanding of commonsense reasoning.[105]

[Figure]

A parse tree represents the syntactic structure of a sentence according to some formal grammar.

**Perception**

Machine perception[106] is the ability to use input from sensors (such as cameras, microphones, tactile sensors, sonar and others) to deduce aspects of the world. Computer vision[107] is the ability to analyze visual input. A few selected subproblems are speech recognition,[108] facial recognition and object recognition.[109]

**Motion and manipulation**

The field of robotics[110] is closely related to AI. Intelligence is required for robots to handle tasks such as object manipulation[111] and navigation, with sub-problems such as localization, mapping, and motion planning. These systems require that an agent is able to: Be spatially cognizant of its surroundings, learn from and build a map of its environment, figure out how to get from one point in space to another, and execute that movement (which often involves compliant motion, a process where movement requires maintaining physical contact with an object).[112][113]

Within developmental robotics, developmental learning approaches are elaborated upon to allow robots to accumulate repertoires of novel skills through autonomous self-exploration, social interaction with human teachers, and the use of guidance mechanisms (active learning, maturation, motor synergies, etc.).[114][115][116][117]

**Social intelligence**

Affective computing is the study and development of systems that can recognize, interpret, process, and simulate human affects.[119][120] It is an interdisciplinary field spanning computer sciences, psychology, and cognitive science.[121] While the origins of the field may be traced as far back as the early philosophical inquiries into emotion,[122] the more modern branch of computer science originated with Rosalind Picard's 1995 paper[123] on "affective computing".[124][125] A motivation for the research is the ability to simulate empathy, where the machine would be able to interpret human emotions and adapts its behavior to give an appropriate response to those emotions.

[Figure]

Kismet, a robot with rudimentary social skills[118]

Emotion and social skills[126] are important to an intelligent agent for two reasons. First, being able to predict the actions of others by understanding their motives and emotional states allow an agent to make better decisions. Concepts such as game theory, decision theory, necessitate that an agent be able to detect and model human emotions. Second, in an effort to facilitate human–computer interaction, an intelligent machine may want to display emotions (even if it does not experience those emotions itself) to appear more sensitive to the emotional dynamics of human interaction.

**General intelligence**

Many researchers think that their work will eventually be incorporated into a machine with artificial general intelligence, combining all the skills mentioned above and even exceeding human ability in most or all these areas.[17][127] A few believe that anthropomorphic features like artificial consciousness or an artificial brain may be required for such a project.[128][129]

Many of the problems above may also require general intelligence, if machines are to solve the problems as well as people do. For example, even specific straightforward tasks, like machine translation, require that a machine read and write in both languages (NLP), follow the author's argument (reason), know what is being talked about (knowledge), and faithfully reproduce the author's original intent (social intelligence). A problem like machine translation is considered "AI-complete", because all of these problems need to be solved simultaneously in order to reach human-level machine performance.

**Approaches**

There is no established unifying theory or paradigm that guides AI research. Researchers disagree about many issues.[130] A few of the most long standing questions that have remained unanswered are these: should artificial intelligence simulate natural intelligence by studying psychology or neurobiology? Or is human biology as irrelevant to AI research as bird biology is to aeronautical engineering?[14] Can intelligent behavior be described using simple, elegant principles (such as logic or optimization)? Or does it necessarily require solving a large number of completely unrelated problems?[15] Can intelligence be reproduced using high-level symbols, similar to words and ideas? Or does it require "sub-symbolic" processing?[16] John Haugeland, who coined the term GOFAI (Good Old-Fashioned Artificial Intelligence), also proposed that AI should more properly be referred to as synthetic intelligence[131] a term which has since been adopted by some non-GOFAI researchers.[132][133]

Stuart Shapiro divides AI research into three approaches, which he calls computational psychology, computational philosophy, and computer science. Computational psychology is used to make computer programs that mimic human behavior.[134] Computational philosophy, is used to develop an adaptive, free-flowing computer mind.[134] Implementing computer science serves the goal of creating computers that can perform tasks that only people could previously accomplish.[134] Together, the humanesque behavior, mind, and actions make up artificial intelligence.

**Cybernetics and brain simulation**

In the 1940s and 1950s, a number of researchers explored the connection between neurobiology, information theory, and cybernetics. Some of them built machines that used electronic networks to exhibit rudimentary intelligence, such as W. Grey Walter's turtles and the Johns Hopkins Beast. Many of these researchers gathered for meetings of the Teleological Society at Princeton University and the Ratio Club in England.[135] By 1960, this approach was largely abandoned, although elements of it would be revived in the 1980s.

**Symbolic**

When access to digital computers became possible in the middle 1950s, AI research began to explore the possibility that human intelligence could be reduced to symbol manipulation. The research was centered in three institutions: Carnegie Mellon University, Stanford and MIT, and each one developed its own style of research. John Haugeland named these approaches to AI "good old fashioned AI" or "GOFAI".[136] During the 1960s, symbolic approaches had achieved great success at simulating high-level thinking in small demonstration programs. Approaches based on cybernetics or neural networks were abandoned or pushed into the background.[137] Researchers in the 1960s and the 1970s were convinced that symbolic approaches would eventually succeed in creating a machine with artificial general intelligence and considered this the goal of their field.

**Cognitive simulation**

Economist Herbert Simon and Allen Newell studied human problem-solving skills and attempted to formalize them, and their work laid the foundations of the field of artificial intelligence, as well as cognitive science, operations research and management science. Their research team used the results of psychological experiments to develop programs that simulated the techniques that people used to solve problems. This tradition, centered at Carnegie Mellon University would eventually culminate in the development of the Soar architecture in the middle 1980s.[138][139]

**Logic-based**

Unlike Newell and Simon, John McCarthy felt that machines did not need to simulate human thought, but should instead try to find the essence of abstract reasoning and problem solving, regardless of whether people used the same algorithms.[14] His laboratory at Stanford (SAIL) focused on using formal logic to solve a wide variety of problems, including knowledge representation, planning and learning.[140] Logic was also the focus of the work at the University of Edinburgh and elsewhere in Europe which led to the development of the programming language Prolog and the science of logic programming.[141]

**Anti-logic or scruffy**

Researchers at MIT (such as Marvin Minsky and Seymour Papert)[142] found that solving difficult problems in vision and natural language processing required ad-hoc solutions – they argued that there was no simple and general principle (like logic) that would capture all the aspects of intelligent behavior. Roger Schank described their "anti-logic" approaches as "scruffy" (as opposed to the "neat" paradigms at CMU and Stanford).[15] Commonsense knowledge bases (such as Doug Lenat's Cyc) are an example of "scruffy" AI, since they must be built by hand, one complicated concept at a time.[143]

**Knowledge-based**

When computers with large memories became available around 1970, researchers from all three traditions began to build knowledge into AI applications.[144] This "knowledge revolution" led to the development and deployment of expert systems (introduced by Edward Feigenbaum), the first truly successful form of AI software.[37] The knowledge revolution was also driven by the realization that enormous amounts of knowledge would be required by many simple AI applications.

**Sub-symbolic**

By the 1980s progress in symbolic AI seemed to stall and many believed that symbolic systems would never be able to imitate all the processes of human cognition, especially perception, robotics, learning and pattern recognition. A number of researchers began to look into "sub-symbolic" approaches to specific AI problems.[16] Sub-symbolic methods manage to approach intelligence without specific representations of knowledge.

**Embodied intelligence**

This includes embodied, situated, behavior-based, and nouvelle AI. Researchers from the related field of robotics, such as Rodney Brooks, rejected symbolic AI and focused on the basic engineering problems that would allow robots to move and survive.[145] Their work revived the non-symbolic viewpoint of the early cybernetics researchers of the 1950s and reintroduced the use of control theory in AI. This coincided with the development of the embodied mind thesis in the related field of cognitive science: the idea that aspects of the body (such as movement, perception and visualization) are required for higher intelligence.

**Computational intelligence and soft computing**

Interest in neural networks and "connectionism" was revived by David Rumelhart and others in the middle of the 1980s.[146] Neural networks are an example of soft computing --- they are solutions to problems which cannot be solved with complete logical certainty, and where an approximate solution is often sufficient. Other soft computing approaches to AI include fuzzy systems, evolutionary computation and many statistical tools. The application of soft computing to AI is studied collectively by the emerging discipline of computational intelligence.[147]

**Statistical**

In the 1990s, AI researchers developed sophisticated mathematical tools to solve specific subproblems. These tools are truly scientific, in the sense that their results are both measurable and verifiable, and they have been responsible for many of AI's recent successes. The shared mathematical language has also permitted a high level of collaboration with more established fields (like mathematics, economics or operations research). Stuart Russell and Peter Norvig describe this movement as nothing less than a "revolution" and "the victory of the neats".[38] Critics argue that these techniques (with few exceptions[148]) are too focused on

particular problems and have failed to address the long-term goal of general intelligence.[149] There is an ongoing debate about the relevance and validity of statistical approaches in AI, exemplified in part by exchanges between Peter Norvig and Noam Chomsky.[150][151]

**Integrating the approaches**

**Intelligent agent paradigm**

An intelligent agent is a system that perceives its environment and takes actions which maximize its chances of success. The simplest intelligent agents are programs that solve specific problems. More complicated agents include human beings and organizations of human beings (such as firms). The paradigm gives researchers license to study isolated problems and find solutions that are both verifiable and useful, without agreeing on one single approach. An agent that solves a specific problem can use any approach that works – some agents are symbolic and logical, some are sub-symbolic neural networks and others may use new approaches. The paradigm also gives researchers a common language to communicate with other fields—such as decision theory and economics—that also use concepts of abstract agents. The intelligent agent paradigm became widely accepted during the 1990s.[152]

**Agent architectures and cognitive architectures**

Researchers have designed systems to build intelligent systems out of interacting intelligent agents in a multi-agent system.[153] A system with both symbolic and sub-symbolic components is a hybrid intelligent system, and the study of such systems is artificial intelligence systems integration. A hierarchical control system provides a bridge between sub-symbolic AI at its lowest, reactive levels and traditional symbolic AI at its highest levels, where relaxed time constraints permit planning and world modelling.[154] Rodney Brooks' subsumption architecture was an early proposal for such a hierarchical system.

**Tools**

In the course of 60 or so years of research, AI has developed a large number of tools to solve the most difficult problems in computer science. A few of the most general of these methods are discussed below.

**Search and optimization**

Many problems in AI can be solved in theory by intelligently searching through many possible solutions:[155] Reasoning can be reduced to performing a search. For example, logical proof can be viewed as searching for a path that leads from premises to conclusions, where each step is the application of an inference rule.[156] Planning algorithms search through trees of goals and subgoals, attempting to find a path to a target goal, a process called means-ends analysis.[157] Robotics algorithms for moving limbs and grasping objects use local searches in configuration space.[111] Many learning algorithms use search algorithms based on optimization.

Simple exhaustive searches[158] are rarely sufficient for most real world problems: the search space (the number of places to search) quickly grows to astronomical numbers. The result is a search that is too slow or never completes. The solution, for many problems, is to use "heuristics" or "rules of thumb" that prioritize choices in favor of those that are more likely to reach a goal, and to do so in a shorter number of steps. In some search methodologies heuristics can also serve to entirely eliminate some choices that are unlikely to lead to a goal (called "pruning the search tree"). Heuristics supply the program with a "best guess" for the path on which the solution lies.[159] Heuristics limit the search for solutions into a smaller sample size.[112]

A very different kind of search came to prominence in the 1990s, based on the mathematical theory of optimization. For many problems, it is possible to begin the search with some form of a guess and then refine the guess incrementally until no more refinements can be made. These algorithms can be visualized as blind hill climbing: we begin the search at a random point on the

landscape, and then, by jumps or steps, we keep moving our guess uphill, until we reach the top. Other optimization algorithms are simulated annealing, beam search and random optimization[160]

Evolutionary computation uses a form of optimization search. For example, they may begin with a population of organisms (the guesses) and then allow them to mutate and recombine, selecting only the fittest to survive each generation (refining the guesses). Forms of evolutionary computation include swarm intelligence algorithms (such as ant colony or particle swarm optimization)[161] and evolutionary algorithms (such as genetic algorithms, gene expression programming and genetic programming).[162]

**Logic**

Logic[163] is used for knowledge representation and problem solving, but it can be applied to other problems as well. For example, the satplan algorithm uses logic for planning[164] and inductive logic programming is a method for learning.[165]

Several different forms of logic are used in AI research. Propositional or sentential logic[166] is the logic of statements which can be true or false. First-order logic[167] also allows the use of quantifiers and predicates, and can express facts about objects, their properties, and their relations with each other. Fuzzy logic,[168] is a version of first-order logic which allows the truth of a statement to be represented as a value between 0 and 1, rather than simply True (1) or False (0). Fuzzy systems can be used for uncertain reasoning and have been widely used in modern industrial and consumer product control systems. Subjective logic models uncertainty in a different and more explicit manner than fuzzy-logic: a given binomial opinion satisfies belief + disbelief + uncertainty = 1 within a Beta distribution. By this method, ignorance can be distinguished from probabilistic statements that an agent makes with high confidence.

Default logics, non-monotonic logics and circumscription[88] are forms of logic designed to help with default reasoning and the qualification problem. Several extensions of logic have been designed to handle specific domains of knowledge, such as: description logics;[76] situation calculus, event calculus and fluent calculus (for representing events and time);[77] causal calculus;[78] belief calculus;[169] and modal logics[79]

**Probabilistic methods for uncertain reasoning**

Many problems in AI (in reasoning, planning, learning, perception and robotics) require the agent to operate with incomplete or uncertain information. AI researchers have devised a number of powerful tools to solve these problems using methods from probability theory and economics.[170]

Bayesian networks[171] are a very general tool that can be used for a large number of problems: reasoning (using the Bayesian inference algorithm),[172] learning (using the expectation-maximization algorithm),[d][174] planning (using decision networks)[175] and perception (using dynamic Bayesian networks).[176] Bayesian networks are used in AdSense to choose what ads to place and on XBox Live to rate and match players.[177] Probabilistic algorithms can also be used for filtering, prediction, smoothing and finding explanations for streams of data, helping perception systems to analyze processes that occur over time (e.g., hidden Markov models or Kalman filters).[176]

[Figure]

Expectation-maximization clustering of Old Faithful eruption data starts from a random guess but then successfully converges on an accurate clustering of the two physically distinct modes of eruption.

A key concept from the science of economics is "utility": a measure of how valuable something is to an intelligent agent. Precise mathematical tools have been developed that analyze how an agent can make choices and plan, using decision theory, decision analysis,[178] and information value theory.[94] These tools include models such as Markov decision processes,[179] dynamic decision networks[176] game theory and mechanism design[180]

**Classifiers and statistical learning methods**

The simplest AI applications can be divided into two types: classifiers ("if shiny then diamond") and controllers ("if shiny then pick up"). Controllers do, however, also classify conditions before inferring actions, and therefore classification forms a central part of many AI systems. Classifiers are functions that use pattern matching to determine a closest match. They can be tuned according to examples, making them very attractive for use in AI. These examples are known as observations or patterns. In supervised learning, each pattern belongs to a certain predefined class. A class can be seen as a decision that has to be made. All the observations combined with their class labels are known as a data set. When a new observation is received, that observation is classified based on previous experience.[181]

A classifier can be trained in various ways; there are many statistical and machine learning approaches. The decision tree[182] is perhaps the most widely used machine learning algorithm.[183] Other widely used classifiers are the neural network,[184] k-nearest neighbor algorithm,[e][186] kernel methods such as the support vector machine (SVM),[f][188] Gaussian mixture model[189] and the extremely popular naive Bayes classifier.[g][191] The performance of these classifiers have been compared over a wide range of tasks. Classifier performance depends greatly on the characteristics of the data to be classified. There is no single classifier that works best on all given problems; this is also referred to as the "no free lunch" theorem. Determining a suitable classifier for a given problem is still more an art than science.[192]

**Artificial neural networks**

Neural networks, or neural nets, were inspired by the architecture of neurons in the human brain. A simple "neuron" $N$ accepts input from multiple other neurons, each of which, when activated (or "fired"), cast a weighted "vote" for or against whether neuron $N$ should itself activate. Learning requires an algorithm to adjust these weights based on the training data; one simple algorithm (dubbed "fire together, wire together") is to increase the weight between two connected neurons when the activation of one triggers the successful activation of another. The net forms "concepts" that are distributed among a subnetwork of shared[h] neurons that tend to fire together; a concept meaning "leg" might be coupled with a subnetwork meaning "foot" that includes the sound for "foot". Neurons have a continuous spectrum of activation; in addition, neurons can process inputs in a nonlinear way rather than weighing straightforward votes. Modern neural nets can learn both continuous functions and, surprisingly, digital logical operations. Neural networks' early successes included predicting the stock market and (in 1995) a mostly self-driving car.[i][193] In the 2010s, advances in neural networks using deep learning thrust AI

[Figure]

A neural network is an interconnected group of nodes, akin to the vast network of neurons in the human brain.

into widespread public consciousness and contributed to an enormous upshift in corporate AI spending; for example, AI-related M&A in 2017 was over 25 times as large as in 2015.[194][195]

The study of non-learning artificial neural networks[184] began in the decade before the field of AI research was founded, in the work of Walter Pitts and Warren McCulloch Frank Rosenblatt invented the perceptron, a learning network with a single layer, similar to the old concept of linear regression. Early pioneers also include Alexey Grigorevich Ivakhnenko, Teuvo Kohonen, Stephen Grossberg, Kunihiko Fukushima, Christoph von der Malsburg, David Willshaw, Shun-Ichi Amari, Bernard Widrow, John Hopfield, Eduardo R. Caianiello, and others.

The main categories of networks are acyclic or feedforward neural networks (where the signal passes in only one direction) and recurrent neural networks (which allow feedback and short-term memories of previous input events). Among the most popular feedforward networks are perceptrons, multi-layer perceptrons and radial basis networks.[196] Neural networks can be applied to the problem of intelligent control (for robotics) or learning, using such techniques as Hebbian learning ("fire together, wire together"), GMDH or competitive learning[197]

Today, neural networks are often trained by the backpropagation algorithm, which had been around since 1970 as the reverse mode of automatic differentiation published by Seppo Linnainmaa,[198][199] and was introduced to neural networks by Paul Werbos.[200][201][202]

Hierarchical temporal memory is an approach that models some of the structural and algorithmic properties of the neocortex.[203]

In short, most neural networks use some form of gradient descent on a hand-created neural topology. However, some research groups, such as Uber, argue that simple neuroevolution to mutate new neural network topologies and weights may be competitive with sophisticated gradient descent approaches. One advantage of neuroevolution is that it may be less prone to get caught in "dead ends".[204]

**Deep feedforward neural networks**

Deep learning is any artificial neural network that can learn a long chain of causal links. For example, a feedforward network with six hidden layers can learn a seven-link causal chain (six hidden layers + output layer) and has a "credit assignment path" (CAP) depth of seven. Many deep learning systems need to be able to learn chains ten or more causal links in length.[205] Deep learning has transformed many important subfields of artificial intelligence, including computer vision, speech recognition, natural language processing and others.[206][207][205]

According to one overview,[208] the expression "Deep Learning" was introduced to the Machine Learning community by Rina Dechter in 1986[209] and gained traction after Igor Aizenberg and colleagues introduced it to Artificial Neural Networks in 2000.[210] The first functional Deep Learning networks were published by Alexey Grigorevich Ivakhnenko and V. G. Lapa in 1965.[211] These networks are trained one layer at a time. Ivakhnenko's 1971 paper[212] describes the learning of a deep feedforward multilayer perceptron with eight layers, already much deeper than many later networks. In 2006, a publication by Geoffrey Hinton and Ruslan Salakhutdinov introduced another way of pre-training many-layered feedforward neural networks (FNNs) one layer at a time, treating each layer in turn as an unsupervised restricted Boltzmann machine, then using supervised backpropagation for fine-tuning.[213] Similar to shallow artificial neural networks, deep neural networks can model complex non-linear relationships. Over the last few years, advances in both machine learning algorithms and computer hardware have led to more efficient methods for training deep neural networks that contain many layers of non-linear hidden units and a very large output layer.[214]

Deep learning often uses convolutional neural networks (CNNs), whose origins can be traced back to the Neocognitron introduced by Kunihiko Fukushima in 1980.[215] In 1989, Yann LeCun and colleagues applied backpropagation to such an architecture. In the early 2000s, in an industrial application CNNs already processed an estimated 10% to 20% of all the checks written in the US.[216] Since 2011, fast implementations of CNNs on GPUs have won many visual pattern recognition competitions.[205]

CNNs with 12 convolutional layers were used in conjunction with reinforcement learning by Deepmind's "AlphaGo Lee", the program that beat a top Go champion in 2016.[217]

**Deep recurrent neural networks**

Early on, deep learning was also applied to sequence learning with recurrent neural networks (RNNs)[218] which are in theory Turing complete[219] and can run arbitrary programs to process arbitrary sequences of inputs. The depth of an RNN is unlimited and depends on the length of its input sequence; thus, an RNN is an example of deep learning.[205] RNNs can be trained by gradient descent[220][221][222] but suffer from the vanishing gradient problem.[206][223] In 1992, it was shown that unsupervised pre-training of a stack of recurrent neural networks can speed up subsequent supervised learning of deep sequential problems.[224]

Numerous researchers now use variants of a deep learning recurrent NN called the long short-term memory (LSTM) network published by Hochreiter & Schmidhuber in 1997.[225] LSTM is often trained by Connectionist Temporal Classification (CTC).[226] At Google, Microsoft and Baidu this approach has revolutionised speech recognition[227][228][229] For example, in 2015, Google's speech recognition experienced a dramatic performance jump of 49% through CTC-trained LSTM, which is now available through Google Voice to billions of smartphone users.[230] Google also used LSTM to improve machine translation,[231] Language Modeling[232] and Multilingual Language Processing.[233] LSTM combined with CNNs also improved automatic image captioning[234] and a plethora of other applications.

**Languages**

Early symbolic AI inspired Lisp[235] and Prolog,[236] which dominated early AI programming. Modern AI development often uses mainstream languages such as Python or C++,[237] or niche languages such as Wolfram Language[238]

**Evaluating progress**

In 1950, Alan Turing proposed a general procedure to test the intelligence of an agent now known as the Turing test. This procedure allows almost all the major problems of artificial intelligence to be tested. However, it is a very difficult challenge and at present all agents fail.[239]

Artificial intelligence can also be evaluated on specific problems such as small problems in chemistry, hand-writing recognition and game-playing. Such tests have been termed subject matter expert Turing tests. Smaller problems provide more achievable goals and there are an ever-increasing number of positive results.

For example, performance at draughts (i.e. checkers) is optimal, performance at chess is high-human and nearing super-human (see computer chess: computers versus human) and performance at many everyday tasks (such as recognizing a face or crossing a room without bumping into something) is sub-human.

A quite different approach measures machine intelligence through tests which are developed from *mathematical* definitions of intelligence. Examples of these kinds of tests start in the late nineties devising intelligence tests using notions from Kolmogorov complexity and data compression.[240] Two major advantages of mathematical definitions are their applicability to nonhuman intelligences and their absence of a requirement for human testers.

A derivative of the Turing test is the Completely Automated Public Turing test to tell Computers and Humans Apart (CAPTCHA). As the name implies, this helps to determine that a user is an actual person and not a computer posing as a human. In contrast to the standard Turing test, CAPTCHA is administered by a machine and targeted to a human as opposed to being administered by a human and targeted to a machine. A computer asks a user to complete a simple test then generates a grade for that test. Computers are unable to solve the problem, so correct solutions are deemed to be the result of a person taking the test. A common type of CAPTCHA is the test that requires the typing of distorted letters, numbers or symbols that appear in an image undecipherable by a computer.[241]

**Applications**

AI is relevant to any intellectual task.[242] Modern artificial intelligence techniques are pervasive and are too numerous to list here. Frequently, when a technique reaches mainstream use, it is no longer considered artificial intelligence; this phenomenon is described as the AI effect.[243]

High-profile examples of AI include autonomous vehicles (such as drones and self-driving cars), medical diagnosis, creating art (such as poetry), proving mathematical theorems, playing games (such as Chess or Go), search engines (such as Google search), online assistants (such as Siri), image recognition in photographs, spam filtering, prediction of judicial decisions[244] and targeting online advertisements.[242][245][246]

With social media sites overtaking TV as a source for news for young people and news organisations increasingly reliant on social media platforms for generating distribution,[247] major publishers now use artificial intelligence (AI) technology to post stories more effectively and generate highervolumes of traffic.[248]

**Competitions and prizes**

There are a number of competitions and prizes to promote research in artificial intelligence. The main areas promoted are: general machine intelligence, conversational behavior, data-mining, robotic cars, robot soccer and games.

[Figure]

An automated online assistant providing customer service on a web page – one of many very primitive applications of artificial intelligence.

**Healthcare**

Artificial intelligence is breaking into the healthcare industry by assisting doctors. According to Bloomberg Technology, Microsoft has developed AI to help doctors find the right treatments for cancer.[249] There is a great amount of research and drugs developed relating to cancer. In detail, there are more than 800 medicines and vaccines to treat cancer. This negatively affects the doctors, because there are too many options to choose from, making it more difficult to choose the right drugs for the patients. Microsoft is working on a project to develop a machine called "Hanover". Its goal is to memorize all the papers necessary to cancer and help predict which combinations of drugs will be most effective for each patient. One project that is being worked on at the moment is fighting myeloid leukemia, a fatal cancer where the treatment has not improved in decades. Another study was reported to have found that artificial intelligence was as good as trained doctors in identifying skin cancers.[250] Another study is using artificial intelligence to try and monitor multiple high-risk patients, and this is done by asking each patient numerous questions based on data acquired from live doctor to patient interaction.[251]

[Figure]

A patient-side surgical arm ofDa Vinci Surgical System

According to CNN, there was a recent study by surgeons at the Children's National Medical Center in Washington which successfully demonstrated surgery with an autonomous robot. The team supervised the robot while it performed soft-tissue surgery, stitching together a pig's bowel during open surgery, and doing so better than a human surgeon, the team claimed.[252] IBM has created its own artificial intelligence computer, the IBM Watson, which has beaten human intelligence (at some levels). Watson not only won at the game show *Jeopardy!* against former champions,[253] but, was declared a hero after successfully diagnosing a women who was suffering from leukemia.[254]

**Automotive**

Advancements in AI have contributed to the growth of the automotive industry through the creation and evolution of self-driving vehicles. As of 2016, there are over 30 companies utilizing AI into the creation of driverless cars. A few companies involved with AI include Tesla, Google, and Apple.[255]

Many components contribute to the functioning of self-driving cars. These vehicles incorporate systems such as braking, lane changing, collision prevention, navigation and mapping. Together, these systems, as well as high performance computers, are integrated into one complex vehicle.[256]

Recent developments in autonomous automobiles have made the innovation of self-driving trucks possible, though they are still in the testing phase. The UK government has passed legislation to begin testing of self-driving truck platoons in 2018.[257] Self-driving truck platoons are a fleet of self-driving trucks following the lead of one non-self-driving truck, so the truck platoons aren't entirely autonomous yet. Meanwhile, the Daimler, a German automobile corporation, is testing the Freightliner Inspiration which is a semi-autonomous truck that will only be used on the highway.[258]

[Figure]

X-ray of a hand, with automatic calculation of bone age by a computer software.

One main factor that influences the ability for a driver-less automobile to function is mapping. In general, the vehicle would be pre-programmed with a map of the area being driven. This map would include data on the approximations of street light and curb heights in order for the vehicle to be aware of its surroundings. However, Google has been working on an algorithm with the purpose of eliminating the need for pre-programmed maps and instead, creating a device that would be able to adjust to a variety of new surroundings.[259] Some self-driving cars are not equipped with steering wheels or brake pedals, so there has also been research focused on creating an algorithm that is capable of maintaining a safe environment for the passengers in the vehicle through awareness of speed and driving conditions.[260]

Another factor that is influencing the ability for a driver-less automobile is the safety of the passenger. To make a driver-less automobile, engineers must program it to handle high risk situations. These situations could include a head on collision with pedestrians. The car's main goal should be to make a decision that would avoid hitting the pedestrians and saving the passengers in the car. But there is a possibility the car would need to make a decision that would put someone in danger. In other words, the car would need to decide to save the pedestrians or the passengers.[261] The programing of the car in these situations is crucial to a successful driver-less automobile.

**Finance and economics**

Financial institutions have long used artificial neural network systems to detect charges or claims outside of the norm, flagging these for human investigation. The use of AI in banking can be traced back to 1987 when Security Pacific National Bank in US set-up a Fraud Prevention Task force to counter the unauthorised use of debit cards. Programs like Kasisto and Moneystream are using AI in financial services.

Banks use artificial intelligence systems today to organize operations, maintain book-keeping, invest in stocks, and manage properties. AI can react to changes overnight or when business is not taking place.[262] In August 2001, robots beat humans in a simulated financial trading competition.[263] AI has also reduced fraud and financial crimes by monitoring behavioral patterns of users for any abnormal changes or anomalies.[264]

The use of AI machines in the market in applications such as online trading and decision making has changed major economic theories.[265] For example, AI based buying and selling platforms have changed the law of supply and demand in that it is now possible to easily estimate individualized demand and supply curves and thus individualized pricing. Furthermore, AI machines reduce information asymmetry in the market and thus making markets more efficient while reducing the volume of trades. Furthermore, AI in the markets limits the consequences of behavior in the markets again making markets more efficient. Other theories where AI has had impact include in rational choice, rational expectations, game theory, Lewis turning point, portfolio optimization and counterfactual thinking

**Video games**

In video games, artificial intelligence is routinely used to generate dynamic purposeful behavior in non-player characters (NPCs). In addition, well-understood AI techniques are routinely used for pathfinding. Some researchers consider NPC AI in games to be a "solved problem" for most production tasks. Games with more atypical AI include the AI director of *Left 4 Dead* (2008) and the neuroevolutionary training of platoons in *Supreme Commander 2* (2010).[266][267]

**Military**

Worldwide annual military spending on robotics rose from 5.1 billion USD in 2010 to 7.5 billion USD in 2015.[268][269] Military drones capable of autonomous action are widely considered a useful asset. In 2017, Vladimir Putin stated that "Whoever becomes the leader in (artificial intelligence) will become the ruler of the world".[270][271] Many artificial intelligence researchers seek to distance themselves from military applications of AI.[272]

**Platforms**

A platform (or "computing platform") is defined as "some sort of hardware architecture or software framework (including application frameworks), that allows software to run". As Rodney Brooks pointed out many years ago,[273] it is not just the artificial intelligence software that defines the AI features of the platform, but rather the actual platform itself that affects the AI that results, i.e., there needs to be work in AI problems on real-world platforms rather than in isolation.

A wide variety of platforms has allowed different aspects of AI to develop, ranging from expert systems such as Cyc to deep-learning frameworks to robot platforms such as the Roomba with open interface.[274] Recent advances in deep artificial neural networks and distributed computing have led to a proliferation of software libraries, including Deeplearning4j, TensorFlow, Theano and Torch.

Collective AI is a platform architecture that combines individual AI into a collective entity, in order to achieve global results from individual behaviors.[275][276] With its collective structure, developers can crowdsource information and extend the functionality of existing AI domains on the platform for their own use, as well as continue to create and share new domains and capabilities for the wider community and greater good.[277] As developers continue to contribute, the overall platform grows more intelligent and is able to perform more requests, providing a scalable model for greater communal benefit.[276] Organizations like SoundHound Inc. and the Harvard John A. Paulson School of Engineering and Applied Sciences have used this collaborative AI model.[278][276]

**Education in AI**

A McKinsey Global Institute study found a shortage of 1.5 million highly trained data and AI professionals and managers[279] and a number of private bootcamps have developed programs to meet that demand, including free programs like The Data Incubator or paid programs like General Assembly.[280]

**Partnership on AI**

Amazon, Google, Facebook, IBM, and Microsoft have established a non-profit partnership to formulate best practices on artificial intelligence technologies, advance the public's understanding, and to serve as a platform about artificial intelligence.[281] They stated: "This partnership on AI will conduct research, organize discussions, provide thought leadership, consult with relevant third parties, respond to questions from the public and media, and create educational material that advance the understanding of AI technologies including machine perception, learning, and automated reasoning."[281] Apple joined other tech companies as a founding member of the Partnership on AI in January 2017. The corporate members will make financial and research contributions to the group, while engaging with the scientific community to bring academics onto the board.[282][276]

**Philosophy and ethics**

There are three philosophical questions related to AI:

1. Is artificial general intelligence possible? Can a machine solve any problem that a human being can solve using intelligence? Or are there hard limits to what a machine can accomplish?
2. Are intelligent machines dangerous? How can we ensure that machines behave ethically and that they are used ethically?
3. Can a machine have a mind, consciousness and mental states in exactly the same sense that human beings do? Can a machine be sentient, and thus deserve certain rights? Can a machine intentionally cause harm?

**The limits of artificial general intelligence**

Can a machine be intelligent? Can it "think"?

**Alan Turing's "polite convention"**

We need not decide if a machine can "think"; we need only decide if a machine can act as intelligently as a human being. This approach to the philosophical problems associated with artificial intelligence forms the basis of the Turing test.[239]

**The Dartmouth proposal**

"Every aspect of learning or any other feature of intelligence can be so precisely described that a machine can be made to simulate it." This conjecture was printed in the proposal for the Dartmouth Conference of 1956, and represents the position of most working AI researchers.[283]

**Newell and Simon's physical symbol system hypothesis**

"A physical symbol system has the necessary and sufficient means of general intelligent action." Newell and Simon argue that intelligence consists of formal operations on symbols.[284] Hubert Dreyfus argued that, on the contrary, human expertise depends on unconscious instinct rather than conscious symbol manipulation and on having a "feel" for the situation rather than explicit symbolic knowledge. (See Dreyfus' critique of AI.)[285][286]

**Gödelian arguments**

Gödel himself,[287] John Lucas (in 1961) and Roger Penrose (in a more detailed argument from 1989 onwards) made highly technical arguments that human mathematicians can consistently see the truth of their own "Gödel statements" and therefore have computational abilities beyond that of mechanical Turing machines.[288] However, the modern consensus in the scientific and mathematical community is that these "Gödelian arguments" fail.[289][290][291]

**The artificial brain argument**

The brain can be simulated by machines and because brains are intelligent, simulated brains must also be intelligent; thus machines can be intelligent. Hans Moravec, Ray Kurzweil and others have argued that it is technologically feasible to copy the brain directly into hardware and software, and that such a simulation will be essentially identical to the original.[129]

**The AI effect**

Machines are *already* intelligent, but observers have failed to recognize it. When Deep Blue beat Garry Kasparov in chess, the machine was acting intelligently. However, onlookers commonly discount the behavior of an artificial intelligence program by arguing that it is not "real" intelligence after all; thus "real" intelligence is whatever intelligent behavior people can do that machines still cannot. This is known as the AI Effect: "AI is whatever hasn't been done yet."

**Potential risks and moral reasoning**

Widespread use of artificial intelligence could have unintended consequences that are dangerous or undesirable. Scientists from the Future of Life Institute, among others, described some short-term research goals to see how AI influences the economy, the laws and ethics that are involved with AI and how to minimize AI security risks. In the long-term, the scientists have proposed to continue

optimizing function while minimizing possible security risks that come along with new technologies.[292]

Machines with intelligence have the potential to use their intelligence to make ethical decisions. Research in this area includes "machine ethics", "artificial moral agents", and the study of "malevolent vs. friendly AI".

**Existential risk**

> The development of full artificial intelligence could spell the end of the human race. Once humans develop artificial intelligence, it will take off on its own and redesign itself at an ever-increasing rate. Humans, who are limited by slow biological evolution, couldn't compete and would be superseded.
>
> — Stephen Hawking[293]

A common concern about the development of artificial intelligence is the potential threat it could pose to humanity. This concern has recently gained attention after mentions by celebrities including the late Stephen Hawking, Bill Gates,[294] and Elon Musk.[295] A group of prominent tech titans including Peter Thiel, Amazon Web Services and Musk have committed $1billion to OpenAI a nonprofit company aimed at championing responsible AI development.[296] The opinion of experts within the field of artificial intelligence is mixed, with sizable fractions both concerned and unconcerned by risk from eventual superhumanly-capable AI.[297]

In his book Superintelligence, Nick Bostrom provides an argument that artificial intelligence will pose a threat to mankind. He argues that sufficiently intelligent AI, if it chooses actions based on achieving some goal, will exhibit convergent behavior such as acquiring resources or protecting itself from being shut down. If this AI's goals do not reflect humanity's – one example is an AI told to compute as many digits of pi as possible – it might harm humanity in order to acquire more resources or prevent itself from being shut down, ultimately to better achieve its goal.

For this danger to be realized, the hypothetical AI would have to overpower or out-think all of humanity, which a minority of experts argue is a possibility far enough in the future to not be worth researching.[298][299] Other counterarguments revolve around humans being either intrinsically or convergently valuable from the perspective of an artificial intelligence.[300]

Concern over risk from artificial intelligence has led to some high-profile donations and investments. In January 2015, Elon Musk donated ten million dollars to the Future of Life Institute to fund research on understanding AI decision making. The goal of the institute is to "grow wisdom with which we manage" the growing power of technology. Musk also funds companies developing artificial intelligence such as Google DeepMind and Vicarious to "just keep an eye on what's going on with artificial intelligence.[301] I think there is potentially a dangerous outcome there.[302][303]

Development of militarized artificial intelligence is a related concern. Currently, 50+ countries are researching battlefield robots, including the United States, China, Russia, and the United Kingdom. Many people concerned about risk from superintelligent AI also want to limit the use of artificial soldiers.[304]

**Devaluation of humanity**

Joseph Weizenbaum wrote that AI applications cannot, by definition, successfully simulate genuine human empathy and that the use of AI technology in fields such as customer service or psychotherapy[305] was deeply misguided. Weizenbaum was also bothered that AI researchers (and some philosophers) were willing to view the human mind as nothing more than a computer program (a position now known as computationalism). To Weizenbaum these points suggest that AI research devalues human life.[306]

**Decrease in demand for human labor**

The relationship between automation and employment is complicated. While automation eliminates old jobs, it also creates new jobs through micro-economic and macro-economic effects.[307] Unlike previous waves of automation, many middle-class jobs may be eliminated by artificial intelligence; The Economist states that "the worry that AI could do to white-collar jobs what steam power did to blue-collar ones during the Industrial Revolution" is "worth taking seriously".[308] Subjective estimates of the risk vary widely; for

example, Michael Osborne and Carl Benedikt Frey estimate 47% of U.S. jobs are at "high risk" of potential automation, while an OECD report classifies only 9% of U.S. jobs as "high risk".[309][310][311] Jobs at extreme risk range from paralegals to fast food cooks, while job demand is likely to increase for care-related professions ranging from personal healthcare to the clergy.[312] Author Martin Ford and others go further and argue that a large number of jobs are routine, repetitive and (to an AI) predictable; Ford warns that these jobs may be automated in the next couple of decades, and that many of the new jobs may not be "accessible to people with average capability", even with retraining. Economists point out that in the past technology has tended to increase rather than reduce total employment, but acknowledge that "we're in uncharted territory" with AI.[21]

**Artificial moral agents**

This raises the issue of how ethically the machine should behave towards both humans and other AI agents. This issue was addressed by Wendell Wallach in his book titled *Moral Machines* in which he introduced the concept of artificial moral agents (AMA).[313] For Wallach, AMAs have become a part of the research landscape of artificial intelligence as guided by its two central questions which he identifies as "Does Humanity Want Computers Making Moral Decisions"[314] and "Can (Ro)bots Really Be Moral".[315] For Wallach the question is not centered on the issue of *whether* machines can demonstrate the equivalent of moral behavior in contrast to the *constraints* which society may place on the development of AMAs.[316]

**Machine ethics**

The field of machine ethics is concerned with giving machines ethical principles, or a procedure for discovering a way to resolve the ethical dilemmas they might encounter, enabling them to function in an ethically responsible manner through their own ethical decision making.[317] The field was delineated in the AAAI Fall 2005 Symposium on Machine Ethics: "Past research concerning the relationship between technology and ethics has largely focused on responsible and irresponsible use of technology by human beings, with a few people being interested in how human beings ought to treat machines. In all cases, only human beings have engaged in ethical reasoning. The time has come for adding an ethical dimension to at least some machines. Recognition of the ethical ramifications of behavior involving machines, as well as recent and potential developments in machine autonomy, necessitate this. In contrast to computer hacking, software property issues, privacy issues and other topics normally ascribed to computer ethics, machine ethics is concerned with the behavior of machines towards human users and other machines. Research in machine ethics is key to alleviating concerns with autonomous systems—it could be argued that the notion of autonomous machines without such a dimension is at the root of all fear concerning machine intelligence. Further, investigation of machine ethics could enable the discovery of problems with current ethical theories, advancing our thinking about Ethics."[318] Machine ethics is sometimes referred to as machine morality, computational ethics or computational morality. A variety of perspectives of this nascent field can be found in the collected edition "Machine Ethics"[317] that stems from the AAAI Fall 2005 Symposium on Machine Ethics.[318]

**Malevolent and friendly AI**

Political scientist Charles T. Rubin believes that AI can be neither designed nor guaranteed to be benevolent.[319] He argues that "any sufficiently advanced benevolence may be indistinguishable from malevolence." Humans should not assume machines or robots would treat us favorably, because there is no *a priori* reason to believe that they would be sympathetic to our system of morality, which has evolved along with our particular biology (which AIs would not share). Hyper-intelligent software may not necessarily decide to support the continued existence of humanity, and would be extremely difficult to stop. This topic has also recently begun to be discussed in academic publications as a real source of risks to civilization, humans, and planet Earth.

Physicist Stephen Hawking, Microsoft founder Bill Gates, and SpaceX founder Elon Musk have expressed concerns about the possibility that AI could evolve to the point that humans could not control it, with Hawking theorizing that this could "spell the end of the human race".[320]

One proposal to deal with this is to ensure that the first generally intelligent AI is 'Friendly AI', and will then be able to control subsequently developed AIs. Some question whether this kind of check could really remain in place.

Leading AI researcher Rodney Brooks writes, "I think it is a mistake to be worrying about us developing malevolent AI anytime in the next few hundred years. I think the worry stems from a fundamental error in not distinguishing the difference between the very real recent advances in a particular aspect of AI, and the enormity and complexity of building sentient volitional intelligence."[321]

**Machine consciousness, sentience and mind**

If an AI system replicates all key aspects of human intelligence, will that system also be sentient – will it have a mind which has conscious experiences? This question is closely related to the philosophical problem as to the nature of human consciousness, generally referred to as the hard problem of consciousness.

**Consciousness**

**Computationalism and functionalism**

Computationalism is the position in the philosophy of mind that the human mind or the human brain (or both) is an information processing system and that thinking is a form of computing.[322] Computationalism argues that the relationship between mind and body is similar or identical to the relationship between software and hardware and thus may be a solution to the mind-body problem. This philosophical position was inspired by the work of AI researchers and cognitive scientists in the 1960s and was originally proposed by philosophers Jerry Fodor and Hilary Putnam.

**Strong AI hypothesis**

The philosophical position that John Searle has named "strong AI" states: "The appropriately programmed computer with the right inputs and outputs would thereby have a mind in exactly the same sense human beings have minds."[323] Searle counters this assertion with his Chinese room argument, which asks us to look *inside* the computer and try to find where the "mind" might be.[324]

**Robot rights**

Mary Shelley's *Frankenstein* considers a key issue in the ethics of artificial intelligence: if a machine can be created that has intelligence, could it also *feel*? If it can feel, does it have the same rights as a human? The idea also appears in modern science fiction such as the film *A.I.: Artificial Intelligence*, in which humanoid machines have the ability to feel emotions. This issue, now known as "robot rights", is currently being considered by, for example, California's Institute for the Future, although many critics believe that the discussion is premature.[325] Some critics of transhumanism argue that any hypothetical robot rights would lie on a spectrum with animal rights and human rights.[326] The subject is profoundly discussed in the 2010 documentary film *Plug & Pray*.[327]

**Superintelligence**

Are there limits to how intelligent machines – or human-machine hybrids – can be? A superintelligence, hyperintelligence, or superhuman intelligence is a hypothetical agent that would possess intelligence far surpassing that of the brightest and most gifted human mind. ''Superintelligence'' may also refer to the form or degree of intelligence possessed by such an agent.[127]

**Technological singularity**

If research into Strong AI produced sufficiently intelligent software, it might be able to reprogram and improve itself. The improved software would be even better at improving itself, leading to recursive self-improvement.[328] The new intelligence could thus increase exponentially and dramatically surpass humans. Science fiction writer Vernor Vinge named this scenario "singularity".[329] Technological singularity is when accelerating progress in technologies will cause a runaway effect wherein artificial intelligence will exceed human intellectual capacity and control, thus radically changing or even ending civilization. Because the capabilities of such an intelligence may be impossible to comprehend, the technological singularity is an occurrence beyond which events are unpredictable or even unfathomable.[329][127]

Ray Kurzweil has used Moore's law (which describes the relentless exponential improvement in digital technology) to calculate that desktop computers will have the same processing power as human brains by the year 2029, and predicts that the singularity will occur in 2045.[329]

**Transhumanism**

> You awake one morning to find your brain has another lobe functioning. Invisible, this auxiliary lobe answers your questions with information beyond the realm of your own memory, suggests plausible courses of action, and asks questions that help bring out relevant facts. You quickly come to rely on the new lobe so much that you stop wondering how it works. You just use it. This is the dream of artificial intelligence.
>
> — *Byte*, April 1985[330]

Robot designer Hans Moravec, cyberneticist Kevin Warwick and inventor Ray Kurzweil have predicted that humans and machines will merge in the future into cyborgs that are more capable and powerful than either.[331] This idea, called transhumanism, which has roots in Aldous Huxley and Robert Ettinger, has been illustrated in fiction as well, for example in the manga *Ghost in the Shell* and the science-fiction series *Dune*.

In the 1980s artist Hajime Sorayama's Sexy Robots series were painted and published in Japan depicting the actual organic human form with lifelike muscular metallic skins and later "the Gynoids" book followed that was used by or influenced movie makers including George Lucas and other creatives. Sorayama never considered these organic robots to be real part of nature but always unnatural product of the human mind, a fantasy existing in the mind even when realized in actual form.

Edward Fredkin argues that "artificial intelligence is the next stage in evolution", an idea first proposed by Samuel Butler's "Darwin among the Machines" (1863), and expanded upon by George Dyson in his book of the same name in 1998.[332]

**In fiction**

Thought-capable artificial beings appeared as storytelling devices since antiquity.[23]

The implications of a constructed machine exhibiting artificial intelligence have been a persistent theme in science fiction since the twentieth century. Early stories typically revolved around intelligent robots. The word "robot" itself was coined by Karel Čapek in his 1921 play *R.U.R.*, the title standing for "Rossum's Universal Robots". Later, the SF writer Isaac Asimov developed the Three Laws of Robotics. He subsequently explored these in his many books, most notably the "Multivac" series about a super-intelligent computer of the same name. Asimov's laws are often brought up during layman discussions of machine ethics;[333] while almost all artificial intelligence researchers are familiar with Asimov's laws through popular culture, they generally consider the laws useless for many reasons, one of which is their ambiguity.[334]

[Figure]

Three synthetic beings (right) in the 1921 play *R.U.R.*

The novel *Do Androids Dream of Electric Sheep?*, by Philip K. Dick, tells a science fiction story about Androids and humans clashing in a futuristic world. Elements of artificial intelligence include the empathy box, mood organ, and the androids themselves. Throughout the novel, Dick portrays the idea that human subjectivity is altered by technology created with artificial intelligence.[335]

Nowadays AI is firmly rooted in popular culture; intelligent robots appear in innumerable works. HAL 9000, the murderous computer in charge of the *Discovery One* spaceship in Arthur C. Clarke's and Stanley Kubrick's *2001: A Space Odyssey* (both 1968), is an example of the common "robotic rampage" archetype in science fiction movies. *The Terminator* (1984) and *The Matrix* (1999) provide additional widely familiar examples. In contrast, the rare loyal robots such as Gort from *The Day the Earth Stood Still* (1951) and Bishop from *Aliens* (1986) are less prominent in popular culture.[336]

**See also**

- Abductive reasoning
- Behavior selection algorithm
- Case-based reasoning
- Commonsense reasoning
- Emergent algorithm
- Evolutionary computing
- Glossary of artificial intelligence
- Machine learning
- Mathematical optimization
- Soft computing
- Swarm intelligence
- Weak AI

**Explanatory notes**

a. The act of doling out rewards can itself be formalized or automated into a "reward function".

b. Terminology varies; see algorithm characterizations

c. Adversarial vulnerabilities can also result in nonlinear systems, or from non-pattern perturbations. Some systems are so brittle that changing a single adversarial pixel predictably induces misclassification.

d. Expectation-maximization, one of the most popular algorithms in machine learning, allows clustering in the presence of unknown latent variables[173]

e. The most widely used analogical AI until the mid-1990s[185]

f. SVM displaced k-nearest neighbor in the 1990s[187]

g. Reportedly the "most widely used learner" at Google due in part to its scalability[190]

h. Each individual neuron is likely to participate in more than one concept.

i. Steering for the 1995 "No Hands Across America" required "only a few human assists".

**References**

1. Definition of AI as the study of intelligent agents:
   - Poole, Mackworth & Goebel 1998, p. 1 (http://people.cs.ubc.ca/~poole/ci/ch1.pdf), which provides the version that is used in this article. Note that they use the term "computational intelligence" as a synonym for artificial intelligence.
   - Russell & Norvig (2003) (who prefer the term "rational agent") and write "The whole-agent view is now widely accepted in the field" (Russell & Norvig 2003, p. 55).
   - Nilsson 1998
   - Legg & Hutter 2007.
2. Russell & Norvig 2009, p. 2.
3. Hofstadter (1980, p. 601)
4. Schank, Roger C. (1991). "Where's the AI". *AI magazine*. Vol. 12 no. 4. p. 38.
5. Russell & Norvig 2009
6. "AlphaGo – Google DeepMind" (https://deepmind.com/alpha-go.html). Archived (https://web.archive.org/web/20160310191926/https://wwwdeepmind.com/alpha-go.html) from the original on 10 March 2016.
7. Optimism of early AI:
   - Herbert Simon quote: Simon 1965, p. 96 quoted in Crevier 1993, p. 109.
   - Marvin Minsky quote: Minsky 1967, p. 2 quoted in Crevier 1993, p. 109.
8. Boom of the 1980s: rise of expert systems, Fifth Generation Project, Alvey, MCC, SCI:
   - McCorduck 2004, pp. 426–441
   - Crevier 1993, pp. 161–162,197–203, 211, 240
   - Russell & Norvig 2003, p. 24
   - NRC 1999, pp. 210–211

9. First AI Winter, Mansfield Amendment, Lighthill report
    - Crevier 1993, pp. 115–117
    - Russell & Norvig 2003 p. 22
    - NRC 1999, pp. 212–213
    - Howe 1994

10. Second AI winter:
    - McCorduck 2004, pp. 430–435
    - Crevier 1993, pp. 209–210
    - NRC 1999, pp. 214–216

11. AI becomes hugely successful in the early 21st century
    - Clark 2015

12. Pamela McCorduck (2004, pp. 424) writes of "the rough shattering of AI in subfields—vision, natural language, decision theory, genetic algorithms, robotics .. and these with own sub-subfield—that would hardly have anything to say to each other."

13. This list of intelligent traits is based on the topics covered by the major AI textbooks, including:
    - Russell & Norvig 2003
    - Luger & Stubblefield 2004
    - Poole, Mackworth & Goebel 1998
    - Nilsson 1998

14. Biological intelligence vs. intelligence in general:
    - Russell & Norvig 2003 pp. 2–3, who make the analogy with aeronautical engineering.
    - McCorduck 2004, pp. 100–101, who writes that there are "two major branches of artificial intelligence: one aimed at producing intelligent behavior regardless of how it was accomplished, and the other aimed at modeling intelligent processes found in nature, particularly human ones."
    - Kolata 1982, a paper in Science, which describes McCarthy's indifference to biological models. Kolata quotes McCarthy as writing: "This is AI, so we don't care if it's psychologically real" "Archived copy" (https://books.google.com/books?id=PEkqAAAAMAAJ&q=%22we+don't+care+if+it's+psychologically+real%22&dq=%22we+don't+care+if+it's+psychologically+real%22&output=html&pgis=1) Archived (https://web.archive.org/web/20160707182203/https://books.google.com/books?id=PEkqAAAAMAAJ&q=%22we+don't+care+if+it's+psychologically+real%22&dq=%22we+don't+care+if+it's+psychologically+real%22&output=html&pgis=1) from the original on 7 July 2016 Retrieved 16 February 2016.. McCarthy recently reiterated his position at the AI@50 conference where he said "Artificial intelligence is not, by definition, simulation of human intelligence" (Maker 2006).

15. Neats vs. scruffies:
    - McCorduck 2004, pp. 421–424, 486–489
    - Crevier 1993, pp. 168
    - Nilsson 1983, pp. 10–11

16. Symbolic vs. sub-symbolic AI:
    - Nilsson (1998, p. 7), who uses the term "sub-symbolic".

17. General intelligence (strong AI) is discussed in popular introductions to AI:
    - Kurzweil 1999 and Kurzweil 2005

18. See the Dartmouth proposal under Philosophy, below.

19. This is a central idea of Pamela McCorduck's Machines Who Think She writes: "I like to think of artificial intelligence as the scientific apotheosis of a venerable cultural tradition." (McCorduck 2004, p. 34) "Artificial intelligence in one form or another is an idea that has pervaded Western intellectual history, a dream in urgent need of being realized." (McCorduck 2004, p. xviii) "Our history is full of attempts—nutty, eerie, comical, earnest, legendary and real—to make artificial intelligences, to reproduce what is the essential us—bypassing the ordinary means. Back and forth between myth and reality, our imaginations supplying what our workshops couldn't, we have engaged for a long time in this odd form of self-reproduction." (McCorduck 2004, p. 3) She traces the desire back to its Hellenistic roots and calls it the urge to "forge the Gods." (McCorduck 2004, pp. 340–400)

20. "Stephen Hawking believes AI could be mankind's last accomplishment" (https://betanews.com/2016/10/21/artificial-intelligence-stephen-hawking/) *BetaNews*. 21 October 2016. Archived (https://web.archive.org/web/20170828183930/https://betanews.com/2016/10/21/artificial-intelligence-stephen-hawking/) from the original on 28 August 2017.

21. Ford, Martin; Colvin, Geoff (6 September 2015). "Will robots create more jobs than they destroy?" (https://www.theguardian.com/technology/2015/sep/06/will-robots-create-destroy-jobs) *The Guardian*. Retrieved 13 January 2018.

22. AI applications widely used behind the scenes:
    - Russell & Norvig 2003, p. 28
    - Kurzweil 2005, p. 265
    - NRC 1999, pp. 216–222

23. AI in myth:
    - McCorduck 2004, pp. 4–5
    - Russell & Norvig 2003, p. 939

24. AI in early science fiction.
    - McCorduck 2004, pp. 17–25

25. Formal reasoning:
    - Berlinski, David (2000). *The Advent of the Algorithm*. Harcourt Books. ISBN 0-15-601391-6. OCLC 46890682 (https://www.worldcat.org/oclc/46890682)

26. "Artificial Intelligence." Encyclopedia of Emerging Industries, edited by Lynn M. Pearce, 6th ed., Gale, 2011, pp. 73-80. Gale Virtual Reference Library, http://link.galegroup.com/apps/doc/CX1930200017/GVRL?u=mcc_pv&sid=GVRL&xid=cd5adac2. Accessed 31 Mar. 2018.

27. Russell & Norvig 2009, p. 16.

28. Dartmouth conference
    - McCorduck 2004, pp. 111–136
    - Crevier 1993, pp. 47–49, who writes "the conference is generally recognized as the official birthdate of the new science."
    - Russell & Norvig 2003, p. 17, who call the conference "the birth of artificial intelligence."
    - NRC 1999, pp. 200–201

29. Hegemony of the Dartmouth conference attendees:
    - Russell & Norvig 2003, p. 17, who write "for the next 20 years the field would be dominated by these people and their students."
    - McCorduck 2004, pp. 129–130

30. Russell & Norvig 2003, p. 18.

31. Schaeffer J. (2009) Didn't Samuel Solve That Game?. In: One Jump Ahead. Springer, Boston, MA

32. Samuel, A. L. (July 1959). "Some Studies in Machine Learning Using the Game of Checkers". *IBM Journal of Research and Development*. **3** (3): 210–229. doi:10.1147/rd.33.0210 (https://doi.org/10.1147%2Frd.33.0210)

33. "Golden years" of AI (successful symbolic reasoning programs 1956–1973):
    - McCorduck 2004, pp. 243–252
    - Crevier 1993, pp. 52–107
    - Moravec 1988, p. 9
    - Russell & Norvig 2003 pp. 18–21
    The programs described are Arthur Samuel's checkers program for the IBM 701, Daniel Bobrow's STUDENT, Newell and Simon's Logic Theorist and Terry Winograd's SHRDLU.

34. DARPA pours money into undirected pure research into AI during the 1960s:
    - McCorduck 2004, pp. 131
    - Crevier 1993, pp. 51, 64–65
    - NRC 1999, pp. 204–205

35. AI in England:
    - Howe 1994

36. Lighthill 1973.

37. Expert systems:
   - **ACM 1998**, I.2.1
   - **Russell & Norvig 2003** pp. 22–24
   - **Luger & Stubblefield 2004** pp. 227–331
   - **Nilsson 1998**, chpt. 17.4
   - **McCorduck 2004**, pp. 327–335, 434–435
   - **Crevier 1993**, pp. 145–62, 197–203

38. Formal methods are now preferred ("Victory of the neats"):
   - **Russell & Norvig 2003** pp. 25–26
   - **McCorduck 2004**, pp. 486–487

39. **McCorduck 2004**, pp. 480–483.

40. **Markoff 2011**.

41. "Ask the AI experts: What's driving today's progress in AI?" (https://www.mckinsey.com/business-functions/mckinsey-analytics/our-insights/ask-the-ai-experts-whats-driving-todays-progress-in-ai). *McKinsey & Company*. Retrieved 13 April 2018.

42. Administrator. "Kinect's AI breakthrough explained" (http://www.i-programmer.info/news/105-artificial-intelligence/2176-kinects-ai-breakthrough-explained.html). *i-programmer.info*. Archived (https://web.archive.org/web/2016020103124 2/http://www.i-programmer.info/news/105-artificial-intelligence/2176-kinects-ai-breakthroug h-explained.html) from the original on 1 February 2016.

43. Rowinski, Dan (15 January 2013). "Virtual Personal Assistants & The Future Of Your Smartphone [Infographic]" (http://readwrite.com/2013/01/15/virtual-personal-assistants-the-future-of-your-smartphone-infographic). *ReadWrite*. Archived (https://web.archive.org/web/20151222083034/http://readwrite.com/2013/01/15/virtual-personal-assistants-t he-future-of-your-smartphone-infographic) from the original on 22 December 2015.

44. "Artificial intelligence: Google's AlphaGo beats Go master Lee Se-dol" (http://www.bbc.com/news/technology-35785 75). *BBC News*. 12 March 2016. Archived (https://web.archive.org/web/20160826103910/http://www.bbc.com/news/t echnology-35785875) from the original on 26 August 2016. Retrieved 1 October 2016.

45. "After Win in China, AlphaGo's Designers Explore New AI" (https://www.wired.com/2017/05/win-china-alphagos-desi gners-explore-new-ai/). 27 May 2017. Archived (https://web.archive.org/web/20170602234726/https://www.wired.co m/2017/05/win-china-alphagos-designers-explore-new-ai/) from the original on 2 June 2017.

46. "World's Go Player Ratings" (http://www.goratings.org/). May 2017. Archived (https://web.archive.org/web/20170401 123616/https://www.goratings.org/) from the original on 1 April 2017.

47. "柯洁迎19岁生日 雄踞人类世界排名第一已两年" (http://sports.sina.com.cn/go/2016-08-02/doc-ifxunyya3020238.shtm l) (in Chinese). May 2017. Archived (https://web.archive.org/web/20170811222849/http://sports.sina.com.cn/go/2016 -08-02/doc-ifxunyya3020238.shtml) from the original on 11 August 2017.

48. Clark, Jack (8 December 2015). "Why 2015 Was a Breakthrough Year in Artificial Intelligence" (https://www.bloomber g.com/news/articles/2015-12-08/why-2015-was-a-breakthrough-year-in-artificial-intelligence). *Bloomberg News*. Archived (https://web.archive.org/web/20161123053855/https://www.bloomberg.com/news/articles/2015-12-08/why-2015-was-a-breakthrough-year-in-artificial-intelligence) from the original on 23 November 2016. Retrieved 23 November 2016. "After a half-decade of quiet breakthroughs in artificial intelligence, 2015 has been a landmark year. Computers are smarter and learning faster than ever."

49. **Domingos 2015**, Chapter 5.

50. **Domingos 2015**, Chapter 7.

51. Lindenbaum, M., Markovitch, S., & Rusakov, D. (2004). Selective sampling for nearest neighbor classifiers. Machine learning, 54(2), 125-152.

52. **Domingos 2015**, Chapter 1.

53. Intractability and efficiency and the combinatorial explosion
   - **Russell & Norvig 2003** pp. 9, 21–22

54. **Domingos 2015**, Chapter 2, Chapter 3.

55. Hart, P. E.; Nilsson, N. J.; Raphael, B. (1972). "Correction to "A Formal Basis for the Heuristic Determination of Minimum Cost Paths" ". *SIGART Newsletter* (37): 28–29. doi:10.1145/1056777.1056779 (https://doi.org/10.1145%2F 1056777.1056779).

56. Domingos 2015, Chapter 2, Chapter 4, Chapter 6.

57. "Can neural network computers learn from experience, and if so, could they ever become what we would call 'smart'?" (https://www.scientificamerican.com/article/can-neural-network-comput/) *Scientific American*. 2018. Retrieved 24 March 2018.

58. Domingos 2015, Chapter 6, Chapter 7.

59. Domingos 2015, p. 286.

60. "Single pixel change fools AI programs" (http://www.bbc.com/news/technology-41845878). *BBC News*. 3 November 2017. Retrieved 12 March 2018.

61. "AI Has a Hallucination Problem That's Proving Tough to Fix" (https://www.wired.com/story/ai-has-a-hallucination-problem-thats-proving-tough-to-fix/) *WIRED*. 2018. Retrieved 12 March 2018.

62. Goodfellow, Ian J., Jonathon Shlens, and Christian Szegedy. "Explaining and harnessing adversarial examples." arXiv preprint arXiv:1412.6572 (2014).

63. Matti, D.; Ekenel, H. K.; Thiran, J. P. (2017). "Combining LiDAR space clustering and convolutional neural networks for pedestrian detection". *2017 14th IEEE International Conference on Advanced Video and Signal Based Surveillance (AVSS)*: 1–6. doi:10.1109/AVSS.2017.8078512 (https://doi.org/10.1109%2FAVSS.2017.8078512). ISBN 978-1-5386-2939-0

64. Ferguson, Sarah; Luders, Brandon; Grande, Robert C.; How, Jonathan P. (2015). "Real-Time Predictive Modeling and Robust Avoidance of Pedestrians with Uncertain, Changing Intentions". *Algorithmic Foundations of Robotics XI*. Springer Tracts in Advanced Robotics. Springer, Cham. **107**: 161–177. doi:10.1007/978-3-319-16595-0_10 (https://doi.org/10.1007%2F978-3-319-16595-0_10). ISBN 978-3-319-16594-3

65. "Cultivating Common Sense | DiscoverMagazine.com" (http://discovermagazine.com/2017/april-2017/cultivating-common-sense). *Discover Magazine*. 2017. Retrieved 24 March 2018.

66. Davis, Ernest; Marcus, Gary (24 August 2015). "Commonsense reasoning and commonsense knowledge in artificial intelligence" (https://m.cacm.acm.org/magazines/2015/9/191169-commonsense-reasoning-and-commonsense-knowledge-in-artificial-intelligence/) *Communications of the ACM* **58** (9): 92–103. doi:10.1145/2701413 (https://doi.org/10.1145%2F2701413)

67. Winograd, Terry (January 1972). "Understanding natural language". *Cognitive Psychology*. **3** (1): 1–191. doi:10.1016/0010-0285(72)90002-3 (https://doi.org/10.1016%2F0010-0285%2872%2990002-3)

68. "Don't worry: Autonomous cars aren't coming tomorrow (or next year)" (http://autoweek.com/article/technology/fully-autonomous-vehicles-are-more-decade-down-road). *Autoweek*. 2016. Retrieved 24 March 2018.

69. Knight, Will (2017). "Boston may be famous for bad drivers, but it's the testing ground for a smarter self-driving car" (https://www.technologyreview.com/s/608871/finally-a-driverless-car-with-some-commonsense/). *MIT Technology Review*. Retrieved 27 March 2018.

70. Prakken, Henry (31 August 2017). "On the problem of making autonomous vehicles conform to traffic law". *Artificial Intelligence and Law*. **25** (3): 341–363. doi:10.1007/s10506-017-9210-0 (https://doi.org/10.1007%2Fs10506-017-9210-0).

71. Problem solving, puzzle solving, game playing and deduction:
    - Russell & Norvig 2003 chpt. 3–9,
    - Poole, Mackworth & Goebel 1998 chpt. 2,3,7,9,
    - Luger & Stubblefield 2004 chpt. 3,4,6,8,
    - Nilsson 1998, chpt. 7–12

72. Uncertain reasoning:
    - Russell & Norvig 2003 pp. 452–644,
    - Poole, Mackworth & Goebel 1998 pp. 345–395,
    - Luger & Stubblefield 2004 pp. 333–381,
    - Nilsson 1998, chpt. 19

73. Psychological evidence of sub-symbolic reasoning:
   - Wason & Shapiro (1966) showed that people do poorly on completely abstract problems, but if the problem is restated to allow the use of intuitive social intelligence, performance dramatically improves. (See Wason selection task)
   - Kahneman, Slovic & Tversky (1982) have shown that people are terrible at elementary problems that involve uncertain reasoning. (See list of cognitive biases for several examples).
   - Lakoff & Núñez (2000) have controversially argued that even our skills at mathematics depend on knowledge and skills that come from "the body", i.e. sensorimotor and perceptual skills. (See Where Mathematics Comes From)

74. Knowledge representation
   - ACM 1998, I.2.4,
   - Russell & Norvig 2003, pp. 320–363,
   - Poole, Mackworth & Goebel 1998, pp. 23–46, 69–81, 169–196, 235–277, 281–298, 319–345,
   - Luger & Stubblefield 2004, pp. 227–243,
   - Nilsson 1998, chpt. 18

75. Knowledge engineering
   - Russell & Norvig 2003, pp. 260–266,
   - Poole, Mackworth & Goebel 1998, pp. 199–233,
   - Nilsson 1998, chpt. ≈17.1–17.4

76. Representing categories and relations: Semantic networks, description logics, inheritance (including frames and scripts):
   - Russell & Norvig 2003, pp. 349–354,
   - Poole, Mackworth & Goebel 1998, pp. 174–177,
   - Luger & Stubblefield 2004, pp. 248–258,
   - Nilsson 1998, chpt. 18.3

77. Representing events and time: Situation calculus, event calculus, fluent calculus (including solving the frame problem):
   - Russell & Norvig 2003, pp. 328–341,
   - Poole, Mackworth & Goebel 1998, pp. 281–298,
   - Nilsson 1998, chpt. 18.2

78. Causal calculus
   - Poole, Mackworth & Goebel 1998, pp. 335–337

79. Representing knowledge about knowledge: Belief calculus, modal logics:
   - Russell & Norvig 2003, pp. 341–344,
   - Poole, Mackworth & Goebel 1998, pp. 275–277

80. Sikos, Leslie F. (June 2017). Description Logics in Multimedia Reasoning (https://www.springer.com/us/book/978331 9540658). Cham: Springer. doi:10.1007/978-3-319-54066-5 (https://doi.org/10.1007%2F978-3-319-54066-5) ISBN 978-3-319-54066-5. Archived (https://web.archive.org/web/20170829120912/https://www.springer.com/us/boo k/9783319540658) from the original on 29 August 2017.

81. Ontology:
   - Russell & Norvig 2003, pp. 320–328

82. Smoliar, Stephen W.; Zhang, HongJiang (1994). "Content based video indexing and retrieval". IEEE multimedia. 1.2: 62–72.

83. Neumann, Bernd; Möller, Ralf (January 2008). "On scene interpretation with description logics". Image and Vision Computing. 26 (1): 82–101. doi:10.1016/j.imavis.2007.08.013 (https://doi.org/10.1016%2Fj.imavis.2007.08.013)

84. Kuperman, G. J.; Reichley, R. M.; Bailey, T. C. (1 July 2006). "Using Commercial Knowledge Bases for Clinical Decision Support: Opportunities, Hurdles, and Recommendations". Journal of the American Medical Informatics Association. 13 (4): 369–371. doi:10.1197/jamia.M2055 (https://doi.org/10.1197%2Fjamia.M2055)

85. MCGARRY, KEN (1 December 2005). "A survey of interestingness measures for knowledge discovery". *The Knowledge Engineering Review* **20** (01): 39. doi:10.1017/S0269888905000408 (https://doi.org/10.1017%2FS026988 8905000408).

86. Bertini, M; Del Bimbo, A; Torniai, C (2006). "Automatic annotation and semantic retrieval of video sequences using multimedia ontologies". *MM '06 Proceedings of the 14th ACM international conference on Multimedia*. 14th ACM international conference on Multimedia. Santa Barbara: ACM. pp. 679–682.

87. Qualification problem
    - McCarthy & Hayes 1969
    - Russell & Norvig 2003

    While McCarthy was primarily concerned with issues in the logical representation of actions, Russell & Norvig 2003 apply the term to the more general issue of default reasoning in the vast network of assumptions underlying all our commonsense knowledge.

88. Default reasoning and default logic, non-monotonic logics, circumscription, closed world assumption, abduction (Poole *et al.* places abduction under "default reasoning". Luger *et al.* places this under "uncertain reasoning"):
    - Russell & Norvig 2003, pp. 354–360,
    - Poole, Mackworth & Goebel 1998, pp. 248–256, 323–335,
    - Luger & Stubblefield 2004, pp. 335–363,
    - Nilsson 1998, ~18.3.3

89. Breadth of commonsense knowledge:
    - Russell & Norvig 2003, p. 21,
    - Crevier 1993, pp. 113–114,
    - Moravec 1988, p. 13,
    - Lenat & Guha 1989 (Introduction)

90. Dreyfus & Dreyfus 1986

91. Gladwell 2005.

92. Expert knowledge as embodied intuition:
    - Dreyfus & Dreyfus 1986 (Hubert Dreyfus is a philosopher and critic of AI who was among the first to argue that most useful human knowledge was encoded sub-symbolically. See Dreyfus' critique of AI)
    - Gladwell 2005 (Gladwell's *Blink* is a popular introduction to sub-symbolic reasoning and knowledge.)
    - Hawkins & Blakeslee 2005 (Hawkins argues that sub-symbolic knowledge should be the primary focus of AI research.)

93. Planning:
    - ACM 1998, ~I.2.8,
    - Russell & Norvig 2003, pp. 375–459,
    - Poole, Mackworth & Goebel 1998, pp. 281–316,
    - Luger & Stubblefield 2004, pp. 314–329,
    - Nilsson 1998, chpt. 10.1–2, 22

94. Information value theory:
    - Russell & Norvig 2003, pp. 600–604

95. Classical planning:
    - Russell & Norvig 2003, pp. 375–430,
    - Poole, Mackworth & Goebel 1998, pp. 281–315,
    - Luger & Stubblefield 2004, pp. 314–329,
    - Nilsson 1998, chpt. 10.1–2, 22

96. Planning and acting in non-deterministic domains: conditional planning, execution monitoring, replanning and continuous planning:
    - Russell & Norvig 2003, pp. 430–449

97. Multi-agent planning and emergent behavior:
    - Russell & Norvig 2003, pp. 449–455

98. Alan Turing discussed the centrality of learning as early as 1950, in his classic paper "Computing Machinery and Intelligence". (Turing 1950) In 1956, at the original Dartmouth AI summer conference, Ray Solomonoff wrote a report on unsupervised probabilistic machine learning: "An Inductive Inference Machine". (Solomonoff 1956)

99. This is a form of Tom Mitchell's widely quoted definition of machine learning: "A computer program is set to learn from an experience $E$ with respect to some task $T$ and some performance measure $P$ if its performance on $T$ as measured by $P$ improves with experience $E$."

100. Learning:
   - ACM 1998, I.2.6,
   - Russell & Norvig 2003, pp. 649–788,
   - Poole, Mackworth & Goebel 1998, pp. 397–438,
   - Luger & Stubblefield 2004, pp. 385–542,
   - Nilsson 1998, chpt. 3.3, 10.3, 17.5, 20

101. Reinforcement learning:
   - Russell & Norvig 2003, pp. 763–788
   - Luger & Stubblefield 2004, pp. 442–449

102. Natural language processing:
   - ACM 1998, I.2.7
   - Russell & Norvig 2003, pp. 790–831
   - Poole, Mackworth & Goebel 1998, pp. 91–104
   - Luger & Stubblefield 2004, pp. 591–632

103. "Versatile question answering systems: seeing in synthesis" (https://www.academia.edu/2475776/Versatile_question_answering_systems_seeing_in_synthesis), Archived (https://web.archive.org/web/20160201125047/http://www.academia.edu/2475776/Versatile_question_answering_systems_seeing_in_synthesis) 1 February 2016 at the Wayback Machine., Mittal et al., IJIIDS, 5(2), 119–142, 2011

104. Applications of natural language processing, including information retrieval (i.e. text mining) and machine translation:
   - Russell & Norvig 2003, pp. 840–857,
   - Luger & Stubblefield 2004, pp. 623–630

105. Cambria, Erik; White, Bebo (May 2014). "Jumping NLP Curves: A Review of Natural Language Processing Research [Review Article]". *IEEE Computational Intelligence Magazine*. **9** (2): 48–57. doi:10.1109/MCI.2014.2307227 (https://doi.org/10.1109%2FMCI.2014.2307227).

106. Machine perception:
   - Russell & Norvig 2003, pp. 537–581, 863–898
   - Nilsson 1998, ~chpt. 6

107. Computer vision:
   - ACM 1998, I.2.10
   - Russell & Norvig 2003, pp. 863–898
   - Nilsson 1998, chpt. 6

108. Speech recognition:
   - ACM 1998, ~I.2.7
   - Russell & Norvig 2003, pp. 568–578

109. Object recognition:
   - Russell & Norvig 2003, pp. 885–892

110. Robotics:
   - ACM 1998, I.2.9,
   - Russell & Norvig 2003, pp. 901–942,
   - Poole, Mackworth & Goebel 1998, pp. 443–460

111. Moving and configuration space:
   - Russell & Norvig 2003, pp. 916–932

112. Tecuci 2012.

113. Robotic mapping (localization, etc):
    - Russell & Norvig 2003, pp. 908–915

114. Weng et al. 2001.

115. Lungarella et al. 2003.

116. Asada et al. 2009.

117. Oudeyer 2010.

118. *Kismet*.

119. Thro 1993.

120. Edelson 1991.

121. Tao & Tan 2005.

122. James 1884.

123. Picard 1995.

124. Kleine-Cosack 2006: "The introduction of emotion to computer science was done by Pickard (sic) who created the field of affective computing."

125. Diamond 2003: "Rosalind Picard, a genial MIT professor, is the field's godmother; her 1997 book, *Affective Computing*, triggered an explosion of interest in the emotional side of computers and their users."

126. Emotion and affective computing
    - Minsky 2006

127. Roberts, Jacob (2016). "Thinking Machines: The Search for Artificial Intelligence" (https://www.sciencehistory.org/distillations/magazine/thinking-machines-the-search-for-artificial-intelligence). *Distillations*. **2** (2): 14–23. Retrieved 20 March 2018.

128. Gerald Edelman, Igor Aleksander and others have argued that artificial consciousness is required for strong AI. (Aleksander 1995; Edelman 2007)

129. Artificial brain arguments: AI requires a simulation of the operation of the human brain
    - Russell & Norvig 2003, p. 957
    - Crevier 1993, pp. 271 and 279
    A few of the people who make some form of the argument:
    - Moravec 1988
    - Kurzweil 2005, p. 262
    - Hawkins & Blakeslee 2005
    The most extreme form of this argument (the brain replacement scenario) was put forward by Clark Glymour in the mid-1970s and was touched on by Zenon Pylyshyn and John Searle in 1980.

130. Nils Nilsson writes: "Simply put, there is wide disagreement in the field about what AI is all about" (Nilsson 1983, p. 10).

131. Haugeland 1985, p. 255.

132. Law 1994.

133. Bach 2008.

134. Shapiro, Stuart C. (1992), "Artificial Intelligence", in Stuart C. Shapiro (ed.), *Encyclopedia of Artificial Intelligence*, 2nd edition (New York: John Wiley & Sons): 54–57. 4 December 2016.

135. AI's immediate precursors:
    - McCorduck 2004, pp. 51–107
    - Crevier 1993, pp. 27–32
    - Russell & Norvig 2003, pp. 15, 940
    - Moravec 1988, p. 3

136. Haugeland 1985, pp. 112–117

137. The most dramatic case of sub-symbolic AI being pushed into the background was the devastating critique of perceptrons by Marvin Minsky and Seymour Papert in 1969. See History of AI, AI winter, or Frank Rosenblatt

138. Cognitive simulation, Newell and Simon, AI at CMU (then called Carnegie Tech):
     - McCorduck 2004, pp. 139–179, 245–250, 322–323 (EPAM)
     - Crevier 1993, pp. 145–149
139. Soar (history):
     - McCorduck 2004, pp. 450–451
     - Crevier 1993, pp. 258–263
140. McCarthy and AI research at SAIL and SRI International:
     - McCorduck 2004, pp. 251–259
     - Crevier 1993
141. AI research at Edinburgh and in France, birth of Prolog:
     - Crevier 1993, pp. 193–196
     - Howe 1994
142. AI at MIT under Marvin Minsky in the 1960s :
     - McCorduck 2004, pp. 259–305
     - Crevier 1993, pp. 83–102, 163–176
     - Russell & Norvig 2003 p. 19
143. Cyc:
     - McCorduck 2004, p. 489, who calls it "a determinedly scruffy enterprise"
     - Crevier 1993, pp. 239–243
     - Russell & Norvig 2003 p. 363−365
     - Lenat & Guha 1989
144. Knowledge revolution:
     - McCorduck 2004, pp. 266–276, 298–300, 314, 421
     - Russell & Norvig 2003 pp. 22–23
145. Embodied approaches to AI:
     - McCorduck 2004, pp. 454–462
     - Brooks 1990
     - Moravec 1988
146. Revival of connectionism:
     - Crevier 1993, pp. 214–215
     - Russell & Norvig 2003 p. 25
147. Computational intelligence
     - IEEE Computational Intelligence Society (http://www.ieee-cis.org/) Archived (https://web.archive.org/web/20080509191840/http://wwwieee-cis.org/) 9 May 2008 at the Wayback Machine.
148. Hutter 2012.
149. Langley 2011.
150. Katz 2012.
151. Norvig 2012.
152. The intelligent agent paradigm:
     - Russell & Norvig 2003 pp. 27, 32–58, 968–972
     - Poole, Mackworth & Goebel 1998 pp. 7–21
     - Luger & Stubblefield 2004 pp. 235–240
     - Hutter 2005, pp. 125–126
     The definition used in this article, in terms of goals, actions, perception and environment, is due to Russell & Norvig (2003). Other definitions also include knowledge and learning as additional criteria.
153. Agent architectures, hybrid intelligent systems
     - Russell & Norvig (2003 pp. 27, 932, 970–972)
     - Nilsson (1998, chpt. 25)

154. Hierarchical control system
  - Albus 2002

155. Search algorithms
  - Russell & Norvig 2003 pp. 59–189
  - Poole, Mackworth & Goebel 1998 pp. 113–163
  - Luger & Stubblefield 2004 pp. 79–164, 193–219
  - Nilsson 1998, chpt. 7–12

156. Forward chaining, backward chaining, Horn clauses, and logical deduction as search:
  - Russell & Norvig 2003 pp. 217–225, 280–294
  - Poole, Mackworth & Goebel 1998 pp. ~46–52
  - Luger & Stubblefield 2004 pp. 62–73
  - Nilsson 1998, chpt. 4.2, 7.2

157. State space search and planning:
  - Russell & Norvig 2003 pp. 382–387
  - Poole, Mackworth & Goebel 1998 pp. 298–305
  - Nilsson 1998, chpt. 10.1–2

158. Uninformed searches (breadth first search, depth first search and general state space search):
  - Russell & Norvig 2003 pp. 59–93
  - Poole, Mackworth & Goebel 1998 pp. 113–132
  - Luger & Stubblefield 2004 pp. 79–121
  - Nilsson 1998, chpt. 8

159. Heuristic or informed searches (e.g., greedy best first and A*):
  - Russell & Norvig 2003 pp. 94–109,
  - Poole, Mackworth & Goebel 1998 pp. pp. 132–147,
  - Luger & Stubblefield 2004 pp. 133–150,
  - Nilsson 1998, chpt. 9,
  - Poole & Mackworth 2017, Section 3.6

160. Optimization searches:
  - Russell & Norvig 2003 pp. 110–116,120–129
  - Poole, Mackworth & Goebel 1998 pp. 56–163
  - Luger & Stubblefield 2004 pp. 127–133

161. Artificial life and society based learning:
  - Luger & Stubblefield 2004 pp. 530–541

162. Genetic programming and genetic algorithms
  - Luger & Stubblefield 2004 pp. 509–530,
  - Nilsson 1998, chpt. 4.2,
  - Holland 1975,
  - Koza 1992,
  - Poli, Langdon & McPhee 2008

163. Logic:
  - ACM 1998, ~I.2.3,
  - Russell & Norvig 2003 pp. 194–310,
  - Luger & Stubblefield 2004 pp. 35–77,
  - Nilsson 1998, chpt. 13–16

164. Satplan:
   - Russell & Norvig 2003 pp. 402–407,
   - Poole, Mackworth & Goebel 1998 pp. 300–301,
   - Nilsson 1998, chpt. 21

165. Explanation based learning relevance based learning inductive logic programming case based reasoning
   - Russell & Norvig 2003 pp. 678–710,
   - Poole, Mackworth & Goebel 1998 pp. 414–416,
   - Luger & Stubblefield 2004 pp. ~422–442,
   - Nilsson 1998, chpt. 10.3, 17.5

166. Propositional logic
   - Russell & Norvig 2003 pp. 204–233,
   - Luger & Stubblefield 2004 pp. 45–50
   - Nilsson 1998, chpt. 13

167. First-order logic and features such as equality:
   - ACM 1998, ~I.2.4,
   - Russell & Norvig 2003 pp. 240–310,
   - Poole, Mackworth & Goebel 1998 pp. 268–275,
   - Luger & Stubblefield 2004 pp. 50–62,
   - Nilsson 1998, chpt. 15

168. Fuzzy logic:
   - Russell & Norvig 2003 pp. 526–527

169. "The Belief Calculus and Uncertain Reasoning", Yen-Teh Hsia

170. Stochastic methods for uncertain reasoning:
   - ACM 1998, ~I.2.3,
   - Russell & Norvig 2003 pp. 462–644,
   - Poole, Mackworth & Goebel 1998 pp. 345–395,
   - Luger & Stubblefield 2004 pp. 165–191, 333–381,
   - Nilsson 1998, chpt. 19

171. Bayesian networks
   - Russell & Norvig 2003 pp. 492–523,
   - Poole, Mackworth & Goebel 1998 pp. 361–381,
   - Luger & Stubblefield 2004 pp. ~182–190, ≈363–379,
   - Nilsson 1998, chpt. 19.3–4

172. Bayesian inference algorithm:
   - Russell & Norvig 2003 pp. 504–519,
   - Poole, Mackworth & Goebel 1998 pp. 361–381,
   - Luger & Stubblefield 2004 pp. ~363–379,
   - Nilsson 1998, chpt. 19.4 & 7

173. Domingos 2015, p. 210.

174. Bayesian learning and the expectation-maximization algorithm
   - Russell & Norvig 2003 pp. 712–724,
   - Poole, Mackworth & Goebel 1998 pp. 424–433,
   - Nilsson 1998, chpt. 20

175. Bayesian decision theory and Bayesian decision networks
   - Russell & Norvig 2003 pp. 597–600

176. Stochastic temporal models:
   - Russell & Norvig 2003 pp. 537–581
   Dynamic Bayesian networks
   - Russell & Norvig 2003 pp. 551–557
   Hidden Markov model
   - (Russell & Norvig 2003 pp. 549–551)
   Kalman filters:
   - Russell & Norvig 2003 pp. 551–557
177. Domingos 2015, p. 160.
178. decision theory and decision analysis:
   - Russell & Norvig 2003 pp. 584–597,
   - Poole, Mackworth & Goebel 1998 pp. 381–394
179. Markov decision processes and dynamic decision networks:
   - Russell & Norvig 2003 pp. 613–631
180. Game theory and mechanism design:
   - Russell & Norvig 2003 pp. 631–643
181. Statistical learning methods and classifiers:
   - Russell & Norvig 2003 pp. 712–754,
   - Luger & Stubblefield 2004 pp. 453–541
182. Decision tree:
   - Russell & Norvig 2003 pp. 653–664,
   - Poole, Mackworth & Goebel 1998 pp. 403–408,
   - Luger & Stubblefield 2004 pp. 408–417
183. Domingos 2015, p. 88.
184. Neural networks and connectionism:
   - Russell & Norvig 2003 pp. 736–748,
   - Poole, Mackworth & Goebel 1998 pp. 408–414,
   - Luger & Stubblefield 2004 pp. 453–505,
   - Nilsson 1998, chpt. 3
185. Domingos 2015, p. 187.
186. K-nearest neighbor algorithm:
   - Russell & Norvig 2003 pp. 733–736
187. Domingos 2015, p. 188.
188. kernel methods such as the support vector machine:
   - Russell & Norvig 2003 pp. 749–752
189. Gaussian mixture model:
   - Russell & Norvig 2003 pp. 725–727
190. Domingos 2015, p. 152.
191. Naive Bayes classifier:
   - Russell & Norvig 2003 pp. 718
192. Classifier performance:
   - van der Walt & Bernard 2006
193. Domingos 2015, Chapter 4.
194. "Why Deep Learning Is Suddenly Changing Your Life" (http://fortune.com/ai-artificial-intelligence-deep-machine-learning/). *Fortune*. 2016. Retrieved 12 March 2018.
195. "Google leads in the race to dominate artificial intelligence" (https://www.economist.com/news/business/21732125-tech-giants-are-investing-billions-transformative-technology-google-leads-race). *The Economist*. 2017. Retrieved 12 March 2018.

196. Feedforward neural networks, perceptrons and radial basis networks
   - Russell & Norvig 2003, pp. 739–748, 758
   - Luger & Stubblefield 2004, pp. 458–467

197. Competitive learning, Hebbian coincidence learning, Hopfield networks and attractor networks:
   - Luger & Stubblefield 2004, pp. 474–505

198. Seppo Linnainmaa (1970). The representation of the cumulative rounding error of an algorithm as a Taylor expansion of the local rounding errors. Master's Thesis (in Finnish), Univ. Helsinki, 6–7.

199. Griewank, Andreas (2012). Who Invented the Reverse Mode of Differentiation?. Optimization Stories, Documenta Matematica, Extra Volume ISMP (2012), 389–400.

200. Paul Werbos, "Beyond Regression: New Tools for Prediction and Analysis in the Behavioral Sciences" *PhD thesis, Harvard University*, 1974.

201. Paul Werbos (1982). Applications of advances in nonlinear sensitivity analysis. In System modeling and optimization (pp. 762–770). Springer Berlin Heidelberg. Online (http://werbos.com/Neural/SensitivityIFIPSeptember1981.pdf) Archived (https://web.archive.org/web/20160414055503/http://werbos.com/Neural/SensitivityIFIPSeptember1981.pdf) 14 April 2016 at the Wayback Machine.

202. Backpropagation:
   - Russell & Norvig 2003, pp. 744–748,
   - Luger & Stubblefield 2004, pp. 467–474,
   - Nilsson 1998, chpt. 3.3

203. Hierarchical temporal memory
   - Hawkins & Blakeslee 2005

204. "Artificial intelligence can 'evolve' to solve problems" (http://www.sciencemag.org/news/2018/01/artificial-intelligence-can-evolve-solve-problems). *Science | AAAS*. 10 January 2018. Retrieved 7 February 2018.

205. Schmidhuber, J. (2015). "Deep Learning in Neural Networks: An Overview". *Neural Networks*. **61**: 85–117. arXiv:1404.7828 (https://arxiv.org/abs/1404.7828). doi:10.1016/j.neunet.2014.09.003 (https://doi.org/10.1016%2Fj.neunet.2014.09.003). PMID 25462637 (https://www.ncbi.nlm.nih.gov/pubmed/25462637).

206. Ian Goodfellow, Yoshua Bengio, and Aaron Courville (2016). Deep Learning. MIT Press. Online (http://www.deeplearningbook.org) Archived (https://web.archive.org/web/20160416111010/http://www.deeplearningbook.org/) 16 April 2016 at the Wayback Machine.

207. Hinton, G.; Deng, L.; Yu, D.; Dahl, G.; Mohamed, A.; Jaitly, N.; Senior, A.; Vanhoucke, V.; Nguyen, P.; Sainath, T.; Kingsbury, B. (2012). "Deep Neural Networks for Acoustic Modeling in Speech Recognition --- The shared views of four research groups". *IEEE Signal Processing Magazine*. **29** (6): 82–97. doi:10.1109/msp.2012.2205597 (https://doi.org/10.1109%2Fmsp.2012.2205597).

208. Schmidhuber, Jürgen (2015). "Deep Learning" (http://www.scholarpedia.org/article/Deep_Learning). *Scholarpedia*. **10** (11): 32832. doi:10.4249/scholarpedia.32832 (https://doi.org/10.4249%2Fscholarpedia.32832). Archived (https://web.archive.org/web/20160419024349/http://www.scholarpedia.org/article/Deep_Learning) from the original on 19 April 2016.

209. Rina Dechter (1986). Learning while searching in constraint-satisfaction problems. University of California, Computer Science Department, Cognitive Systems Laboratory. Online (https://www.researchgate.net/publication/221605378_Learning_While_Searching_in_Constraint-Satisfaction-Problems) Archived (https://web.archive.org/web/20160419054654/https://www.researchgate.net/publication/221605378_Learning_While_Searching_in_Constraint-Satisfaction-Problems) 19 April 2016 at the Wayback Machine.

210. Igor Aizenberg, Naum N. Aizenberg, Joos P.L. Vandewalle (2000). Multi-Valued and Universal Binary Neurons: Theory, Learning and Applications. Springer Science & Business Media.

211. Ivakhnenko, Alexey (1965). *Cybernetic Predicting Devices*. Kiev: Naukova Dumka.

212. Ivakhnenko, Alexey (1971). "Polynomial theory of complex systems". *IEEE Transactions on Systems, Man and Cybernetics (4)*: 364–378.

213. Hinton 2007.

214. Research, AI (23 October 2015). "Deep Neural Networks for Acoustic Modeling in Speech Recognition" (http://airesearch.com/ai-research-papers/deep-neural-networks-for-acoustic-modeling-in-speech-recognition/). *airesearch.com*. Retrieved 23 October 2015.

215. Fukushima, K. (1980). "Neocognitron: A self-organizing neural network model for a mechanism of pattern recognition unaffected by shift in position". *Biological Cybernetics* **36** (4): 193–202. doi:10.1007/bf00344251 (https://doi.org/10.1007%2Fbf00344251) PMID 7370364 (https://www.ncbi.nlm.nih.gov/pubmed/7370364).

216. Yann LeCun (2016). Slides on Deep Learning Online (https://indico.cern.ch/event/510372/) Archived (https://web.archive.org/web/20160423021403/https://indico.cern.ch/event/510372/) 23 April 2016 at the Wayback Machine

217. Silver, David; Schrittwieser, Julian; Simonyan, Karen; Antonoglou, Ioannis; Huang, Aja; Guez, Arthur; Hubert, Thomas; Baker, Lucas; Lai, Matthew; Bolton, Adrian; Chen, Yutian; Lillicrap, Timothy; Fan, Hui; Sifre, Laurent; Driessche, George van den; Graepel, Thore; Hassabis, Demis (19 October 2017). "Mastering the game of Go without human knowledge" (https://www.nature.com/nature/journal/v550/n7676/full/nature24270.html) *Nature*. **550** (7676): 354–359. doi:10.1038/nature24270 (https://doi.org/10.1038%2Fnature24270) ISSN 0028-0836 (https://www.worldcat.org/issn/0028-0836) "AlphaGo Lee... 12 convolutional layers".

218. Recurrent neural networks, Hopfield nets:
    - Russell & Norvig 2003, p. 758
    - Luger & Stubblefield 2004, pp. 474–505

219. Hyötyniemi, Heikki (1996). "Turing machines are recurrent neural networks". *Proceedings of STeP '96/Publications of the Finnish Artificial Intelligence Society*: 13–24.

220. P. J. Werbos. Generalization of backpropagation with application to a recurrent gas market model" *Neural Networks* 1, 1988.

221. A. J. Robinson and F. Fallside. The utility driven dynamic error propagation network. Technical Report CUED/F-INFENG/TR.1, Cambridge University Engineering Department, 1987.

222. R. J. Williams and D. Zipser. Gradient-based learning algorithms for recurrent networks and their computational complexity. In Back-propagation: Theory, Architectures and Applications. Hillsdale, NJ: Erlbaum, 1994.

223. Sepp Hochreiter (1991), Untersuchungen zu dynamischen neuronalen Netzen (http://people.idsia.ch/~juergen/SeppHochreiter1991ThesisAdvisorSchmidhuber.pdf) Archived (https://web.archive.org/web/20150306075401/http://people.idsia.ch/~juergen/SeppHochreiter1991ThesisAdvisorSchmidhuber.pdf) 6 March 2015 at the Wayback Machine, Diploma thesis. Institut f. Informatik, Technische Univ. Munich. Advisor: J. Schmidhuber

224. Schmidhuber, J. (1992). "Learning complex, extended sequences using the principle of history compression" *Neural Computation*. **4** (2): 234–242. CiteSeerX 10.1.1.49.3934 (https://citeseerx.ist.psu.edu/viewdoc/summary?doi=10.1.1.49.3934). doi:10.1162/neco.1992.4.2.234 (https://doi.org/10.1162%2Fneco.1992.4.2.234)

225. Hochreiter, Sepp; and Schmidhuber, Jürgen; *Long Short-Term Memory*, Neural Computation, 9(8):1735–1780, 1997

226. Alex Graves, Santiago Fernandez, Faustino Gomez, and Jürgen Schmidhuber (2006). Connectionist temporal classification: Labelling unsegmented sequence data with recurrent neural nets. Proceedings of ICML 06, pp. 369–376.

227. Hannun, Awni; Case, Carl; Casper, Jared; Catanzaro, Bryan; Diamos, Greg; Elsen, Erich; Prenger, Ryan; Satheesh, Sanjeev; Sengupta, Shubho; Coates, Adam; Ng, Andrew Y. (2014). "Deep Speech: Scaling up end-to-end speech recognition". arXiv:1412.5567 (https://arxiv.org/abs/1412.5567) [cs.CL (https://arxiv.org/archive/cs.CL)].

228. Hasim Sak and Andrew Senior and Francoise Beaufays (2014). Long Short-Term Memory recurrent neural network architectures for large scale acoustic modeling. Proceedings of Interspeech 2014.

229. Li, Xiangang; Wu, Xihong (2015). "Constructing Long Short-Term Memory based Deep Recurrent Neural Networks for Large Vocabulary Speech Recognition". arXiv:1410.4281 (https://arxiv.org/abs/1410.4281) [cs.CL (https://arxiv.org/archive/cs.CL)].

230. Haşim Sak, Andrew Senior, Kanishka Rao, Françoise Beaufays and Johan Schalkwyk (September 2015): Google voice search: faster and more accurate. (http://googleresearch.blogspot.ch/2015/09/google-voice-search-faster-and-more.html) Archived (https://web.archive.org/web/20160309191532/http://googleresearch.blogspot.ch/2015/09/google-voice-search-faster-and-more.html) 9 March 2016 at the Wayback Machine.

231. Sutskever, Ilya; Vinyals, Oriol; Le, Quoc V (2014). "Sequence to Sequence Learning with Neural Networks". arXiv:1409.3215 (https://arxiv.org/abs/1409.3215) [cs.CL (https://arxiv.org/archive/cs.CL)].

232. Jozefowicz, Rafal; Vinyals, Oriol; Schuster, Mike; Shazeer, Noam; Wu, Yonghui (2016). "Exploring the Limits of Language Modeling". arXiv:1602.02410 (https://arxiv.org/abs/1602.02410) [cs.CL (https://arxiv.org/archive/cs.CL)].

233. Gillick, Dan; Brunk, Clif; Vinyals, Oriol; Subramanya, Amarnag (2015). "Multilingual Language Processing From Bytes". arXiv:1512.00103 (https://arxiv.org/abs/1512.00103) [cs.CL (https://arxiv.org/archive/cs.CL)].

234. Vinyals, Oriol; Toshev, Alexander; Bengio, Samy; Erhan, Dumitru (2015). "Show and Tell: A Neural Image Caption Generator". arXiv:1411.4555 (https://arxiv.org/abs/1411.4555) [cs.CV (https://arxiv.org/archive/cs.CV)].

235. Lisp:
    - Luger & Stubblefield 2004, pp. 723–821
    - Crevier 1993, pp. 59–62,
    - Russell & Norvig 2003, p. 18

236. Prolog:
    - Poole, Mackworth & Goebel 1998, pp. 477–491,
    - Luger & Stubblefield 2004, pp. 641–676, 575–581

237. "C++ Java" (https://www.infoworld.com/article/3186599/artificial-intelligence/the-5-best-programming-languages-for-ai-development.html). infoworld.com. Retrieved 6 December 2017.

238. Ferris, Robert (7 April 2016). "How Steve Jobs' friend changed the world of math" (https://www.cnbc.com/2016/04/07/stephen-wolfram-why-this-brilliant-physicist-ditched-his-job.html). CNBC. Retrieved 28 February 2018.

239. The Turing test:
    Turing's original publication:
    - Turing 1950
    Historical influence and philosophical implications:
    - Haugeland 1985, pp. 6–9
    - Crevier 1993, p. 24
    - McCorduck 2004, pp. 70–71
    - Russell & Norvig 2003, pp. 2–3 and 948

240. Mathematical definitions of intelligence:
    - Hernandez-Orallo 2000
    - Dowe & Hajek 1997
    - Hernandez-Orallo & Dowe 2010

241. O'Brien & Marakas 2011

242. Russell & Norvig 2009, p. 1.

243. CNN 2006.

244. N. Aletras; D. Tsarapatsanis; D. Preotiuc-Pietro; V. Lampos (2016). "Predicting judicial decisions of the European Court of Human Rights: a Natural Language Processing perspective" (https://peerj.com/articles/cs-93/). PeerJ Computer Science. Archived (https://web.archive.org/web/20161029084624/https://peerj.com/articles/cs-93/) from the original on 29 October 2016.

245. "The Economist Explains: Why firms are piling into artificial intelligence" (https://www.economist.com/blogs/economist-explains/2016/04/economist-explains). The Economist. 31 March 2016. Archived (https://web.archive.org/web/20160508010311/http://www.economist.com/blogs/economist-explains/2016/04/economist-explains) from the original on 8 May 2016. Retrieved 19 May 2016.

246. Lohr, Steve (28 February 2016). "The Promise of Artificial Intelligence Unfolds in Small Steps" (https://www.nytimes.com/2016/02/29/technology/the-promise-of-artificial-intelligence-unfolds-in-small-steps.html?ref=technology). The New York Times. Archived (https://web.archive.org/web/20160229171843/http://www.nytimes.com/2016/02/29/technology/the-promise-of-artificial-intelligence-unfolds-in-small-steps.html?ref=technology) from the original on 29 February 2016. Retrieved 29 February 2016.

247. Wakefield, Jane (15 June 2016). "Social media 'outstrips TV' as news source for young people" (http://www.bbc.co.uk/news/uk-36528256) BBC News. Archived (https://web.archive.org/web/20160624000744/http://www.bbc.co.uk/news/uk-36528256) from the original on 24 June 2016.

248. Smith, Mark (22 July 2016). "So you think you chose to read this article?" (http://www.bbc.co.uk/news/business-36837824). BBC News. Archived (https://web.archive.org/web/20160725205007/http://www.bbc.co.uk/news/business-36837824) from the original on 25 July 2016.

249. Dina Bass (20 September 2016). "Microsoft Develops AI to Help Cancer Doctors Find the Right Treatments" (https://www.bloomberg.com/news/articles/2016-09-20/microsoft-develops-ai-to-help-cancer-doctors-find-the-right-treatments). Bloomberg. Archived (https://web.archive.org/web/20170511103625/https://www.bloomberg.com/news/articles/2016-09-20/microsoft-develops-ai-to-help-cancer-doctors-find-the-right-treatments) from the original on 11 May 2017.

250. Gallagher, James (26 January 2017). "Artificial intelligence 'as good as cancer doctors'" (http://www.bbc.co.uk/news/health-38717928). *BBC News*. Archived (https://web.archive.org/web/20170126133849/http://www.bbc.co.uk/news/health-38717928) from the original on 26 January 2017. Retrieved 26 January 2017.

251. Langen, Pauline A.; Katz, Jeffrey S.; Dempsey, Gayle, eds. (18 October 1994), *Remote monitoring of high-risk patients using artificial intelligence* (http://www.google.com/patents/US5357427) (US5357427 A), archived (https://web.archive.org/web/20170228090520/http://www.google.com/patents/US5357427) from the original on 28 February 2017, retrieved 27 February 2017

252. Senthilingam, Meera (12 May 2016). "Are Autonomous Robots Your next Surgeons?" (http://www.cnn.com/2016/05/12/health/robot-surgeon-bowel-operation/). *CNN*. Cable News Network. Archived (https://web.archive.org/web/20161203154119/http://www.cnn.com/2016/05/12/health/robot-surgeon-bowel-operation) from the original on 3 December 2016. Retrieved 4 December 2016.

253. Markoff, John (16 February 2011). "On 'Jeopardy!' Watson Win Is All but Trivial" (https://www.nytimes.com/2011/02/17/science/17jeopardy-watson.html?pagewanted=all&mcubz=3). *The New York Times*. Archived (https://web.archive.org/web/20170922050941/http://www.nytimes.com/2011/02/17/science/17jeopardy-watson.html?pagewanted=all&mcubz=3) from the original on 22 September 2017.

254. Ng, Alfred (7 August 2016). "IBM's Watson gives proper diagnosis after doctors were stumped" (http://www.nydailynews.com/news/world/ibm-watson-proper-diagnosis-doctors-stumped-article-1.2741857). *NY Daily News*. Archived (https://web.archive.org/web/20170922101344/http://www.nydailynews.com/news/world/ibm-watson-proper-diagnosis-doctors-stumped-article-1.2741857) from the original on 22 September 2017.

255. "33 Corporations Working On Autonomous Vehicles". CB Insights. N.p., 11 August 2016. 12 November 2016.

256. West, Darrell M. "Moving forward: Self-driving vehicles in China, Europe, Japan, Korea, and the United States". Center for Technology Innovation at Brookings. N.p., September 2016. 12 November 2016.

257. Burgess, Matt. "The UK is about to Start Testing Self-Driving Truck Platoons" (https://www.wired.co.uk/article/uk-trial-self-driving-trucks-platoons-roads). *WIRED*. Archived (https://web.archive.org/web/20170922055917/http://www.wired.co.uk/article/uk-trial-self-driving-trucks-platoons-roads) from the original on 22 September 2017. Retrieved 20 September 2017.

258. Davies, Alex. "World's First Self-Driving Semi-Truck Hits the Road" (https://www.wired.com/2015/05/worlds-first-self-driving-semi-truck-hits-road/). *WIRED*. Archived (https://web.archive.org/web/20171028222802/https://www.wired.com/2015/05/worlds-first-self-driving-semi-truck-hits-road/) from the original on 28 October 2017. Retrieved 20 September 2017.

259. McFarland, Matt. "Google's artificial intelligence breakthrough may have a huge impact on self-driving cars and much more". *The Washington Post* 25 February 2015. Infotrac Newsstand. 24 October 2016

260. "Programming safety into self-driving cars". National Science Foundation. N.p., 2 February 2015. 24 October 2016.

261. ArXiv, E. T. (26 October 2015). Why Self-Driving Cars Must Be Programmed to Kill. Retrieved 17 November 2017, from https://www.technologyreview.com/s/542626/why-self-driving-cars-must-be-programmed-to-kill/

262. O'Neill,, Eleanor (31 July 2016). "Accounting, automation and AI" (https://www.icas.com/ca-today-news/how-accountancy-and-finance-are-using-artificial-intelligence). *www.icas.com*. Archived (https://web.archive.org/web/20161118165901/https://www.icas.com/ca-today-news/how-accountancy-and-finance-are-using-artificial-intelligence) from the original on 18 November 2016. Retrieved 18 November 2016.

263. Robots Beat Humans in Trading Battle. (http://news.bbc.co.uk/2/hi/business/1481339.stm) Archived (https://web.archive.org/web/20090909001249/http://news.bbc.co.uk/2/hi/business/1481339.stm) 9 September 2009 at the Wayback Machine. BBC.com (8 August 2001)

264. "CTO Corner: Artificial Intelligence Use in Financial Services – Financial Services Roundtable" (http://fsroundtable.org/cto-corner-artificial-intelligence-use-in-financial-services/). *Financial Services Roundtable*. 2 April 2015. Archived (https://web.archive.org/web/20161118165842/http://fsroundtable.org/cto-corner-artificial-intelligence-use-in-financial-services/) from the original on 18 November 2016. Retrieved 18 November 2016.

265. Marwala, Tshilidzi; Hurwitz, Evan (2017). *Artificial Intelligence and Economic Theory: Skynet in the Market*. London: Springer. ISBN 978-3-319-66104-9.

266. "Why AI researchers like video games" (https://www.economist.com/news/science-and-technology/21721890-games-help-them-understand-reality-why-ai-researchers-video-games). *The Economist*. Archived (https://web.archive.org/web/20171005051028/https://www.economist.com/news/science-and-technology/21721890-games-help-them-understand-reality-why-ai-researchers-video-games) from the original on 5 October 2017.

267. Yannakakis, G. N. (2012, May). Game AI revisited. In Proceedings of the 9th conference on Computing Frontiers (pp. 285–292). ACM.

268. "Getting to grips with military robotics" (https://www.economist.com/news/special-report/21735478-autonomous-robots-and-swarms-will-change-nature-warfare-getting-grips) *The Economist*. 25 January 2018. Retrieved 7 February 2018.

269. "Autonomous Systems: Infographic" (https://www.siemens.com/innovation/en/home/pictures-of-the-future/digitalization-and-software/autonomous-systems-infographic.html) *www.siemens.com*. Retrieved 7 February 2018.

270. "Artificial Intelligence Fuels New Global Arms Race" (https://www.wired.com/story/for-superpowers-artificial-intelligence-fuels-new-global-arms-race/) *WIRED*. Retrieved 24 December 2017.

271. Clifford, Catherine (29 September 2017). "In the same way there was a nuclear arms race, there will be a race to build A.I., says tech exec" (https://www.cnbc.com/2017/09/28/hootsuiteceo-next-version-of-arms-race-will-be-a-race-to-build-ai.html). *CNBC*. Retrieved 24 December 2017.

272. Metz, Cade (15 March 2018). "Pentagon Wants Silicon Valley's Help on A.I." (https://www.nytimes.com/2018/03/15/technology/military-artificial-intelligence.html) *The New York Times*. Retrieved 19 March 2018.

273. Brooks 1991.

274. "Hacking Roomba" (http://hackingroomba.com/) *hackingroomba.com*. Archived (https://web.archive.org/web/20091018023300/http://hackingroomba.com/) from the original on 18 October 2009.

275. "A self-organizing thousand-robot swarm" (https://www.seas.harvard.edu/news/2014/08/self-organizing-thousand-robot-swarm). *www.seas.harvard.edu*. 14 August 2014. Archived (https://web.archive.org/web/20170504100513/http://www.seas.harvard.edu/news/2014/08/self-organizing-thousand-robot-swarm) from the original on 4 May 2017.

276. "Watch An Autonomous Robot Swarm Form 2D Starfishes" (https://creators.vice.com/en_us/article/watch-an-autonomous-robot-swarm-form-2d-starfishes) *Creators*.

277. Rainie, Lee; Janna; erson (8 February 2017). "Theme 2: Good things lie ahead" (http://www.pewinternet.org/2017/02/08/theme-2-good-things-lie-ahead/) Archived (https://web.archive.org/web/20170703135509/http://www.pewinternet.org/2017/02/08/theme-2-good-things-lie-ahead/) from the original on 3 July 2017.

278. Lynley, Matthew. "SoundHound raises $75M to bring its voice-enabled AI everywhere" (https://techcrunch.com/2017/01/31/soundhound-raises-75m-to-bring-its-voice-enabled-ai-everywhere/) Archived (https://web.archive.org/web/20170913141634/https://techcrunch.com/2017/01/31/soundhound-raises-75m-to-bring-its-voice-enabled-ai-everywhere/) from the original on 13 September 2017.

279. Manyika, James; Chui, Michael; Bughin, Jaques; Brown, Brad; Dobbs, Richard; Roxburgh, Charles; Byers, Angela Hung (May 2011). "Big Data: The next frontier for innovation, competition, and productivity" (http://www.mckinsey.com/Insights/MGI/Research/Technology_and_Innovation/Big_data_The_next_frontier_for_innovation) *McKinsey Global Institute*. Archived (https://web.archive.org/web/20130306232114/http://www.mckinsey.com/insights/mgi/research/technology_and_innovation/big_data_the_next_frontier_for_innovation) from the original on 6 March 2013. Retrieved 16 January 2016.

280. "NY gets new boot camp for data scientists: It's free but harder to get into than Harvard" (https://venturebeat.com/2014/04/15/ny-gets-new-bootcamp-for-data-scientists-its-free-but-harder-to-get-into-than-harvard/) *Venture Beat*. Archived (https://web.archive.org/web/20160215235820/http://venturebeat.com/2014/04/15/ny-gets-new-bootcamp-for-data-scientists-its-free-but-harder-to-get-into-than-harvard/) from the original on 15 February 2016. Retrieved 21 February 2016.

281. "Partnership on Artificial Intelligence to Benefit People and Society". N.p., n.d. 24 October 2016.

282. Fiegerman, Seth. "Facebook, Google, Amazon Create Group to Ease AI Concerns". CNNMoney. n.d. 4 December 2016.

283. Dartmouth proposal:
   - McCarthy et al. 1955 (the original proposal)
   - Crevier 1993, p. 49 (historical significance)

284. The physical symbol systems hypothesis:
   - Newell & Simon 1976, p. 116
   - McCorduck 2004, p. 153
   - Russell & Norvig 2003, p. 18

285. Dreyfus criticized the necessary condition of the physical symbol system hypothesis, which he called the "psychological assumption": "The mind can be viewed as a device operating on bits of information according to formal rules." (Dreyfus 1992, p. 156)

286. Dreyfus' critique of artificial intelligence
    - Dreyfus 1972, Dreyfus & Dreyfus 1986
    - Crevier 1993, pp. 120–132
    - McCorduck 2004, pp. 211–239
    - Russell & Norvig 2003 pp. 950–952,

287. Gödel 1951: in this lecture, Kurt Gödel uses the incompleteness theorem to arrive at the following disjunction: (a) the human mind is not a consistent finite machine, or (b) there exist Diophantine equations for which it cannot decide whether solutions exist. Gödel finds (b) implausible, and thus seems to have believed the human mind was not equivalent to a finite machine, i.e., its power exceeded that of any finite machine. He recognized that this was only a conjecture, since one could never disprove (b). Yet he considered the disjunctive conclusion to be a "certain fact".

288. The Mathematical Objection:
    - Russell & Norvig 2003 p. 949
    - McCorduck 2004, pp. 448–449

    Making the Mathematical Objection:
    - Lucas 1961
    - Penrose 1989

    Refuting Mathematical Objection:
    - Turing 1950 under "(2) The Mathematical Objection"
    - Hofstadter 1979

    Background:
    - Gödel 1931, Church 1936, Kleene 1935, Turing 1937

289. Graham Oppy (20 January 2015). "Gödel's Incompleteness Theorems" (http://plato.stanford.edu/entries/goedel-incompleteness/#GdeArgAgaMec). *Stanford Encyclopedia of Philosophy*. Retrieved 27 April 2016. "These Gödelian anti-mechanist arguments are, however, problematic, and there is wide consensus that they fail.".

290. Stuart J. Russell, Peter Norvig (2010). "26.1.2: Philosophical Foundations/Weak AI: Can Machines Act Intelligently?/The mathematical objection". *Artificial Intelligence: A Modern Approach* (3rd ed.). Upper Saddle River, NJ: Prentice Hall. ISBN 0-13-604259-7. "...even if we grant that computers have limitations on what they can prove, there is no evidence that humans are immune from those limitations.".

291. Mark Colyvan. An introduction to the philosophy of mathematics. Cambridge University Press, 2012. From 2.2.2, 'Philosophical significance of Gödel's incompleteness results': "The accepted wisdom (with which I concur) is that the Lucas-Penrose arguments fail."

292. Russel, Stuart., Daniel Dewey, and Max Tegmark. Research Priorities for Robust and Beneficial Artificial Intelligence. AI Magazine 36:4 (2015). 8 December 2016.

293. "Stephen Hawking warns artificial intelligence could end mankind" (http://www.bbc.com/news/technology-30290540). *BBC News*. Archived (https://web.archive.org/web/20151030054329/http://www.bbc.com/news/technology-30290540) from the original on 30 October 2015. Retrieved 30 October 2015.

294. Holley, Peter (28 January 2015). "Bill Gates on dangers of artificial intelligence: 'I don't understand why some people are not concerned'" (https://www.washingtonpost.com/news/theswitch/wp/2015/01/28/bill-gates-on-dangers-of-artificial-intelligence-dont-understand-why-some-people-are-not-concerned/). *The Washington Post*. ISSN 0190-8286 (https://www.worldcat.org/issn/0190-8286). Archived (https://web.archive.org/web/20151030054330/https://www.washingtonpost.com/news/the-switch/wp/2015/01/28/bill-gates-on-dangers-of-artificial-intelligence-dont-understand-why-some-people-are-not-concerned/) from the original on 30 October 2015. Retrieved 30 October 2015.

295. Gibbs, Samuel. "Elon Musk: artificial intelligence is our biggest existential threat" (https://www.theguardian.com/technology/2014/oct/27/elon-musk-artificial-intelligence-ai-biggest-existential-threat). *The Guardian*. Archived (https://web.archive.org/web/20151030054330/http://www.theguardian.com/technology/2014/oct/27/elon-musk-artificial-intelligence-ai-biggest-existential-threat) from the original on 30 October 2015. Retrieved 30 October 2015.

296. Post, Washington. "Tech titans like Elon Musk are spending $1 billion to save you from terminators" (http://www.chicagotribune.com/bluesky/technology/ct-tech-titans-against-terminators-20151214-story.html). Archived (https://web.archive.org/web/20160607121118/http://www.chicagotribune.com/bluesky/technology/ct-tech-titans-against-terminators-20151214-story.html) from the original on 7 June 2016.

297. Müller, Vincent C.; Bostrom, Nick (2014). "Future Progress in Artificial Intelligence: A Poll Among Experts" (http://www.sophia.de/pdf/2014_PT-AI_polls.pdf) (PDF). *AI Matters*. **1** (1): 9–11. doi:10.1145/2639475.2639478 (https://doi.org/10.1145%2F2639475.2639478) Archived (https://web.archive.org/web/20160115114604/http://www.sophia.de/pdf/2014_PT-AI_polls.pdf) (PDF) from the original on 15 January 2016.

298. "Is artificial intelligence really an existential threat to humanity?" (http://thebulletin.org/artificial-intelligence-really-existential-threat-humanity8577) *Bulletin of the Atomic Scientists*. Archived (https://web.archive.org/web/20151030054330/http://thebulletin.org/artificial-intelligence-really-existential-threat-humanity8577) from the original on 30 October 2015. Retrieved 30 October 2015.

299. "The case against killer robots, from a guy actually working on artificial intelligence" (http://fusion.net/story/54583/the-case-against-killer-robots-from-a-guy-actually-building-ai/) *Fusion.net*. Archived (https://web.archive.org/web/20160204175716/http://fusion.net/story/54583/the-case-against-killer-robots-from-a-guy-actually-building-ai/) from the original on 4 February 2016 Retrieved 31 January 2016.

300. "Will artificial intelligence destroy humanity? Here are 5 reasons not to worry" (https://www.vox.com/2014/8/22/6043635/5-reasons-we-shouldnt-worry-about-super-intelligent-computers-taking) *Vox*. Archived (https://web.archive.org/web/20151030092203/http://www.vox.com/2014/8/22/6043635/5-reasons-we-shouldnt-worry-about-super-intelligent-computers-taking) from the original on 30 October 2015 Retrieved 30 October 2015.

301. "The mysterious artificial intelligence company Elon Musk invested in is developing game-changing smart computers" (http://www.techinsider.io/mysterious-artificial-intelligence-company-elon-musk-investment-2015-10). *Tech Insider*. Archived (https://web.archive.org/web/20151030165333/http://www.techinsider.io/mysterious-artificial-intelligence-company-elon-musk-investment-2015-10) from the original on 30 October 2015 Retrieved 30 October 2015.

302. Clark, Jack. "Musk-Backed Group Probes Risks Behind Artificial Intelligence" (https://www.bloomberg.com/news/articles/2015-07-01/musk-backed-group-probes-risks-behind-artificial-intelligence) *Bloomberg.com*. Archived (https://web.archive.org/web/20151030202356/http://www.bloomberg.com/news/articles/2015-07-01/musk-backed-group-probes-risks-behind-artificial-intelligence) from the original on 30 October 2015 Retrieved 30 October 2015.

303. "Elon Musk Is Donating $10M Of His Own Money To Artificial Intelligence Research" (http://www.fastcompany.com/3041007/fast-feed/elon-musk-is-donating-10m-of-his-own-money-to-artificial-intelligence-research) *Fast Company*. Archived (https://web.archive.org/web/20151030202356/http://www.fastcompany.com/3041007/fast-feed/elon-musk-is-donating-10m-of-his-own-money-to-artificial-intelligence-research) from the original on 30 October 2015 Retrieved 30 October 2015.

304. "Stephen Hawking, Elon Musk, and Bill Gates Warn About Artificial Intelligence" (http://observer.com/2015/08/stephen-hawking-elon-musk-and-bill-gates-warn-about-artificial-intelligence/) *Observer*. Archived (https://web.archive.org/web/20151030053323/http://observer.com/2015/08/stephen-hawking-elon-musk-and-bill-gates-warn-about-artificial-intelligence/) from the original on 30 October 2015 Retrieved 30 October 2015.

305. In the early 1970s, Kenneth Colby presented a version of Weizenbaum's ELIZA known as DOCTOR which he promoted as a serious therapeutic tool. (Crevier 1993, pp. 132–144)

306. Joseph Weizenbaum's critique of AI:
   - Weizenbaum 1976
   - Crevier 1993, pp. 132–144
   - McCorduck 2004, pp. 356–373
   - Russell & Norvig 2003, p. 961

   Weizenbaum (the AI researcher who developed the first chatterbot program, ELIZA) argued in 1976 that the misuse of artificial intelligence has the potential to devalue human life.

307. E McGaughey, 'Will Robots Automate Your Job Away? Full Employment, Basic Income, and Economic Democracy' (2018) SSRN, part 2(3) (https://papers.ssrn.com/sol3/papers.cfm?abstract_id=3044448)

308. "Automation and anxiety" (https://www.economist.com/news/special-report/21700758-will-smarter-machines-cause-mass-unemployment-automation-and-anxiety) *The Economist*. 9 May 2015. Retrieved 13 January 2018.

309. Lohr, Steve (2017). "Robots Will Take Jobs, but Not as Fast as Some Fear, New Report Says" (https://www.nytimes.com/2017/01/12/technology/robots-will-take-jobs-but-not-as-fast-as-some-fear-new-report-says.html) *The New York Times*. Retrieved 13 January 2018.

310. Frey, Carl Benedikt; Osborne, Michael A (1 January 2017). "The future of employment: How susceptible are jobs to computerisation?" (https://www.sciencedirect.com/science/article/pii/S0040162516302244) *Technological Forecasting and Social Change* **114**: 254–280. doi:10.1016/j.techfore.2016.08.019 (https://doi.org/10.1016%2Fj.techfore.2016.08.019). ISSN 0040-1625 (https://www.worldcat.org/issn/0040-1625)

311. Arntz, Melanie, Terry Gregory, and Ulrich Zierahn. "The risk of automation for jobs in OECD countries: A comparative analysis." OECD Social, Employment, and Migration Working Papers 189 (2016). p. 33.

312. Mahdawi, Arwa (26 June 2017). "What jobs will still be around in 20 years? Read this to prepare your future" (https://www.theguardian.com/us-news/2017/jun/26/jobs-future-automation-robots-skills-creative-health). *The Guardian*. Retrieved 13 January 2018.

313. Wendell Wallach (2010). *Moral Machines*, Oxford University Press.

314. Wallach, pp 37–54.

315. Wallach, pp 55–73.

316. Wallach, Introduction chapter

317. Michael Anderson and Susan Leigh Anderson (2011), Machine Ethics, Cambridge University Press.

318. "Machine Ethics" (https://web.archive.org/web/20141129044821/http://www.aaai.org/Library/Symposia/Fall/fs05-06). *aaai.org*. Archived from the original (http://www.aaai.org/Library/Symposia/Fall/fs05-06) on 29 November 2014.

319. Rubin, Charles (Spring 2003). "Artificial Intelligence and Human Nature | The New Atlantis" (https://web.archive.org/web/20120611115223/http://www.thenewatlantis.com/publications/artificial-intelligence-and-human-nature). 1: 88–100. Archived from the original (http://www.thenewatlantis.com/publications/artificial-intelligence-and-human-nature) on 11 June 2012.

320. Rawlinson, Kevin. "Microsoft's Bill Gates insists AI is a threat" (http://www.bbc.co.uk/news/31047780) BBC News. Archived (https://web.archive.org/web/20150129183607/http://www.bbc.co.uk/news/31047780) from the original on 29 January 2015. Retrieved 30 January 2015.

321. Brooks, Rodney (10 November 2014). "artificial intelligence is a tool, not a threat" (https://web.archive.org/web/20141112130954/http://www.rethinkrobotics.com/artificial-intelligence-tool-threat/) Archived from the original (http://www.rethinkrobotics.com/artificial-intelligence-tool-threat/) on 12 November 2014.

322. Horst, Steven, (2005) "The Computational Theory of Mind" (http://plato.stanford.edu/entries/computational-mind/) in *The Stanford Encyclopedia of Philosophy*

323. This version is from Searle (1999), and is also quoted in Dennett 1991, p. 435. Searle's original formulation was "The appropriately programmed computer really is a mind, in the sense that computers given the right programs can be literally said to understand and have other cognitive states." (Searle 1980, p. 1). Strong AI is defined similarly by Russell & Norvig (2003 p. 947): "The assertion that machines could possibly act intelligently (or, perhaps better, act as if they were intelligent) is called the 'weak AI' hypothesis by philosophers, and the assertion that machines that do so are actually thinking (as opposed to simulating thinking) is called the 'strong AI' hypothesis."

324. Searle's Chinese room argument:
    - Searle 1980. Searle's original presentation of the thought experiment.
    - Searle 1999.

    Discussion:
    - Russell & Norvig 2003 pp. 958–960
    - McCorduck 2004, pp. 443–445
    - Crevier 1993, pp. 269–271

325. Robot rights:
    - Russell & Norvig 2003 p. 964
    - *BBC News* 2006

    Prematurity of:
    - Henderson 2007

    In fiction:
    - McCorduck (2004, p. 190-25) discusses *Frankenstein* and identifies the key ethical issues as scientific hubris and the suffering of the monster, i.e. robot rights.

326. Evans, Woody (2015). "Posthuman Rights: Dimensions of Transhuman Worlds" (http://revistas.ucm.es/index.php/TEKN/article/view/49072/46310) *Teknokultura*. Universidad Complutense, Madrid. Archived (https://web.archive.org/web/20161228094440/http://revistas.ucm.es/index.php/TEKN/article/view/49072/46310) from the original on 28 December 2016. Retrieved 5 December 2016.

327. maschafilm. "Content: Plug & Pray Film – Artificial Intelligence – Robots -" (http://www.plugandpray-film.de/en/content.html). *plugandpray-film.de*. Archived (https://web.archive.org/web/20160212040134/http://www.plugandpray-film.de/en/content.html) from the original on 12 February 2016.

328. Omohundro, Steve (2008). *The Nature of Self-Improving Artificial Intelligence* presented and distributed at the 2007 Singularity Summit, San Francisco, CA.

329. Technological singularity:
    - Vinge 1993
    - Kurzweil 2005
    - Russell & Norvig 2003, p. 963

330. Lemmons, Phil (April 1985). "Artificial Intelligence" (https://archive.org/stream/byte-magazine-1985-04/1985_04_BYT E_10-04_Artificial_Intelligence#page/n125/mode/2up) *BYTE*. p. 125. Archived (https://web.archive.org/web/201504 20115129/http://archive.org/stream/byte-magazine-1985-04/1985_04_BYTE_10-04_Artificial_Intelligence#page/n12 5/mode/2up) from the original on 20 April 2015. Retrieved 14 February 2015.

331. Transhumanism:
    - Moravec 1988
    - Kurzweil 2005
    - Russell & Norvig 2003, p. 963

332. AI as evolution:
    - Edward Fredkin is quoted in McCorduck (2004, p. 401).
    - Butler 1863
    - Dyson 1998

333. Anderson, Susan Leigh. "Asimov's "three laws of robotics" and machine metaethics." AI & Society 22.4 (2008): 477–493.

334. McCauley, Lee (2007). "AI armageddon and the three laws of robotics". *Ethics and Information Technology*. **9** (2): 153–164. CiteSeerX 10.1.1.85.8904 (https://citeseerx.ist.psu.edu/viewdoc/summary?doi=10.1.1.85.8904). doi:10.1007/s10676-007-9138-2 (https://doi.org/10.1007%2Fs10676-007-9138-2)

335. Galvan, Jill (1 January 1997). "Entering the Posthuman Collective in Philip K. Dick's "Do Androids Dream of Electric Sheep?"". *Science Fiction Studies* **24** (3): 413–429. JSTOR 4240644 (https://www.jstor.org/stable/4240644)

336. Buttazzo, G. (July 2001). "Artificial consciousness: Utopia or real possibility?" (http://ieeexplore.ieee.org/document/9 33500/?reload=true). *Computer (IEEE)*. **34** (7): 24–30. doi:10.1109/2.933500 (https://doi.org/10.1109%2F2.933500) Archived (https://web.archive.org/web/20161230092217/http://ieeexplore.ieee.org/document/933500/?reload=true) from the original on 30 December 2016. Retrieved 29 December 2016.

**AI textbooks**

- Hutter, Marcus (2005). *Universal Artificial Intelligence*. Berlin: Springer. ISBN 978-3-540-22139-5
- Jackson, Philip (1985). *Introduction to Artificial Intelligence* (2nd ed.). Dover. ISBN 0-486-24864-X
- Luger, George; Stubblefield, William (2004). *Artificial Intelligence: Structures and Strategies for Complex Problem Solving* (5th ed.). Benjamin/Cummings. ISBN 0-8053-4780-1
- Neapolitan, Richard; Jiang, Xia (2018). *Artificial Intelligence: With an Introduction to Machine Learning*. Chapman & Hall/CRC. ISBN 978-1-13850-238-3
- Nilsson, Nils (1998). *Artificial Intelligence: A New Synthesis*. Morgan Kaufmann. ISBN 978-1-55860-467-4
- Russell, Stuart J.; Norvig, Peter (2003), *Artificial Intelligence: A Modern Approach* (2nd ed.), Upper Saddle River, New Jersey: Prentice Hall, ISBN 0-13-790395-2
- Russell, Stuart J.; Norvig, Peter (2009). *Artificial Intelligence: A Modern Approach* (3rd ed.). Upper Saddle River, New Jersey: Prentice Hall. ISBN 0-13-604259-7.
- Poole, David; Mackworth, Alan; Goebel, Randy (1998). *Computational Intelligence: A Logical Approach*. New York: Oxford University Press. ISBN 0-19-510270-3
- Winston, Patrick Henry (1984). *Artificial Intelligence*. Reading, MA: Addison-Wesley. ISBN 0-201-08259-4
- Rich, Elaine (1983). *Artificial Intelligence*. McGraw-Hill. ISBN 0-07-052261-8
- Bundy, Alan (1980). *Artificial Intelligence: An Introductory Course* (2nd ed.). Edinburgh University Press. ISBN 0-85224-410-X

- Poole, David; Mackworth, Alan (2017). *Artificial Intelligence: Foundations of Computational Agents* (2nd ed.). Cambridge University Press. ISBN 9781107195394.

**History of AI**

- Crevier, Daniel (1993), *AI: The Tumultuous Search for Artificial Intelligence*, New York, NY: BasicBooks, ISBN 0-465-02997-3.
- McCorduck, Pamela (2004), *Machines Who Think* (2nd ed.), Natick, MA: A. K. Peters, Ltd. ISBN 1-56881-205-1.
- Newquist, HP (1994). *The Brain Makers: Genius, Ego, And Greed In The Quest For Machines That Think*. New York: Macmillan/SAMS. ISBN 0-672-30412-0.
- Nilsson, Nils (2009). *The Quest for Artificial Intelligence: A History of Ideas and Achievements*. New York: Cambridge University Press. ISBN 978-0-521-12293-1.

**Other sources**

- Asada, M.; Hosoda, K.; Kuniyoshi, Y.; Ishiguro, H.; Inui, T.; Yoshikawa, Y.; Ogino, M.; Yoshida, C. (2009). "Cognitive developmental robotics: a survey". *IEEE Transactions on Autonomous Mental Development*. **1** (1): 12–34. doi:10.1109/tamd.2009.2021702. Archived from the original on 4 October 2013.
- "ACM Computing Classification System: Artificial intelligence". *ACM*. 1998. Archived from the original on 12 October 2007. Retrieved 30 August 2007.
- Goodman, Joanna (2016). *Robots in Law: How Artificial Intelligence is Transforming Legal Services* (1st ed.). Ark Group. ISBN 978-1-78358-264-8.
- Albus, J. S. (2002). "4-D/RCS: A Reference Model Architecture for Intelligent Unmanned Ground Vehicles" (PDF). In Gerhart, G.; Gunderson, R.; Shoemaker, C. *Proceedings of the SPIE AeroSense Session on Unmanned Ground Vehicle Technology*. **3693**. pp. 11–20. Archived from the original (PDF) on 25 July 2004.
- Aleksander, Igor (1995). *Artificial Neuroconsciousness: An Update*. IWANN. Archived from the original on 2 March 1997. BibTex Archived 2 March 1997 at the Wayback Machine.
- Bach, Joscha (2008). "Seven Principles of Synthetic Intelligence". In Wang, Pei; Goertzel, Ben; Franklin, Stan. *Artificial General Intelligence, 2008: Proceedings of the First AGI Conference*. IOS Press. pp. 63–74. ISBN 978-1-58603-833-5.
- "Robots could demand legal rights". *BBC News*. 21 December 2006. Retrieved 3 February 2011.
- Brooks, Rodney (1990). "Elephants Don't Play Chess" (PDF). *Robotics and Autonomous Systems*. **6**: 3–15. doi:10.1016/S0921-8890(05)80025-9. Archived (PDF) from the original on 9 August 2007.
- Brooks, R. A. (1991). "How to build complete creatures rather than isolated cognitive simulators". In VanLehn, K. *Architectures for Intelligence*. Hillsdale, NJ: Lawrence Erlbaum Associates. pp. 225–239. CiteSeerX 10.1.1.52.9510.
- Buchanan, Bruce G. (2005). "A (Very) Brief History of Artificial Intelligence" (PDF). *AI Magazine*: 53–60. Archived from the original (PDF) on 26 September 2007.
- Butler, Samuel (13 June 1863). "Darwin among the Machines". Letters to the Editor. *The Press*. Christchurch, New Zealand. Retrieved 16 October 2014 – via Victoria University of Wellington.
- Clark, Jack (8 December 2015). "Why 2015 Was a Breakthrough Year in Artificial Intelligence". *Bloomberg News*. Archived from the original on 23 November 2016. Retrieved 23 November 2016. "After a half-decade of quiet breakthroughs in artificial intelligence, 2015 has been a landmark year. Computers are smarter and learning faster than ever."
- "AI set to exceed human brain power". *CNN*. 26 July 2006. Archived from the original on 19 February 2008.
- Dennett, Daniel (1991). *Consciousness Explained*. The Penguin Press. ISBN 0-7139-9037-6.
- Diamond, David (December 2003). "The Love Machine; Building computers that care". *Wired*. Archived from the original on 18 May 2008.
- Domingos, Pedro (2015). *The Master Algorithm: How the Quest for the Ultimate Learning Machine Will Remake Our World*. Basic Books. ISBN 9780465061921.
- Dowe, D. L.; Hajek, A. R. (1997). "A computational extension to the Turing Test". *Proceedings of the 4th Conference of the Australasian Cognitive Science Society*. Archived from the original on 28 June 2011.

- Dreyfus, Hubert (1972). *What Computers Can't Do*. New York: MIT Press. ISBN 0-06-011082-1.
- Dreyfus, Hubert; Dreyfus, Stuart (1986). *Mind over Machine: The Power of Human Intuition and Expertise in the Era of the Computer*. Oxford, UK: Blackwell. ISBN 0-02-908060-6.
- Dreyfus, Hubert (1992). *What Computers Still Can't Do*. New York: MIT Press. ISBN 0-262-54067-3.
- Dyson, George (1998). *Darwin among the Machines*. Allan Lane Science. ISBN 0-7382-0030-1.
- Edelman, Gerald (23 November 2007). "Gerald Edelman – Neural Darwinism and Brain-based Devices". *Talking Robots*. Archived from the original on 8 October 2009.
- Edelson, Edward (1991). *The Nervous System*. New York: Chelsea House. ISBN 978-0-7910-0464-7.
- Fearn, Nicholas (2007). *The Latest Answers to the Oldest Questions: A Philosophical Adventure with the World's Greatest Thinkers*. New York: Grove Press. ISBN 0-8021-1839-9.
- Gladwell, Malcolm (2005). *Blink*. New York: Little, Brown and Co. ISBN 0-316-17232-4.
- Gödel, Kurt (1951). *Some basic theorems on the foundations of mathematics and their implications*. Gibbs Lecture. In Feferman, Solomon, ed. (1995). *Kurt Gödel: Collected Works, Vol. III: Unpublished Essays and Lectures*. Oxford University Press. pp. 304–23. ISBN 978-0-19-514722-3.
- Haugeland, John (1985). *Artificial Intelligence: The Very Idea*. Cambridge, Mass.: MIT Press. ISBN 0-262-08153-9.
- Hawkins, Jeff; Blakeslee, Sandra (2005). *On Intelligence*. New York, NY: Owl Books. ISBN 0-8050-7853-3.
- Henderson, Mark (24 April 2007). "Human rights for robots? We're getting carried away". *The Times Online*. London.
- Hernandez-Orallo, Jose (2000). "Beyond the Turing Test". *Journal of Logic, Language and Information* **9** (4): 447–466. doi:10.1023/A:1008367325700.
- Hernandez-Orallo, J.; Dowe, D. L. (2010). "Measuring Universal Intelligence: Towards an Anytime Intelligence Test". *Artificial Intelligence Journal* **174** (18): 1508–1539. CiteSeerX 10.1.1.295.9079. doi:10.1016/j.artint.2010.09.006.
- Hinton, G. E. (2007). "Learning multiple layers of representation". *Trends in Cognitive Sciences* **11** (10): 428–434. doi:10.1016/j.tics.2007.09.004.
- Hofstadter, Douglas (1979). *Gödel, Escher, Bach: an Eternal Golden Braid*. New York, NY: Vintage Books. ISBN 0-394-74502-7.
- Holland, John H. (1975). *Adaptation in Natural and Artificial Systems*. University of Michigan Press. ISBN 0-262-58111-6.
- Howe, J. (November 1994). "Artificial Intelligence at Edinburgh University: a Perspective". Retrieved 30 August 2007.
- Hutter, M. (2012). "One Decade of Universal Artificial Intelligence". *Theoretical Foundations of Artificial General Intelligence*. Atlantis Thinking Machines. **4**. doi:10.2991/978-94-91216-62-6_5. ISBN 978-94-91216-61-9.
- James, William (1884). "What is Emotion". *Mind*. **9** (34): 188–205. doi:10.1093/mind/os-IX.34.188. Cited by Tao & Tan 2005.
- Kahneman, Daniel; Slovic, D.; Tversky, Amos (1982). *Judgment under uncertainty: Heuristics and biases*. New York: Cambridge University Press. ISBN 0-521-28414-7.
- Katz, Yarden (1 November 2012). "Noam Chomsky on Where Artificial Intelligence Went Wrong". *The Atlantic*. Retrieved 26 October 2014.
- "Kismet". MIT Artificial Intelligence Laboratory Humanoid Robotics Group. Retrieved 25 October 2014.
- Koza, John R. (1992). *Genetic Programming (On the Programming of Computers by Means of Natural Selection)*. MIT Press. ISBN 0-262-11170-5.
- Kleine-Cosack, Christian (October 2006). "Recognition and Simulation of Emotions" (PDF). Archived from the original (PDF) on 28 May 2008.
- Kolata, G. (1982). "How can computers get common sense?". *Science*. **217** (4566): 1237–1238. doi:10.1126/science.217.4566.1237. PMID 17837639.
- Kumar, Gulshan; Kumar, Krishan (2012). "The Use of Artificial-Intelligence-Based Ensembles for Intrusion Detection: A Review". *Applied Computational Intelligence and Soft Computing* **2012**: 1–20. doi:10.1155/2012/850160.
- Kurzweil, Ray (1999). *The Age of Spiritual Machines*. Penguin Books. ISBN 0-670-88217-8.
- Kurzweil, Ray (2005). *The Singularity is Near*. Penguin Books. ISBN 0-670-03384-7.
- Lakoff, George; Núñez, Rafael E. (2000). *Where Mathematics Comes From: How the Embodied Mind Brings Mathematics into Being*. Basic Books. ISBN 0-465-03771-2.

- Langley, Pat (2011). "The changing science of machine learning". *Machine Learning*. **82** (3): 275–279. doi:10.1007/s10994-011-5242-y
- Law, Diane (June 1994). *Searle, Subsymbolic Functionalism and Synthetic Intelligence* (Technical report). University of Texas at Austin. p. AI94-222. CiteSeerX 10.1.1.38.8384
- Legg, Shane; Hutter, Marcus (15 June 2007) *A Collection of Definitions of Intelligence* (Technical report). IDSIA. arXiv:0706.3639. 07-07.
- Lenat, Douglas; Guha, R. V. (1989). *Building Large Knowledge-Based Systems*. Addison-Wesley. ISBN 0-201-51752-3.
- Lighthill, James (1973). "Artificial Intelligence: A General Survey". *Artificial Intelligence: a paper symposium*. Science Research Council.
- Lucas, John (1961). "Minds, Machines and Gödel". In Anderson, A.R. *Minds and Machines*. Archived from the original on 19 August 2007. Retrieved 30 August 2007.
- Lungarella, M.; Metta, G.; Pfeifer, R.; Sandini, G. (2003). "Developmental robotics: a survey". *Connection Science*. **15** (4): 151–190. CiteSeerX 10.1.1.83.7615. doi:10.1080/09540090310001655110
- Maker, Meg Houston (2006). "AI@50: AI Past, Present, Future". Dartmouth College. Archived from the original on 3 January 2007. Retrieved 16 October 2008.
- Markoff, John (16 February 2011). "Computer Wins on 'Jeopardy!': Trivial, It's Not". *The New York Times*. Retrieved 25 October 2014.
- McCarthy, John; Minsky, Marvin; Rochester, Nathan; Shannon, Claude (1955). "A Proposal for the Dartmouth Summer Research Project on Artificial Intelligence". Archived from the original on 26 August 2007. Retrieved 30 August 2007..
- McCarthy, John; Hayes, P. J. (1969). "Some philosophical problems from the standpoint of artificial intelligence". *Machine Intelligence*. **4**: 463–502. CiteSeerX 10.1.1.85.5082. Archived from the original on 10 August 2007. Retrieved 30 August 2007.
- McCarthy, John (12 November 2007). "What Is Artificial Intelligence?". Archived from the original on 18 November 2015.
- Minsky, Marvin (1967). *Computation: Finite and Infinite Machines*. Englewood Cliffs, N.J.: Prentice-Hall. ISBN 0-13-165449-7.
- Minsky, Marvin (2006). *The Emotion Machine*. New York, NY: Simon & Schusterl. ISBN 0-7432-7663-9.
- Moravec, Hans (1988). *Mind Children*. Harvard University Press. ISBN 0-674-57616-0.
- Norvig, Peter (25 June 2012). "On Chomsky and the Two Cultures of Statistical Learning". Peter Norvig. Archived from the original on 19 October 2014.
- NRC (United States National Research Council) (1999). "Developments in Artificial Intelligence". *Funding a Revolution: Government Support for Computing Research*. National Academy Press.
- Needham, Joseph (1986). *Science and Civilization in China*. Volume 2. Caves Books Ltd.
- Newell, Allen; Simon, H. A. (1976). "Computer Science as Empirical Inquiry: Symbols and Search". *Communications of the ACM*. **19** (3): 113–126. doi:10.1145/360018.360022. Archived from the original on 7 October 2008.
- Nilsson, Nils (1983). "Artificial Intelligence Prepares for 2001" (PDF). *AI Magazine*. **1** (1). Presidential Address to the Association for the Advancement of Artificial Intelligence
- O'Brien, James; Marakas, George (2011) *Management Information Systems* (10th ed.). McGraw-Hill/Irwin. ISBN 978-0-07-337681-3
- O'Connor, Kathleen Malone (1994). "The alchemical creation of life (takwin) and other concepts of Genesis in medieval Islam". University of Pennsylvania.
- Oudeyer, P-Y. (2010). "On the impact of robotics in behavioral and cognitive sciences: from insect navigation to human cognitive development" (PDF). *IEEE Transactions on Autonomous Mental Development*. **2** (1): 2–16. doi:10.1109/tamd.2009.2039057
- Penrose, Roger (1989). *The Emperor's New Mind: Concerning Computer, Minds and The Laws of Physics*. Oxford University Press. ISBN 0-19-851973-7.
- Picard, Rosalind (1995). *Affective Computing* (PDF) (Technical report). MIT. 321. Lay summary – *Abstract*.
- Poli, R.; Langdon, W. B.; McPhee, N. F. (2008). *A Field Guide to Genetic Programming*. Lulu.com. ISBN 978-1-4092-0073-4 – via gp-field-guide.org.uk.

- Rajani, Sandeep (2011). "Artificial Intelligence – Man or Machine" (PDF). *International Journal of Information Technology and Knowledge Management* **4** (1): 173–176. Archived from the original (PDF) on 18 January 2013.
- Ronald, E. M. A. and Sipper, M. Intelligence is not enough: On the socialization of talking machines, Minds and Machines, vol. 11, no. 4, pp. 567–576, November 2001.
- Ronald, E. M. A. and Sipper, M. What use is a Turing chatterbox?, Communications of the ACM, vol. 43, no. 10, pp. 21–23, October 2000.
- Searle, John (1980). "Minds, Brains and Programs". *Behavioral and Brain Sciences* **3** (3): 417–457. doi:10.1017/S0140525X00005756 Archived from the original on 18 January 2010.
- Searle, John (1999). *Mind, language and society*. New York, NY: Basic Books. ISBN 0-465-04521-9. OCLC 231867665.
- Shapiro, Stuart C. (1992). "Artificial Intelligence". In Shapiro, Stuart C. *Encyclopedia of Artificial Intelligence* (PDF) (2nd ed.). New York: John Wiley. pp. 54–57. ISBN 0-471-50306-1.
- Simon, H. A. (1965). *The Shape of Automation for Men and Management*. New York: Harper & Row.
- Skillings, Jonathan (3 July 2006). "Getting Machines to Think Like Us". *cnet*. Retrieved 3 February 2011.
- Solomonoff, Ray (1956). *An Inductive Inference Machine* (PDF). Dartmouth Summer Research Conference on Artificial Intelligence – via std.com, pdf scanned copy of the original. Later published as Solomonoff, Ray (1957). "An Inductive Inference Machine". *IRE Convention Record*. Section on Information Theory, part 2. pp. 56–62.
- Tao, Jianhua; Tan, Tieniu (2005). *Affective Computing and Intelligent Interaction*. Affective Computing: A Review. LNCS 3784. Springer. pp. 981–995. doi:10.1007/11573548.
- Tecuci, Gheorghe (March–April 2012). "Artificial Intelligence". *Wiley Interdisciplinary Reviews: Computational Statistics*. Wiley. **4** (2): 168–180. doi:10.1002/wics.200.
- Thro, Ellen (1993). *Robotics: The Marriage of Computers and Machines*. New York: Facts on File. ISBN 978-0-8160-2628-9.
- Turing, Alan (October 1950), "Computing Machinery and Intelligence", *Mind*, **LIX** (236): 433–460, doi:10.1093/mind/LIX.236.433, ISSN 0026-4423.
- van der Walt, Christiaan; Bernard, Etienne (2006). "Data characteristics that determine classifier performance" (PDF). Archived from the original (PDF) on 25 March 2009. Retrieved 5 August 2009.
- Vinge, Vernor (1993). "The Coming Technological Singularity: How to Survive in the Post-Human Era".
- Wason, P. C.; Shapiro, D. (1966). "Reasoning". In Foss, B. M. *New horizons in psychology*. Harmondsworth: Penguin.
- Weizenbaum, Joseph (1976). *Computer Power and Human Reason*. San Francisco: W.H. Freeman & Company. ISBN 0-7167-0464-1.
- Weng, J.; McClelland; Pentland, A.; Sporns, O.; Stockman, I.; Sur, M.; Thelen, E. (2001). "Autonomous mental development by robots and animals" (PDF). *Science*. **291** (5504): 599–600. doi:10.1126/science.291.5504.599 – via msu.edu.
- "Applications of AI". *www-formal.stanford.edu*. Retrieved 25 September 2016.

**Further reading**

- DH Autor, 'Why Are There Still So Many Jobs? The History and Future of Workplace Automation' (2015) 29(3) Journal of Economic Perspectives 3.
- TechCast Article Series, John Sagi, "Framing Consciousness"
- Boden, Margaret, *Mind As Machine*, Oxford University Press, 2006
- Gopnik, Alison, "Making AI More Human: Artificial intelligence has staged a revival by starting to incorporate what we know about how children learn", *Scientific American*, vol. 316, no. 6 (June 2017), pp. 60–65.
- Johnston, John (2008) *The Allure of Machinic Life: Cybernetics, Artificial Life, and the New AI*, MIT Press
- Marcus, Gary, "Am I Human?: Researchers need new ways to distinguish artificial intelligence from the natural kind", *Scientific American*, vol. 316, no. 3 (March 2017), pp. 58–63. *Multiple* tests of artificial-intelligence efficacy are needed because, "just as there is no single test of athletic prowess, there cannot be one ultimate test of intelligence." One such test, a "Construction Challenge", would test perception and physical action—"two important elements of intelligent behavior that were entirely absent from the original Turing test." Another proposal has been to give machines the same standardized tests of science and other disciplines that schoolchildren take. A so far insuperable stumbling block to artificial intelligence is an incapacity for reliable disambiguation. "[V]irtually every sentence [that

people generate] is ambiguous, often in multiple ways." A prominent example is known as the "pronoun disambiguation problem": a machine has no way of determining to whom or what a pronoun in a sentence—such as "he", "she" or "it"—refers.

- E McGaughey, 'Will Robots Automate Your Job Away? Full Employment, Basic Income, and Economic Democracy' (2018) SSRN, part 2(3).
- Myers, Courtney Boyd ed. (2009). "The AI Report". *Forbes* June 2009
- Raphael, Bertram (1976). *The Thinking Computer*. W.H.Freeman and Company ISBN 0-7167-0723-3.
- Serenko, Alexander (2010). "The development of an AI journal ranking based on the revealed preference approach" (PDF). *Journal of Informetrics* **4** (4): 447–459. doi:10.1016/j.joi.2010.04.001
- Serenko, Alexander; Michael Dohan (2011). "Comparing the expert survey and citation impact journal ranking methods: Example from the field of Artificial Intelligence" (PDF). *Journal of Informetrics* **5** (4): 629–649. doi:10.1016/j.joi.2011.06.002
- Sun, R. & Bookman, L. (eds.), *Computational Architectures: Integrating Neural and Symbolic Processes*. Kluwer Academic Publishers, Needham, MA. 1994.
- Tom Simonite (29 December 2014). "2014 in Computing: Breakthroughs in Artificial Intelligence". *MIT Technology Review*.

**External links**

- What Is AI? – An introduction to artificial intelligence by John McCarthy—a co-founder of the field, and the person who coined the term.
- The Handbook of Artificial Intelligence Volume I by Avron Barr and Edward A. Feigenbaum (Stanford University)
- "Artificial Intelligence". *Internet Encyclopedia of Philosophy*
- Thomason, Richmond. "Logic and Artificial Intelligence". In Zalta, Edward N. *Stanford Encyclopedia of Philosophy*
- AI at Curlie (based on DMOZ)
- AITopics – A large directory of links and other resources maintained by the Association for the Advancement of Artificial Intelligence, the leading organization of academic AI researchers.
- List of AI Conferences – A list of 225 AI conferences taking place all over the world.

---

## Author Comment (AC5) · 10 Jun 2018

I used AI to research earthquake, not make a Robot....then do not worry
* * *

---

## Author Comment (AC7) · 5 Jul 2018

Final responds to two reviewers from Author; I have given the responds in GI discussion. Now the responds are the same You can find in the GI discussion Addressing to the comments of Pro. Dr Eppelbaum Dear Pro. Dr Eppelbaum (1) This MS, without hesitation, will be interesting for GI readers. ANS: Thank you. (2) However, without significant and careful English editing, reviewing of this MS is impossible. ANS: I sent to an editor to rewrite the English for the paper and arrange the submitted format ===================== Dear Reviewer#1 Thank you for your comments. Now I

address your comments point-by-point and marked the changes with red words. (1) I wrote more clear as follows; The aim of this paper is to determine whether the EEW can be performed by a better real-time and on-line performable training method in BPNN rather than the previous works as stated. The microseismic data in the records are firstly used as training data for the BPNN model; in each station shown, the behaviour of microseismic data at each station records the ray tracing path, allowing for the prediction of upcoming signal. When the large predicted errors are presented, then it is expected that the behaviour of the microseismic data has changed because of this model reflecting the pattern of microseismic data. (2) Because the earthquake forces mostly acted on the center of gravity of the sliding soil mass, and the influences of vertical ground motions were on the seismic-induced displacements of the structures. Therefore I wrote the reasons more clear as follows; The vertical component of an earthquake was the most dangerous because the earthquake forces mostly acted on the centre of gravity of the sliding soil mass, and the influences of vertical ground motions were on the seismic-induced displacements of the structures (Sawicki, et al. 2007; Zhao, et al. 2017). (3) I re-wrote in the text as follows; surveying of the consideration of local building damages from past events under different local geological conditions. (4) Thank you. I have done it by attached file. PS: red words are added by author. Please also note the supplement to this comment: https://www.geosci-instrum-method-data-syst-discuss.net/gi-2018-13/gi-2018-13- AC4-supplement.pdf

By the way in section 2, for reader to understand clearly, I add some statements with red words. I also upload the revise paper.

Jyh-Woei Lin, Chun-Tang Chao, Juing-Shian Chiou 05, July, 2018, Taiwan =============== Dear Reviewer 2

Thank you for you comments. Now I address your comments pointby- point. Author: Lin, Jyh-Woei 26, May, 2018 1. On page 1, no line 13: The word 'a' in the sentence 'a trade-off decision-making process ......' should be capital. ANS: In Abstract, I have change as follows; Abstract. A new Elementary Modified Levenberg– Marquardt Algorithm (M-LMA) was used to minimise backpropagation errors in training a backprop-agation neural network (BPNN) to predict the records related to the Chi- Chi earth-quake from four seismic stations, Station-TAP003, Station-TAP005, Station- TCU084 and Station-TCU078, with the learning rates of 0.3, 0.05, 0.2 and 0.28, respectively. For these four recording stations, the M-LMA has been shown to produce smaller pre-dicted errors compared to Levenberg–Marquardt Algorithm (LMA). A sudden predicted error could be an indicator for Early Earthquake Warning (EEW), which indicated the initiation of strong motion due to large earthquakes. A Trade-Off Decision- Making Process with BPNN (TDPB), using two alarms, adjusted the threshold of the magni-tude of predicted error without a mistaken alarm. This approach was not necessary to consider the problems of characterising the wave phases and pre-processing, but did not require complex hardware; an existing seismic monitoring network-covered re-searched area was already sufficient for these purposes. In page 6, line 30, the text is also changed as follows; A decision-making process called " Trade-Off Decision- Mak-ing Process with BPNN (TDPB)" was performed. In this study, the past records of the Chi-Chi earthquake was examined by TDPB, and then the thresholds and were subjec-tively determined. 2.The yellow colored spot in figure 1 is too light to be distinguished. ANS: I have also change in the figure caption for Figure.1 with more clear colors as follows; Figure 1 The figure shows the position of Chelungpu fault (No.11) on a map of Taiwan. Slip on this fault caused the Chi-Chi earthquake, which occurred at 01:47:15 on September 21, 1999 (TST), at a depth of 8.00 km, with a Richter magnitude (ML) of 7.3. The epicentre was at the coordinates (23.85_ N, 120.82_ E) (Orange-colour spot near the Chelungpu fault for No.11). The four corresponding positions of the research stations are shown by a dark blue coloured spot (Station-TAP003), baby blue coloured (Station-TAP005) spot, red coloured spot (Station-TCU084) and dark red coloured spot (Station-TCU078) in this figure. Station-TCU078 is very close to the epicentre. 3.Figure 4 which is mentioned in page 7 can't be found in figure captions.

---

## Editor Comment (EC1) · L.V. Eppelbaum (Editor) · 4 Aug 2018

Dear Editors,

I have browsed through the articles of the Geoscientific Instrumentation, Methods and Data Systems Journal. A paper called "Backpropagation Neural Network as Earthquake Early Warning Tool using a new Elementary Modified Levenberg–Marquardt Algorithm to minimise Backpropagation Errors" [1] caught my attention.

It seems the interactive discussion is closed for non authors, therefore this mail. Is it

already accepted? I would strongly recommend to have a more in-depth look at this paper for the following reasons:

+ The paper appears unscientific

+ The author self-references to a Journal (Hikari) where the author is one of the only contributors. I would not consider the author's papers from this Journal [2][3], which are referenced throughout their submitted paper, as peer-reviewed nor scientifically strong.

+ The Journal Hikari, which I am not familiar with, appears on Beall's List of predatory publishers. [4]

+ It is not a good paper!

+ The paper is unstructured and the sentences are overly complex, which makes the paper incomprehensible

+ Some sentences do not even make sense.

+ It does not get clear what they want to show. What is their contribution?

+ Instead of explaining everything in their paper, they refer to their previous publication, which is not (!) enlightening and of similar poor quality.

+ The figures are of poor quality, results cannot not be deduced from the figures.

+ Content

+ There is no evidence that their method is better! It is not even clear if it is "their" method

+ They state that they have a better predictive error: They never show values.

+ They do not accurately explain how they train their network (What is their input size? what is the output? how many samples are predicted?)

+ They do not follow the basic principles of training of a neural network (training set,

test set). At least it does not become clear.

+ Prediction of time series data is complex. I do not believe that they can predict microseismic data with a two layer ANN and it does not become clear from the paper how they can do it.

In general, please recommend to your reviewers that they provide a short summary of what they understood from the paper. I do not want to imply that they did not understand the content but it makes it a more thorough review.

[1] https://www.geosci-instrum-method-data-syst-discuss.net/gi-2018-13/

[2] http://www.m-hikari.com/asms/asms2017/asms1-2017/p/linASMS1-2017-2.pdf

[3] http://www.m-hikari.com/asms/asms2017/asms1-2017/p/linASMS1-2017.pdf

[4] https://beallslist.weebly.com/

Best,

Matthias Meyer

---

## Author Comment (AC8) · 4 Aug 2018

This paper was reviewed by two reviewers. They think this paper can be accepted. I think it can be accepted. Lin, Jyh-Woei